# Age-progressive interplay of HSP-proteostasis, ECM-cell junctions and biomechanics ensures *C. elegans* astroglial architecture

Francesca Coraggio[1], Mahak Bhushan[1,6], Spyridon Roumeliotis[1,6], Francesca Caroti[1], Carlo Bevilacqua [2], Robert Prevedel [1,2,3,4,5] & Georgia Rapti [1,3,4] ✉

Tissue integrity is sensitive to temperature, tension, age, and is sustained throughout life by adaptive cell-autonomous or extrinsic mechanisms. Safeguarding the remarkably-complex architectures of neurons and glia ensures age-dependent integrity of functional circuits. Here, we report mechanisms sustaining the integrity of *C. elegans* CEPsh astrocyte-like glia. We combine large-scale genetics with manipulation of genes, cells, and their environment, quantitative imaging of cellular/ subcellular features, tissue material properties and extracellular matrix (ECM). We identify mutants with age-progressive, environment-dependent defects in glial architecture, consequent disruption of neuronal architecture, and abnormal aging. Functional loss of epithelial Hsp70/Hsc70-cochaperone BAG2 causes ECM disruption, altered tissue biomechanics, and hypersensitivity of glia to environmental temperature and mechanics. Glial-cell junctions ensure epithelia-ECM-CEPsh glia association. Modifying glial junctions or ECM mechanics safeguards glial integrity against disrupted BAG2-proteostasis. Overall, we present a finely-regulated interplay of proteostasis-ECM and cell junctions with conserved components that ensures age-progressive robustness of glial architecture.

Maintaining tissue integrity throughout age progression and across diverse environments is an outcome of the evolution of multicellular life. Interacting tissues must coordinate growth, shape changes, movement, and exerted forces in time and space. Cells are susceptible to temperature and metabolic changes, and continuously subject to mechanical loads, gravity, hydrostatic pressure, shear stress[1–4]. Such factors influence animals at various scales, from molecular to tissue organization[5,6]. To sustain tissue integrity, that declines with age, cells need to adaptively respond to faced stresses by cell-autonomous or extrinsic mechanisms[7]. They can respond to environmental factors and age progression by controlling composition of intercellular junctions or surrounding extracellular matrix (ECM)[4,8]. Studies in epithelial tissues indicate that attachments to neighboring tissues or the ECM affect epithelial remodeling and force-delivery to cells, proving critical for tissue organization[3,6,9–11]. The complex mixture of ECM structural and functional proteins can provide spatio-temporal cues for sculpting epithelial morphogenesis and maintaining tissue organization[6,9,11]. The effect of stressors on tissue architecture often depends on ECM

[1]Developmental Biology Unit, European Molecular Biology Laboratory, Heidelberg, Germany. [2]Cell Biology and Biophysics Unit, European Molecular Biology Laboratory, Heidelberg, Germany. [3]Epigenetics and Neurobiology Unit, European Molecular Biology Laboratory, Rome, Italy. [4]Interdisciplinary Center of Neurosciences, Heidelberg University, Heidelberg, Germany. [5]German Center for Lung Research (DZL), Heidelberg, Germany. [6]These authors contributed equally: Mahak Bhushan, Spyridon Roumeliotis. ✉e-mail: grapti@embl.de

composition and proteostasis[4,5], which can undergo remodeling through balanced production and turnover[9,11,12]. Proteostasis needs to be adapted in age progression[10,13] and declines in aging or neurodegenerative diseases[7,14], leading to cognitive dysfunction[4,14,15]. ECM proteostasis can be regulated by temperature, metabolic shifts, and mechanical strains[4,5] and environmental features such as mechanics can affect epithelial tissue morphogenesis and integrity, throughout development[6,10]. Investigating how proteostasis, ECM, and cell interactions integrate responses to external factors, to sustain tissue integrity against pathology, is key to comprehending tissue biology.

Nervous system tissues display a remarkable complexity of high-order cell interactions, and functions, that should be maintained throughout life. The interconnected architecture of circuits is primarily patterned during embryogenesis, through an interplay of cell recognition, glia-neuron interactions and molecular signaling, instructing precise positioning of neurons, glial cells, processes, and connections[16–19]. Failure to preserve integrity of these highly specialized morphologies and connections results in compromised circuit functions, neuropathology and neurodegeneration[1,2,13,20]. Therefore, it is crucial to understand the molecular mechanisms underlying maintenance of neuronal and glial cell architecture to ensure proper circuit structure and function, as age progresses. Studies focusing on neurons' susceptibility to environmental challenges, including mechanical stress during growth, movement, or injuries, identified mechanisms dedicated to neuron maintenance[20–22]. In *C. elegans*, the brain-like neuropil composed of 181 neurites is patterned in embryogenesis and completes its formation before the 2nd larval stage (L2). Afterwards, the neuropil's components enlarge relative to animal size to maintain the underlying architecture. The neurons largely maintain their structural and functional characteristics, correct positions and connections within their ensembles. This neuronal maintenance requires active, dedicated mechanisms that differ from those acting in development: Ig-containing secreted or transmembrane proteins, L1 CAM and FGFR orthologs sustain positioning of neurons, axons[22], while hemidesmosomes and spectrin protect axons from degeneration[21]. Disruption of these maintenance mechanisms causes neuronal defects sometimes starting from L2 stage and worsening as the animal matures[22,23]. Less is known about mechanisms maintaining glial cell architecture. Specialized glia cells support neurons' development, physical protection, repair, degeneration, and function[19]. Similar to the neuronal deficits, abnormalities in fine glial cell morphologies relate to neuropathology[13]. Glial delicate, ramified processes formed in early development may be vulnerable to mechanical strain and need to withstand elastic deformations during organismal growth. How glial cells face and withstand mechanical stress and other environmental changes to sustain their architecture throughout the animal's life is poorly understood. Furthermore, current techniques to measure mechanical properties such as elasticity and viscosity non-invasively, in vivo, are lacking[24]. Combining such biophysical measurements with studies of genetics and animal physiology would allow for a unique understanding of the interactions between genes, cells and tissue material properties in physiological and pathological conditions of complex tissues, such as the nervous system.

These questions can be addressed using *C. elegans*, a tractable model that allows large-scale genetics and single-gene manipulations. Its invariantly structured and mapped nervous system allows to study morphogenesis, maintenance, and interactions by visualizing and manipulating in single-cell resolution neurons and glial cells, and their subcellular features[18,25,26]. Its brain neuropil, composed of 181 axons and 4 CEPsh glia, provides an in vivo platform to study circuit assembly and maintenance. We previously uncovered neuron-glia interactions underlying the assembly of functional circuit architecture. Embryonic CEPsh glia pioneer brain assembly, employing conserved cues to drive axon pathfinding of pioneer and follower neurons[18]. Post-embryonic CEPsh glia also regulate synaptic positioning and neurotransmission[27].

How CEPsh glia affect neuronal and axon integrity throughout life and growth, beyond synaptic distribution, is less clear.

In prior studies, we and others established that CEPsh glia and vertebrate astrocytes share functional, molecular, and morphological properties. They share transcription factors driving their fate[27], conserved guidance cues driving pathfinding[18], form tripartite synapses[28], influence synapse formation[26], regulate neurotransmission and present related transcriptomic profiles[29]. On a larger scale, although not firmly sealing the neuropil, the CEPsh glial sheath provides a physical barrier between brain axons, basement membrane, adjacent tissues and the body cavity. This is reminiscent of blood-brain barriers in other species[30,31]. CEPsh glia can provide a single-cell-resolution experimental system to study in vivo molecular mechanisms of astroglia architecture.

CEPsh glia form a specialized architecture to support their function. We previously uncovered that CEPsh form early, radial-glia-like processes, later transforming into astrocyte-like, ramified membranes, associating with axons and synapses[18,28]. While CEPsh interior membranes appose axons, their external side contacts epithelia and a basement membrane, separating them from muscle cells[28]. Certain cues from the epithelial cells and the basement membrane regulate their location/ shape[32]. Yet, how CEPsh glia maintain their complex architecture and interactions with their neighboring tissues throughout life and under environmental challenges remains largely elusive.

In this work, we investigate mechanisms maintaining CEPsh glia architecture and interactions. We performed genetic screens and isolated mutants presenting ectopic CEPsh glia membranes, with abnormal architecture. These glial changes are age-progressive, environment-dependent, and result in neuron, axon, and synapse mispositioning or degeneration. The material properties of tissues neighboring the CEPsh glial cells differ in mutants compared to wild-type animals, as we measured by Brillouin microscopy. The causal genetic alteration affects the HSP-co-chaperone UNC-23/BAG2, which acts from epithelia to ensure ECM architecture and the integrity of CEPsh glia membranes. CEPsh glia associate with ECM and epithelial junctions and present cell-junction components. In *unc-23* mutants, partially disrupting glial cell-junction components appears to disconnects CEPsh glia from their disrupted neighbors and safeguards their integrity, as does manipulating the animal's exoskeleton and the mechanics of the tissue's environment. Therefore—by combining advanced genetics, live quantitative imaging, biophysical measurements and manipulation of genes, cells, and their environment—we uncover a key interplay of ECM proteostasis, glial-cell junctions and environmental factors of temperature and material properties, that ensures maintaining of CEPsh glia integrity.

## Results

### CEPsh glia grow congruently with neighboring tissues to envelop the brain

We and others established that CEPsh glia drive development and function of the *C. elegans* brain neuropil, named *nerve ring*[18,27,29]. They pioneer brain embryonic assembly and affect neurite development[18,27], synapse localization[26,32], and neurotransmission[29]. CEPsh glia develop specialized morphologies to support their functions. Each one has an anterior process and a posterior membrane sheath associating with axons and synapses (Fig. 1 and Supplementary Fig. 1). CEPsh glia in the embryo grow thin, non-branching processes, that then transform in astrocyte-like, ramified membranes, associating with axons and synapses[18,28]. This transformation is largely established by the L1 stage (Fig. 1a). Key characteristics of the mature architecture of CEPsh glia, detectable in the mature neuropil by connectomics[28,33] are largely established before the L2 larval stage: (1) the ramifications of the early non-branching membranes of CEPsh glia (Fig. 1a), (2) the membrane tiling of the 4 CEPsh glia to generate a circumferential ring of membranes enveloping the brain axonal *nerve ring* (Fig. 1a), (3) the pairing

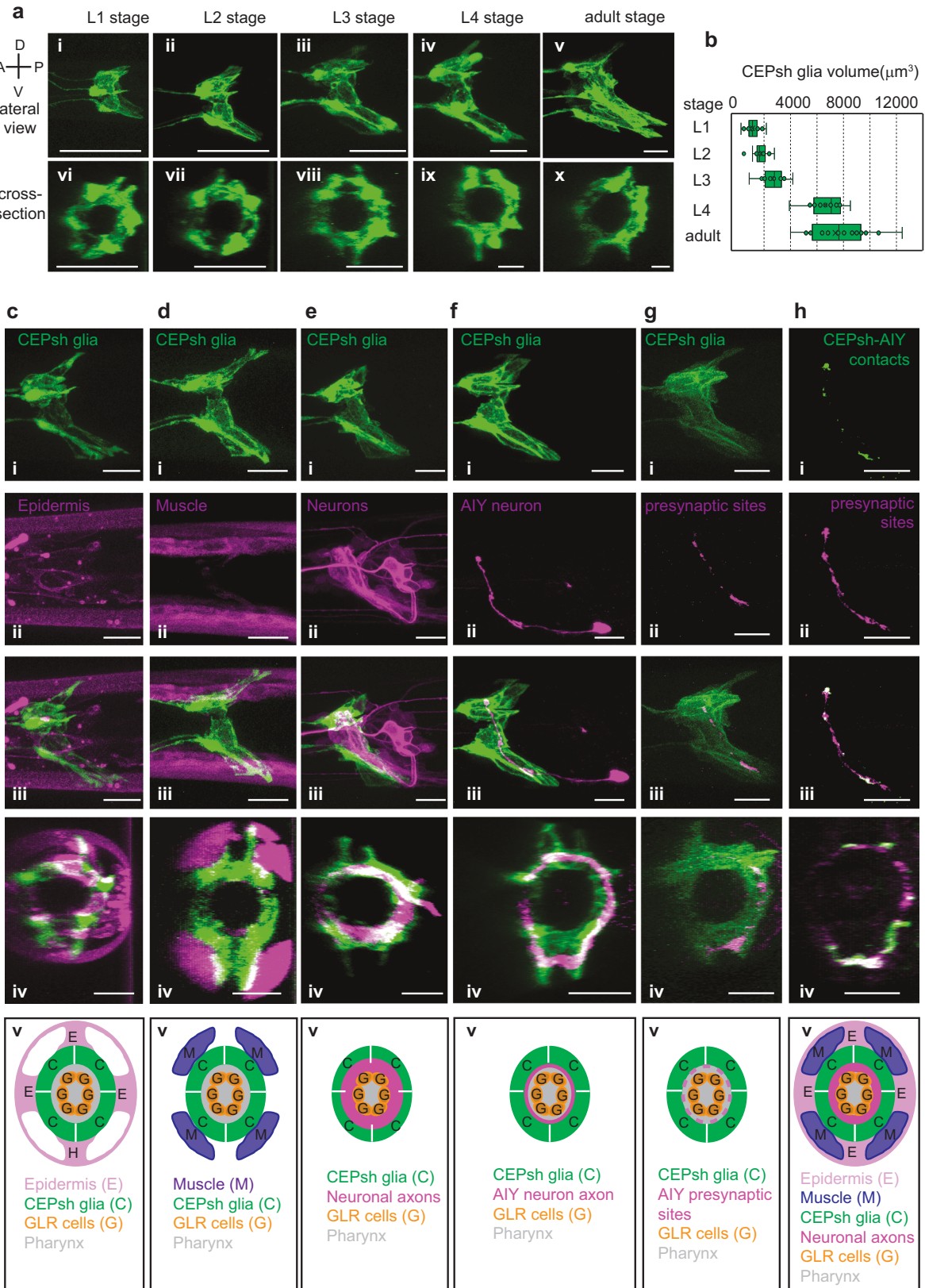

**Fig. 1 | CEPsh glia grow congruently with muscle and epithelia, to envelop axons and synapses. a**, **b** Wild-type CEPsh glia grow throughout larval and adult stages, expanding their membrane volume (**b**). $n = 14$ animals per stage. Primary data are presented in the provided Source Data file. CEPsh glia, in L4 animals, appose to epidermis (**c**), muscle (**d**), and envelop brain-neuropil axons (**e**) such as the AIY axon (**f**) and can associate closely with RAB-3-labeled presynaptic sites (here

of the AIY neuron) (**a–h**) (CEPsh; green, others pseudocolored magenta). Cross-sectional views in c(iv)-h(iv), models in c(v)-h(v). Reporters as listed in "Methods", Supplementary Tables 1 and 3. Scale bars, 10 μm. D dorsal, V ventral, A anterior, P posterior (also see Supplementary Movies 1–3). In box-whisker plots, the dots, center lines, X symbols, box limits, whiskers represent respectively individual values, means, 25th and 75th percentiles, maximum and minimum values.

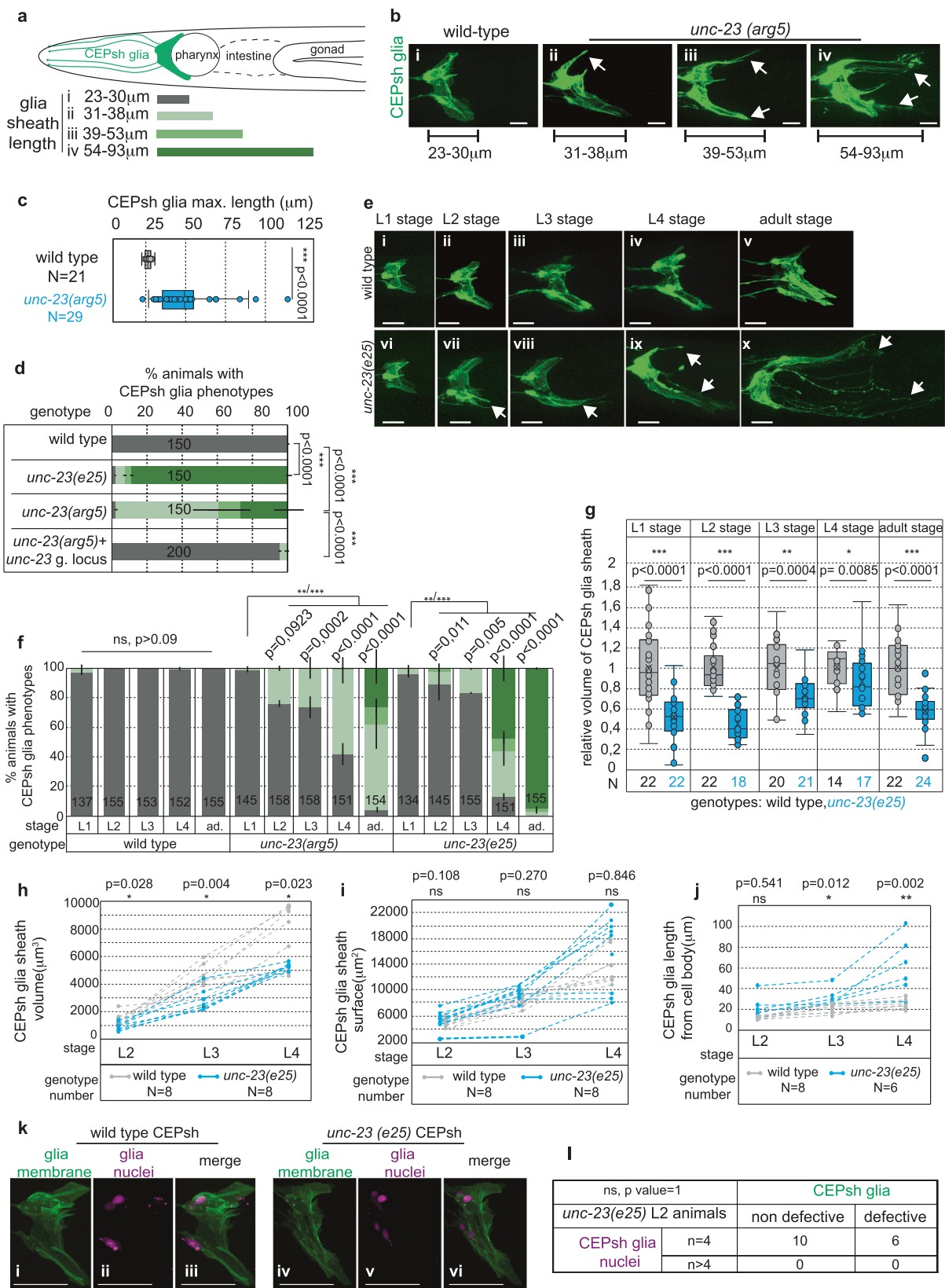

of CEPsh glial membrane processes with axons to establish tripartite-synapse ensheathment (detected by connectomics throughout the animal's stages)[33]. Beyond the L2 stage, the CEPsh glial cells expand their surface in relation to animal growth, to maintain this already-established architecture (Fig. 1b). CEPsh glia growth follows animal growth, in congruence with neighboring epithelia and mesoderm.

CEPsh glial sheath membranes are juxtaposed to the external epidermis (Fig. 1c and Supplementary Movie 1). Dorsal, ventral and lateral epidermal ridges extending from the cuticle, juxtapose the CEPsh glial sheaths directly without a basement membrane in-between[28]. Outside these ridges, the CEPsh glial sheath neighbors four muscle quadrants (Fig. 1d and Supplementary Movie 2), separated from them by an ECM

**Fig. 2 | CEPsh glia present age-progressive loss of membrane sheath integrity in *unc-23* mutants.** CEPsh glia (**a**, schematics) have defective morphology, with increased membrane length in *unc-23* L4 mutants (*n* = 29) compared to WT animals (*n* = 21) (**b-c**). **d** *arg5* defects are mimicked by *unc-23(e25)* mutant and rescued by the wild-type *unc-23* genomic locus (g. locus). **e, f** Defects of glial cell length aggravate with age progression.(ad, adults). **g–j** Compared to wild-type animals, in *unc-23* mutants, CEPsh glia membranes show abnormal decrease of volume throughout stages (**g, h**) and increase of their length but not their surface, as assessed quantitatively. **k, l** CEPsh glia have normal nuclei numbers in L2 stage of *unc-23* mutant animals, at the start of establishment of their membrane defects. **b, e, k** glia

membrane, green; glia nuclei, magenta. Arrows, membrane defects. *p* values: *** for ≤0.0001; **≤0.001, *≤0.01; unpaired two-sided *t*-test (**c**, **g–j**), two-sided Fisher's exact test (**d–f**, **l**). **l** *p* value in contingency table = 1.0000 (two-tailed unpaired Chi-square test). Error bars represent mean ± standard deviation. Total animal numbers are noted inside histogram bars or dot-plot graphs. Number of independent experiments for each condition is *n* = 3, unless otherwise noted. Dots, limits, lines, whiskers in box-whisker plots are defined as in Fig. 1. The total number of animals per condition and independent experiments, the primary data, and statistical analysis (including two-way ANOVA statistical analysis) are presented in the Source Data. Scale bar, animal axes, reporters defined as in Fig. 1.

basement membrane[28]. Underneath the epithelial and mesodermal layers, the post-embryonic ramified CEPsh glia membranes ensheath the neuropil (Fig. 1e and Supplementary Movie 3), associating with individual axons (Fig. 1f). They also extend processes ensheathing synaptic terminals as observed by trans-synaptic labeling of GFP reconstitution across synaptic partners (Fig. 1g, h), and electron microscopy[28]. This precise morphology of CEPsh glia is key for glial-neuron interactions in brain architecture and function.

### *unc-23/BAG2* mutants suffer age-progressive disruption of CEPsh glia

To uncover mechanisms regulating glial membrane architecture, we performed a genetic screen, seeking mutants with abnormal CEPsh glial morphology ("Methods"). Among others, we isolated mutant *arg5*, exhibiting highly penetrant defects in CEPsh glial morphology with membranes occupying ectopic posterior territories up to 3 times farther from the cell body compared to synchronized wild-type animals (Fig. 2a–c). To identify the causal mutation, we subjected *arg5* mutants to whole genome sequencing, SNP mapping, and rescue experiments ("Methods", Supplementary Tables 4 and 7). We identified a G to A mutation in the *unc-23* gene, altering a conserved glutamic acid to a lysine (Glu340Lys) in the dimerization domain of its vertebrate protein homolog[34]. CEPsh glial defects of *arg5* are mimicked by the canonical allele *unc-23(e25)*[35] and rescued when providing wild-type *unc-23* genomic sequences (Fig. 2d). Thus, proper CEPsh glial morphology requires UNC-23, the *C. elegans* homolog of vertebrate cochaperone BAG2 or Bcl-2-Associated Athanogene 2 that was previously implicated in regulating the architecture of the *C. elegans* muscle[36,37], as well as sensory dendrites and glia in the *C. elegans* peripheral nervous system[38].

To understand the UNC-23/BAG2 function, we examined the timing and nature of the observed glial defects. CEPsh glia in *unc-23* mutants present age-dependent defects. In the 1st larvae stage (L1), they have normal architecture compared to wild-type animals but show mild defects at the 2nd larval stage (L2) which increasingly aggravate in subsequent stages (Fig. 2e–h). To understand how defects arise, we examined if ectopic membranes result from abnormal pattern of cell divisions, or membrane growth or retraction, by quantifying cell numbers, membrane volume, surface, length, and growth rates in developmentally synchronized animals. CEPsh glia membranes in *unc-23* mutants compared to wild-type animals show extended length and decreased volume (Fig. 2c, g). To quantify glial growth rate, we adapted immobilization protocols for long-term imaging and performed time-lapse imaging for individual animals throughout post-embryonic stages ("Methods", Supplementary Movies 4 and 5). Time-lapse quantifications revealed that CEPsh glia membranes grow continuously without retraction throughout larval stages, and with comparable growth rates in wild-type animals and *unc-23* mutants. Specifically, CEPsh glia membrane from L2 stage onwards have the same surface but increased length and decreased volume in *unc-23* mutants compared to wild-type animals (Fig. 2h–j). CEPsh glia present the same number of nuclei and anterior processes in *unc-23* mutants and wild-type L2 animals, before membrane defects are established (Fig. 2k, l and Supplementary Fig. 1), indicating that embryonically

born CEPsh glia do not suffer aberrant divisions or fate transformations during their development. Only later, in L4 *unc-23* mutants already showing glial membrane defects, some defective CEPsh glial cells present nuclei that are misaligned, mispositioned, and sometimes appear fragmented (Fig. 2k, l and Supplementary Fig. 2). Overall, disruption of UNC-23/BAG2 function appears dispensable for the development of CEPsh glial architecture (that is established before L2) but is required for its maintenance (L2 stage onwards).

Certain defects of CEPsh glial cell architecture are observed only in *unc-23* mutant adults. Specifically, *unc-23* mutant adults–but not larvae–present defective anterior membrane tips, that lose attachment with neighboring tissues or show altered morphology, in an age-progressive manner (Supplementary Fig. 1). Also, neuronal synaptic vesicles associated with CEPsh glial cells present different defects in L4 and adult *unc-23* mutants (see below and Supplementary Fig. 3). To investigate if UNC-23/BAG2 function is also required in adulthood for age-progressive maintenance of glial architecture, we subjected wild-type animals of late L4 stage to *unc-23* RNAi and assessed the CEPsh glial cell architecture in different days of adulthood (Supplementary Fig. 3a). Interestingly, *unc-23* RNAi in wild-type adult animals results in defects in the architecture of ramified CEPsh glial membranes, that worsen from day-1 to day-5 adults. These defects are similar to defects in L4 animals resulting from RNAi in L1 animals, but milder than the defects in the genetic mutant suggesting that the wild-type UNC-23 accumulating before RNAi application may be a rather stable protein (Supplementary Fig. 3). These findings are in line with UNC-23/BAG2 acting for age-progressive maintenance of glial cell integrity. Overall, UNC-23/BAG2 is required for maintaining CEPsh glia integrity and its functional loss results in age-progressive disruption of glial architecture with less compact, filamentous, discontinuous membranes, occupying ectopic territories (Supplementary Movies 4 and 5).

### Age-progressive glia disruption affects circuit architecture maintenance and relates to abnormal aging

CEPsh glial cells drive brain neuropil assembly and axon pathfinding during embryonic morphogenesis. Embryonic ablation (by EGL-1/BH3 expression in glia, from embryo ventral closure onwards) causes defects in pathfinding of pioneer and follower axons concurrently growing in the nerve ring[18]. Conversely, the genetic ablation of CEPsh glia at L1 stage onwards was reported to cause no defects in axon pathfinding in the nerve ring[18,39]. Yet, postembryonic CEPsh glia envelop axons and synapses throughout larval and adult stages and affect positioning of neuronal synapses[18,26]. Thus, we assessed if CEPsh glia integrity affects circuit architecture after development. The presynaptic sites of AIY interneurons and their CEPsh-glial contacts show decreased density and are sparser, occupying extended areas in *unc-23* mutants compared to wild-type animals (Fig. 3a–c). We queried whether these defects are specific to AIY synapses or relate to overall changes in circuit architecture. Importantly, increased axon length and cell-body mispositioning is observed in AIY and sensory neurons ensheathed by CEPsh glia in *unc-23* mutants compared to wild-type animals (Fig. 3d–h, n). CEPsh glia were also previously implicated in degeneration of dopaminergic neurons[40]. *unc-23* mutants with age-

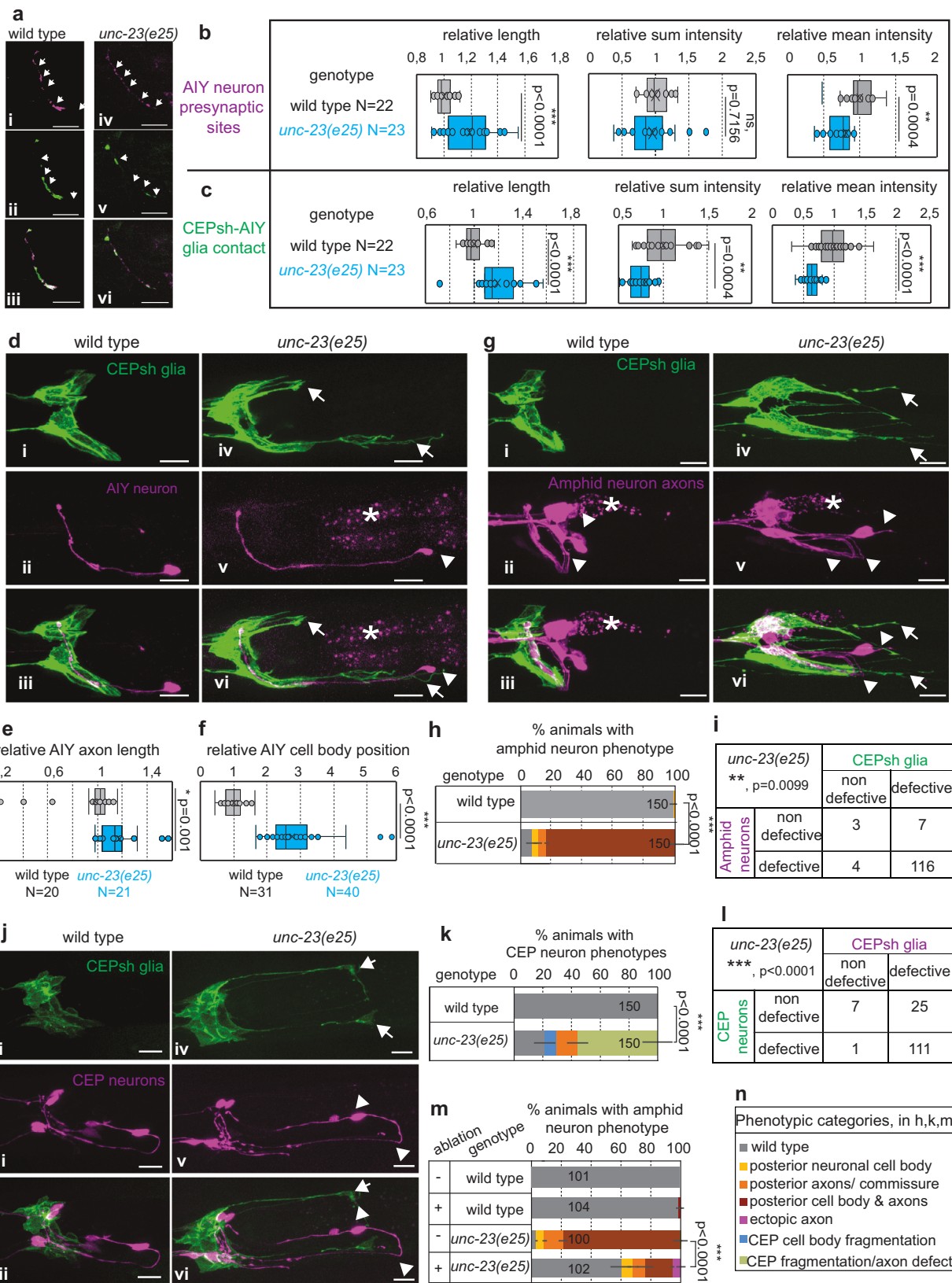

progressive CEPsh glia disruption feature degenerating phenotypes of dopaminergic neurons CEP with cell bodies shrunken or missing (Fig. 3j–l). The observed defects in sensory amphid neurons and CEP neurons correlate significantly with CEPsh glial defects (Fig. 3i, l). To assess the causality of such correlation we ablated postembryonic CEPsh glia, using an established system of caspase expression in

L1 stage of wild-type animals, which does not cause axonal defects (Fig. 3m, "Methods"). This postembryonic ablation of CEPsh glia strongly suppresses the majority of the observed amphid neuron defects in *unc-23* mutants. Since this suppression is partial, we cannot fully exclude that *unc-23* may also affect neurons independently of CEPsh glia. Yet, these ablation experiments suggest that the observed

**Fig. 3 | Synaptic and axon defects, and neurodegeneration are associated with abnormal glia integrity in *unc-23* mutants. a**–**l** Compared to wild-type animals, in *unc-23* mutants the RAB-3-labeled AIY presynaptic sites (magenta, **a**, **b**) and CEPsh-AIY contacts sites (green, **a**, **c**) are sparser, occupying larger areas with decreased density, while AIY (**d**–**f**) and amphid neurons (**g**–**i**) have defective positions and axon length, and CEP neurons show degeneration features (**j**, **k**) (CEPsh glia, green; all other pseudo-colored in magenta, *, gut autofluorescence). Axon defects of amphid neurons (**i**) and CEP neuron defects (**l**) in *unc-23* mutants correlate with glia defects (*p* value = 0.0003 and *p* value < 0.0001 respectively, two-tailed unpaired Chi-square test). **m** Axon defects in *unc-23* mutants are partially suppressed upon postembryonic CEPsh glia ablation (in L1 onwards), that by itself does not cause axon defects. **n** Panel explaining categories of defect types in (**h**–**m**). *n* = 3 independent experiments with total animal numbers noted inside bar graphs

(**h**, **k**, **m**) or noted as N with genotypes in box-whisker plots (**b**, **c**, **e**, **f**). *p* values: ***≤0.0001; **<0.001; *<0.005; ns non-significant. Two-tailed (two-sided) unpaired *t*-test (**b**, **c**, **e**, **f**) and Fisher's test (**h**, **k**, **m**) are used. Arrows, glia defects. Arrowhead, differences between mutant and wild-type in synapses (**a**), or in cell bodies of neurons AIY (**d**), amphid (**g**), CEP (**j**). Scale bar, animal axes, reporters as in Fig. 1. Reporters as listed in "Methods" and Supplementary Tables 3–5 (see Supplementary Fig. 3). Error bars represent mean ± standard deviation. Total animal numbers are noted inside histogram bars or dot-plot graphs. Number of independent experiments for each condition is *n* = 3, unless otherwise noted. Dots, limits, lines, whiskers in box-whisker plots are defined as in Fig. 1. The total number of animals per condition and independent experiments, the primary data, and statistical analysis (including two-way ANOVA statistical analysis) are presented in the Source Data. Scale bar, animal axes, reporters defined as in Fig. 1.

neuronal defects in *unc-23* mutants may largely depend on CEPsh glia disruption.

Finally, previous studies implicate CEPsh glia in animal lifespan[41], while other studies show that synaptic architecture in wild-type animals deteriorates with age progression[35]. Thus, we examined if the *unc-23* mutation that causes CEPsh glial defects, also affects animal lifespan and how is synaptic architecture affected in later adult stages. Five-day adults of *unc-23* mutant with defective CEPsh glia present decreased density, like in L4 animals, as well as decreased overall intensity of presynaptic vesicle markers, which is not observed in L4 animals (Supplementary Fig. 3b, c). The latter is a hallmark of deteriorating neuronal healthspan, as described in previous studies[35]. Thus, CEPsh glial defects are succeeded by additional neuronal defects in *unc-23* mutant adults, compared to larval animals. Moreover, *unc-23* mutants have shorter lifespan compared to wild-type animals, as indicated by their lifespan graph, their median and maximum lifespan (Supplementary Fig. 3d and Supplementary Table 1). Thus, loss of CEPsh glia membrane integrity results in defects in brain axons and concomitant synaptic defects which deteriorate with age, from larvae to adult animals. Altogether, our results highlight that age-progressive integrity of CEPsh glia is key to maintain circuit architecture including axon positioning and synapse density. Disruption of glial cell and circuit integrity is associated with decreased animal lifespan. The mechanistic underpinnings of these effects remain to be further elucidated.

## BAG2-HSP-DNAJ functions protect CEPsh glial integrity in high temperature

We set to examine the mechanism of action of UNC-23 in glial integrity. Its vertebrate homolog, cochaperone BAG2, interacts with stress-induced chaperone Hsp70 and its constitutively-expressed counterpart Hsc70, to regulate proteostasis[42,43]. DNAJB cochaperones also regulate Hsp70/Hsc70, for substrate recognition, binding, and protein quality control[7,42–45]. *C. elegans* DNJ-13/DNAJB1 and UNC-23/BAG2 can act oppositely for binding or release of HSP-1/Hsp70/Hsc70 substrates in other contexts[36]. We assessed if such interactions of HSP-1/Hsp70/Hsc70, DNJ-13/DNAJB1 and UNC-23/BAG2 are involved in CEPsh glial integrity. While *hsp-1* mutants show no defects in CEPsh glia integrity, animals combining *unc-23* mutation with *hsp-1* mutation or knock-down of *hsp-1* or *dnj-13* partially restore this integrity (Fig. 4a, b). Similarly, *hsp-1* and *dnj-13* mutations suppress muscle defects of *unc-23* mutants in vivo, and have opposite effects on HSP-1 binding of ATP and substrate in vitro[36] (see also "Discussion"). Importantly, CEPsh glial defects in *unc-23* mutant are rescued after expression of mouse BAG2 cDNA as presented below (Fig. 4e), suggesting a conserved molecular mechanism. Moreover, the HSF-1 transcription factor can also affect UNC-23 function since loss of HSF-1 partially suppresses the CEPsh glial defects of *unc-23* mutants (Supplementary Fig. 4). Whether these effects are direct/indirect and upstream/downstream of the Bag2/HSP function remains unclear, since triple mutants *unc-23; hsp-1; hsf-1* could not be isolated due to lethality/sterility. Yet, our experiments suggest

that the suppression of glial defects in *unc-23/Bag2* mutants by loss of HSP-1 or DNJ-13 is not simply due to activation of HSF-1 and heat shock response. Specifically, CEPsh glial defects are still suppressed in double mutants *unc-23; hsp-1* upon RNAi knock-down of HSF-1, and in double mutants *unc-23; hsf-1* upon RNAi knock-down of HSP-1 or DNJ-13, so this suppression is not dependent on HSP-1 or DNJ-13 (Supplementary Fig. 4). Importantly, the defects of *unc-23* mutants are suppressed by postembryonic RNAi of *hsp-1, dnj-13*, or *hsf-1* after L1, demonstrating that postembryonic BAG2-HSP acts after development, for maintenance of CEPsh glia architecture (Fig. 4 and Supplementary Fig. 4).

Hsp70/Hsc70 is implicated in temperature responses of cells[45], thus we assessed if thermal stress affects glia integrity. Applying low-level chronic heat shock or acute, strong heat-shock in *unc-23* mutants enhances the disruption of glial cell integrity ("Methods", Fig. 4c). We also assessed if the effect of BAG2-Hsp70 in glial cell integrity is affected by nutritional states. Conversely to high temperature, starvation partially suppresses glial integrity phenotypes in *unc-23* mutants (Fig. 4d). Importantly, these temperature and nutritional changes do not affect the integrity of CEPsh glial architecture in wild-type animals (Fig. 4c, d). Thus, temperature and caloric changes affect the roles of UNC-23/BAG2 in CEPsh glial cell and circuit maintenance, through mechanisms that we further study below.

## Epithelial UNC-23 protects CEPsh glia integrity separately from allostery

To understand the site of action of UNC-23/BAG2 for glia integrity, we examined the presence of *unc-23* transcripts in single-cell transcriptomic data (scRNAseq)[46] and tested tissue-specific roles using RNA-interference (RNAi) and rescue experiments. *unc-23* transcripts in L2 larvae are mainly detected in epithelia, muscle, and glial cells including CEPsh. Bacteria-fed (systemic) RNAi against *unc-23* in wild-type animals mimics *unc-23* mutant phenotypes (Fig. 4e). This suggests that UNC-23 does not act in neurons or CEPsh glia, that are insensitive to bacteria-fed RNAi in wild-type animals. Moreover, vector-induced knock-down of *unc-23* in CEPsh glia does not induce glial defects (Fig. 4e). Specific expression of *unc-23* cDNA in epithelia, or the epithelial-expression of mouse BAG2, but not *unc-23* expression in CEPsh glia or muscle, results in highly significant rescue of the CEPsh glia defects in *unc-23* mutants (Fig. 4e). Epithelial expression of *unc-23* cDNA also rescues the neuronal defects observed in amphid sensory neurons of *unc-23* mutants (Fig. 4f). Thus, age-dependent integrity of CEPsh glia and circuit components requires conserved functions of UNC-23 as a BAG2 cochaperone in epithelia, that neighbor CEPsh glia. The HSP-1/Hsp70/Hsc70, DNJ-13/DNAJB1 proteins acting with UNC-23/BAG2 also function non-cell-autonomously since their knock-down suppresses *unc-23* mutant defects, upon RNAi in animals with CEPsh glia insensitive in RNAi effects (Fig. 4b, "Methods").

Since UNC-23/BAG-2 acts in epithelia, we examined whether it is required for epithelial integrity or polarity. We generated and

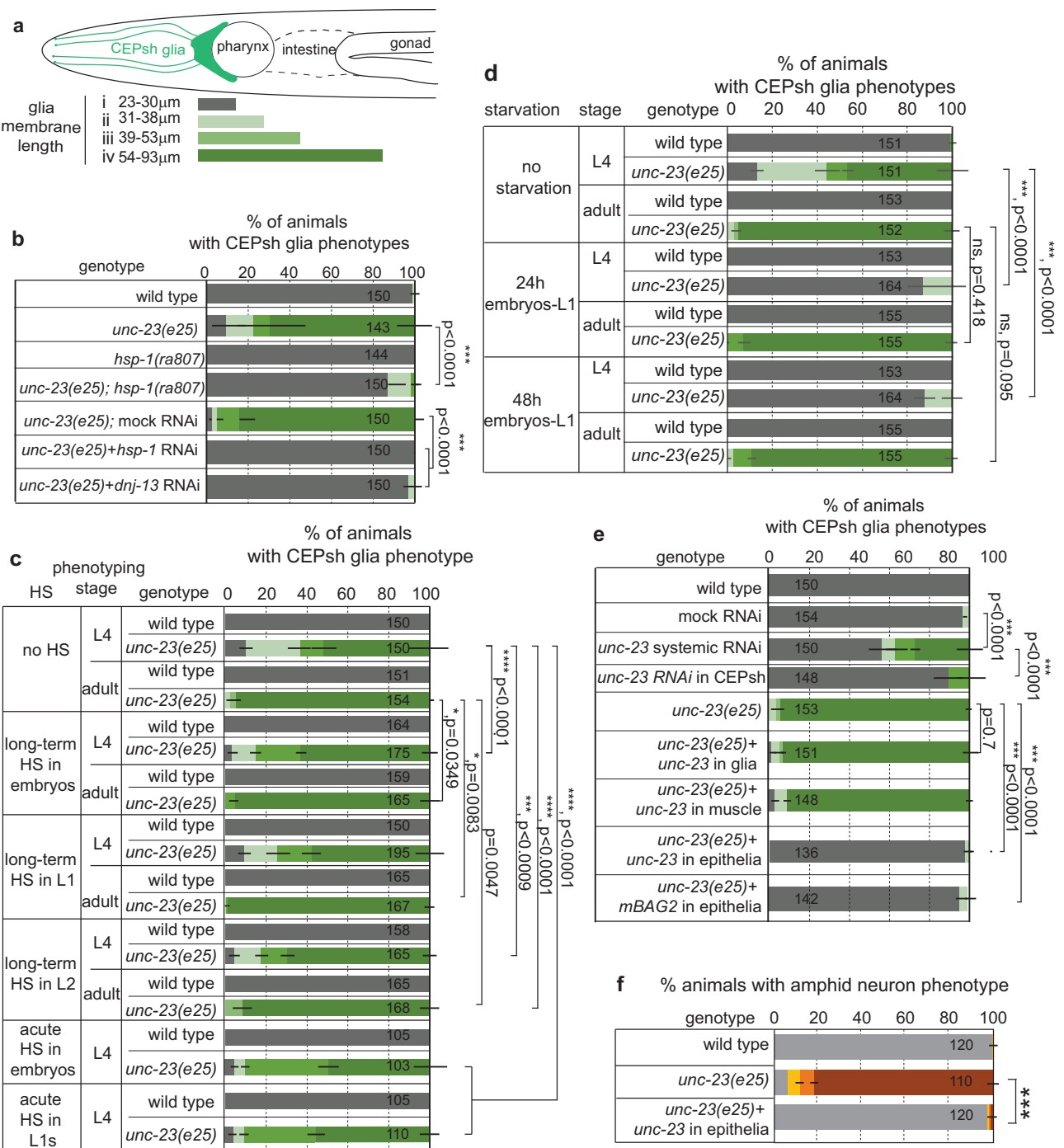

**Fig. 4 | Epithelial UNC-23 acts with HSP-1 and DNJ-13 to regulate glial integrity in relation to temperature and nutrition. a**, **b** CEPsh glia defects (schematics, **a**) in *unc-23* mutants are suppressed by loss of HSP-1 or DNJ-13 using mutants or post-embryonic RNAi (**b**). **c**, **d** The environment's temperature or caloric changes can affect glial cell integrity in *unc-23* mutants. Long-term or short and acute temperature increase (in embryos, L1, L2 animals), accelerates the glial cell defects of *unc-23* mutants in phenotyped L4 animals (**c**) while to the contrary, starvation can partially suppress and delay these defects (**d**). **e**, **f** CEPsh glia defects in *unc-23* mutants are mimicked by post-embryonic RNAi of *unc-23* (**c**), rescued by UNC-23 expression from epithelia, but not from glia or muscle, and rescued by mouse BAG2 (**c**). Epithelial cell

rescue of *unc-23* also rescues the neuronal defects observed in amphid sensory neurons (**d**). Color-coding of neuron phenotypes as presented in Fig. 3n. *n* = 3 independent experiments with total animal numbers noted inside bars in bar graphs. Two-sided chi-square test is used. HS heat shock. Error bars, *p* value, ns defined as in Fig. 2. Error bars represent mean ± standard deviation. Total animal numbers are noted inside histogram bars or dot-plot graphs. Number of independent experiments for each condition is *n* = 3, unless otherwise noted. Dots, limits, lines, whiskers in box-whisker plots are defined as in Fig. 1. The total number of animals per condition and number of independent experiments, the primary data, and statistical analysis (including two-way ANOVA statistical analysis) are presented in the Source Data.

visualized epithelial-expressed membrane-bound fluorophores and the apical-surface SAX-7-derived marker ApiGreen (previously established by others[25]). Importantly, we cannot detect significant differences in the continuity of epithelial membrane and the intensity of

apical-surface marker neighboring the CEPsh glia between *unc-23* mutants and wild-type animals (Supplementary Fig. 5a, b). The integrity of epithelial cell junctions neighboring the CEPsh glia appears similar in *unc-23* mutants and wild-type animals, as assessed by

quantifying an enrichment of the apical-junction component DLG-1 (Discs large MAGUK scaffold protein 1) neighboring the CEPsh glia (Supplementary Fig. 5c–f). Previous immunostaining studies in fixed adult animals proposed defects in epithelial hemidesmosomes in *unc-23* mutants[37]. While we cannot exclude *unc-23* mutant defects in other epithelial properties/ domains far from CEPsh glial membrane sheaths or in animal stages not assessed here, we detect no significant changes in epithelial polarity and cell junctions' integrity neighboring the CEPsh glia that may account for impaired integrity of CEPsh glial, axon-associated, membrane sheaths in *unc-23* mutants.

*C. elegans* epidermis has previously suggested to regulate CEPsh glia through the transporter CIMA-1/ SLC17A5 and EGL-15/FGF receptor, that were described as allostery cues[26]. To investigate if UNC-23/BAG2 acts in the same pathway, we examined if changing the levels of these allostery cues affects the glial defects in *unc-23* mutants. Neither overexpression nor genetic or RNAi-induced knock-down of CIMA-1/SCL17 or EGL-15/FGFR, suppresses the disrupted glial integrity of *unc-23* mutants (Supplementary Fig. 6 and Supplementary Table 2). Therefore, UNC-23/BAG2 can affect CEPsh glial integrity also through unknown pathways of glia-epidermis communication, that appear to be independent of epithelial integrity and known allostery cues. Epithelial-derived UNC-23/Bag2 can also affect morphology of the *C. elegans* peripheral nervous system (the sensory dendrites and peripheral glial cells AMsh) yet through seemingly different mechanisms[38]. The progressive defects of *unc-23* mutants in CNS CEPsh glia (and axons) appear before adulthood, and appear to be not EGL-15/FGFR-dependent, while adult-specific and EGL-15/FGFR-dependent defects are described in the peripheral nervous system of *unc-23* mutant[38].

## Ectopic CEPsh glial sheath upon *unc-23* loss depends on ECM level accumulation

The BAG2-Hsp70/Hsc70 complex is implicated in proteostasis, protein folding, and disaggregation[43,45]. Accordingly, the epithelial UNC-23/BAG2, may safeguard age-dependent glia integrity by regulating proteostasis required for epithelia-glia communication. We reasoned that if abnormal proteostasis of epithelial-secreted factors causes glial defects in *unc-23* mutants, modifying their levels may reverse these defects. To examine this hypothesis, we defined a predicted epithelial secretome, by the presence of signal peptide and epidermis-enriched transcripts in transcriptomes of L2 animals[46]. We focused on transcripts of hyp7 epithelial cells, neighboring the CEPsh glia[28]. We performed an RNAi screen to knock-down epithelial-expressed ECM and secreted factors with epithelia-enriched transcripts ("Methods", Supplementary Table 2). We found that reducing the levels of ECM components UNC-52/Perlecan, EMB-9/COL4A5, and laminin LAM-1/LAMB1-2, partially suppresses the CEPsh glial defects of *unc-23* mutants (Fig. 5a). *unc-52* genetic mutation also suppresses CEPsh glial phenotypes in *unc-23* mutants. This suggests that ECM components act downstream of UNC-23 functions and upstream of the CEPsh glial phenotypes. To study the ECM effects in relation to glia integrity in *unc-23* mutants, we examined UNC-52/Perlecan and EMB-9/ COL4A5 using CRISPR knock-in fluorescent tags[47]. Compared to their continuous matrix in wild-type animals, the Perlecan and COL4A5 present abnormal localization in *unc-23* mutants, are depleted from areas neighboring normal CEPsh glia positions, and accumulate neighboring the ectopic glial territories (Fig. 5b–d and Supplementary Fig. 7). Importantly, Perlecan matrix abnormalities are significantly correlated with glial defects in *unc-23* mutant L4 animals but these ECM defects precede CEPsh glial defects in L2 mutant animals (Fig. 5e, f). Importantly, upon ablation of CEPsh glial cells at L1 stage onwards, the defects of Perlecan/UNC-52 localization in *unc-23* mutants are not suppressed, they remain similar to the unc-23 mutants (Fig. 5g). This is contrary to the suppression of neuronal defects, upon CEPsh glia ablation in L1 *unc-23* mutants. Moreover, *hsp-1* knock-down

(suppressing CEPsh glial cells defects) also suppresses the ECM defects of Perlecan/UNC-52 and Collagen/EMB-9 localization in *unc-23* mutants (Supplementary Fig. 8a–c). Based on all these observations combined, the ECM components appear to act downstream of UNC-23/HSP proteostasis and upstream of CEPsh glia for maintenance of their glial integrity.

Since temperature and starvation changes affect CEPsh glial defects in *unc-23* mutants, we assessed if they also affect ECM responses in relation to HSP proteostasis (in *unc-23* mutants), and if they affect glia and ECM also in wild-type animals. Perlecan localization in wild type animals of L2 or L4 stage does not present defects subject to temperature increase or starvation, compared to wild-type animals in normal conditions (Supplementary Fig. 8d). In line with this observation, temperature increase or starvation do not cause CEPsh glial cell defects in wild type animals as shown above (Fig. 4c). Conversely, the mild defects of Perlecan localization observed in L2 *unc-23* mutants are enhanced upon temperature increase and suppressed upon starvation, compared to untreated mutants (Supplementary Fig. 8d, e). Importantly, no altered locomotion of L2 or L4 animals (as quantified by body bends per minute) is detected in starved versus well-fed *unc-23* mutants (Supplementary Fig. 8f) suggesting that starvation affects UNC-23/BAG2 functions in glia integrity through mechanisms other than altered locomotion. Thus, the progressive ECM defects of *unc-23* mutants, are dependent in HSP-1/Hsc70 function, accelerated by temperature increases and delayed by caloric restriction, while such changes do not affect the robust ECM localization in wild-type animals. Altogether, ECM components act upstream of CEPsh glial cell integrity, accumulate abnormally in *unc-23* mutants because of abnormal HSP function, are affected by temperature and nutritional changes, they correlate with and are followed by defective CEPsh glia, and altering their level suppresses CEPsh glial defects in *unc-23* mutants.

## Cell junctions connect CEPsh glia, epithelia and ECM and affect glia integrity

Since ectopic CEPsh glia correlate with ECM accumulations, we reasoned that receptors of ECM may contribute to this association and the glial disruption. To identify putative receptors of Perlecan, COL4A5, and Laminin, we queried the literature for interacting factors, presenting transcripts in CEPsh glia[46]. Our queries highlighted proteins LET-805, DLG-1 and integrins, partaking in cell attachments. Mutating the integrin INA-1, that presents CEPsh glia transcripts does not suppress CEPsh glia defects in *unc-23* mutants (Fig. 5a). The fibronectin-domain transmembrane receptor LET-805, together with Matrilins MUA-3, MUP-4, compose hemidesmosomes while the scaffolding protein DLG-1 and its binding partners AJM-1 and LET-413 compose adherens junctions, that mediate cell-ECM and cell-cell interactions respectively[10]. These components are expressed in postembryonic CEPsh glia based on transcriptomics (Supplementary Table 3). We examined if cell-junction components present specific localizations in CEPsh glia and may affect glial integrity. A fluorescently-tagged tool visualizing the DLG-1 binding partner AJM-1, expressed specifically in CEPsh glia, presents enrichments in the CEPsh glia membrane sheath in wild-type and *unc-23* mutant animals (Fig. 6a). This suggests that adherens junctions form in CEPsh glia membranes of normal and *unc-23* mutant disrupted architecture (Fig. 6a). The CEPsh glia sheath also closely associates with DLG-1-containing epithelial adherens junctions, in wild-type and *unc-23* mutant animals (Fig. 6b, c). Glial-juxtaposed DLG-1 junctions associate closely with Perlecan in wild-type animals, and this association is maintained in *unc-23* mutants (Fig. 6d, e). Similarly to the CEPsh glia, epithelial DLG-1 junctions assume ectopic posterior localization in *unc-23* mutants, close to Perlecan accumulations (Fig. 6d, e).

We reasoned that if junctional components directly or indirectly connect the CEPsh glia to their neighbors (epithelia, ECM), knocking them down in *unc-23* mutants may release the CEPsh glial membranes

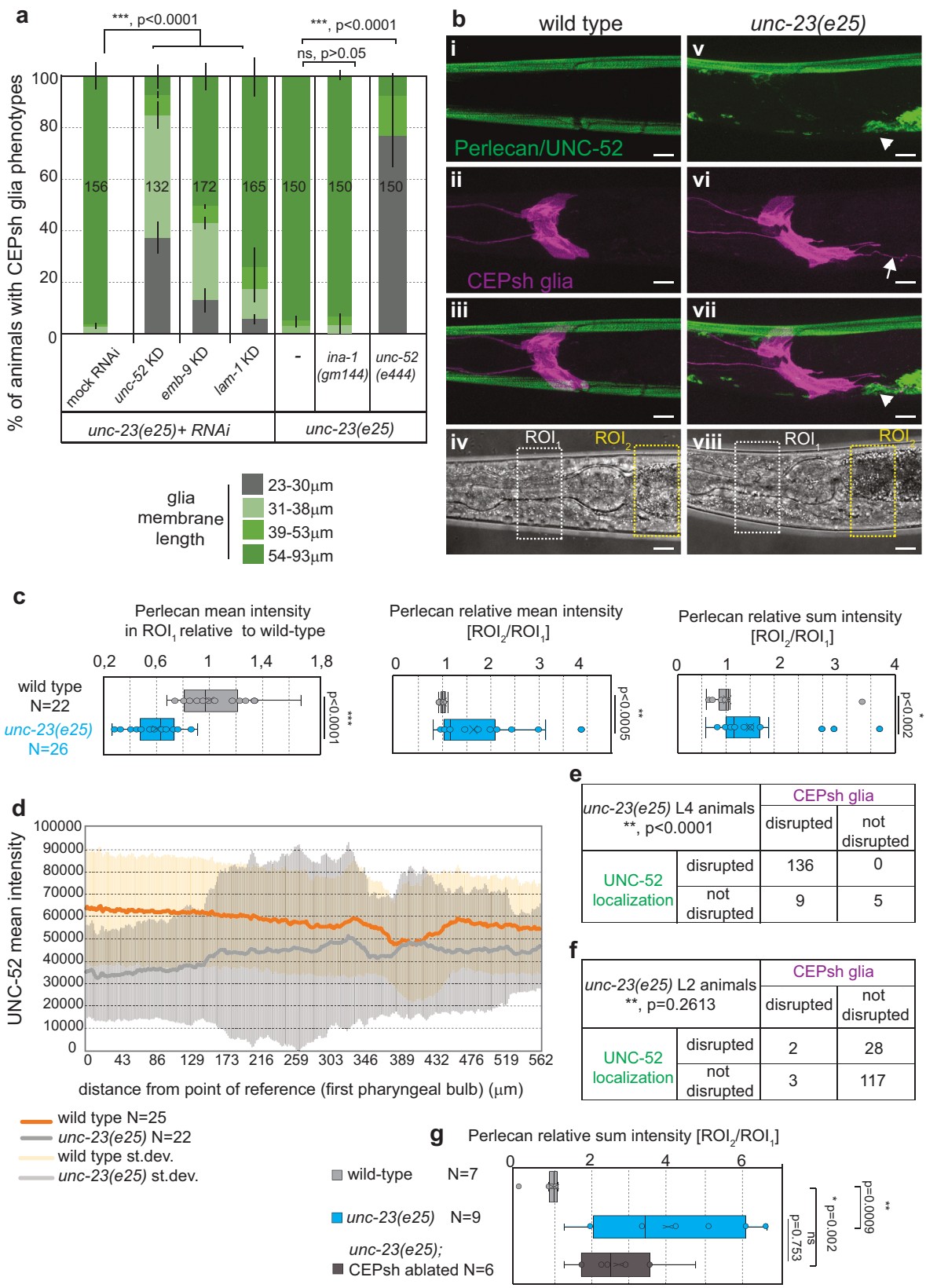

**c** Perlecan mean intensity in ROI$_1$ relative to wild-type — Perlecan relative mean intensity [ROI$_2$/ROI$_1$] — Perlecan relative sum intensity [ROI$_2$/ROI$_1$]

**d** UNC-52 mean intensity vs distance from point of reference (first pharyngeal bulb) (μm)

**e**

| *unc-23(e25)* L4 animals \*\*, p<0.0001 | | CEPsh glia | |
|---|---|---|---|
| | | disrupted | not disrupted |
| UNC-52 localization | disrupted | 136 | 0 |
| | not disrupted | 9 | 5 |

**f**

| *unc-23(e25)* L2 animals \*\*, p=0.2613 | | CEPsh glia | |
|---|---|---|---|
| | | disrupted | not disrupted |
| UNC-52 localization | disrupted | 2 | 28 |
| | not disrupted | 3 | 117 |

**g** Perlecan relative sum intensity [ROI$_2$/ROI$_1$]

from these attachments, and suppress their presence in ectopic territories. To test this hypothesis, we knocked-down cell-junction components. Knocking-down DLG-1 specifically in postembryonic CEPsh glia, by vector-mediated RNAi, suppressed the ectopic CEPsh glia territories in *unc-23* mutants (Fig. 6f). Similarly, mutant CEPsh glia defects are partially suppressed upon bacteria-mediated knock-down of DLG-1

AJM-1, LET-413, LET-805, MUA-3 in the nervous system, using mutations that enhance RNAi-sensitivity in the nervous system, including in CEPsh glia (Fig. 6f and Supplementary Table 2, "Methods"). Transcripts of these components are largely absent from scRNAseq of postembryonic neurons[46], thus this sensitized knock-down largely affects their glial transcripts. Conversely, CEPsh glia defects are not

**Fig. 5 | ECM is disrupted in *unc-23* mutants and altering its composition suppresses CEPsh glial defects. a** Knocking-down epithelial-expressed ECM components Perlecan, Collagen and Laminin by RNAi (KD) or the *unc-52(e444)* genetic mutation of Perlecan partially suppresses CEPsh glia defects in *unc-23* mutants. Two-way Fisher's test is used. Arrows, glia defects. Arrowhead, Perlecan defects. **b, c** Perlecan/UNC-52 (green) has disrupted localization and accumulates posteriorly of the pharynx in *unc-23* mutants, compared to wild-type L4 animals. In *unc-23* mutants, mean intensity of UNC-52::KI is significantly decreased anterior to the 2nd pharyngeal bulb, and the ratio of mean and sum intensity between posterior and anterior regions is increased, compared to wild-type animals. $ROI_1$, $ROI_2$ in (**c**) are as defined in (**b**). **d** Perlecan mean intensity at distance posterior to 1st pharyngeal bulb, represented by mean (gray/orange) ± standard deviation (light gray/yellow)

(**e, f**). Perlecan defects correlate to defects of CEPsh glia in L4 (**e**) and precedes these defects in L2 animals (**f**), $n \geq 100$ animals ($p$ value < 0.0001 in **e, f**, two-tailed unpaired Chi-square test). **g** CEPsh glia ablation (at L1 stage onwards) does not significantly suppress the defects of Perlecan UNC-52 in *unc-23* mutants. Error bars represent mean ± standard deviation. Total animal numbers are noted inside histogram bars or dot-plot graphs. Number of independent experiments for each condition is $n = 3$, unless otherwise noted. Dots, limits, lines, whiskers in box-whisker plots are defined as in Fig. 1. The total number of animals per condition and number of independent experiments, the primary data, and statistical analysis (including two-way ANOVA statistical analysis) are presented in the Source Data. RNAi results and statistics is also provided in Supplementary Table 2. Scale bar, animal axes, reporters defined as in Fig. 1.

suppressed by knocking-down epithelial junctional components, in *unc-23* mutants with RNAi-insensitive nervous system (Fig. 6f, "Methods"). Such RNAi in non-sensitized backgrounds can affect the transcripts in non-nervous system cells as well as in head glial cells other than the CEPsh (amphid sheath, AMsh and other head anterior sheath and socket glia) (Supplementary Table 2). Thus, the differential RNAi effect in non-sensitized and sensitized backgrounds can highlight effects in CEPsh glia, but not in epithelia, muscle or other head glial cells. Overall, this suggests that glia junctional components affect CEPsh glial integrity in *unc-23* mutants. Importantly, knocking-down these components in *unc-23* mutants does not suppress Perlecan and muscle integrity defects (Fig. 6g), thus it suppresses glia mislocalization not by recovering a proper localization of the ECM matrix but by disconnecting it from the glia. In summary, CEPsh glia axon-enveloping membranes juxtapose epithelial adheres junctions closely associating with ECM and harbor localized glial junction components, while interrupting these components releases the association of glia to abnormal ECM accumulations, safeguarding integrity of glial architecture away from ectopic territories.

## The mechanical properties of the animal and its environment affects glia integrity

To understand how glia and ECM deformations in *unc-23* mutants are affected by the environment, we altered the mechanical properties of the animals' environment. Defective muscle integrity in *unc-23* mutants is suppressed when growing animals in liquid, but the underlying mechanism is unclear[37]. CEPsh glia integrity is partially recovered upon growth of *unc-23* mutants in liquid. Perlecan matrix is also recovered, and its recovery is strongly correlated with recovery of glia architecture (Fig. 7a, b). Conversely, concurrent recovery of muscle integrity does not correlate with recovered glial integrity, suggesting that the defects in CEPsh glial integrity are not caused directly by the muscle defects in *unc-23* mutants (Fig. 7c). Growth in liquid or on solid media presents differences in locomotory patterns as well as in exerted forces. Specifically, animals adopt thrashing when swimming and crawling on solid media[48] and exchange 1000–10,000 larger forces on solid compared to liquid media[49,50]. To distinguish between effects of locomotion and forces, we induced paralysis on solid media to decrease the forces without causing thrashing. Such paralysis reverted the CEPsh glial defects of *unc-23* mutants (Fig. 7d). Thus, glial integrity upon UNC-23/ BAG2 functional loss may be sensitive to larger exerted forces.

In conjunction to exerted forces, tissue deformation depends on its material properties. All biological matter exhibits viscoelastic properties, displaying viscous and elastic characteristics when undergoing deformation. We measured in vivo biomechanical properties to query if they are different in *unc-23* mutants compared to wild-type animals. To this aim, we performed Brillouin microscopy, a non-destructive, label-free and contact-free method of optical elastography[24]. Therefore, Brillouin microscopy provides unprecedented capability to measure tissue mechanical properties in vivo. In particular, Brillouin microscopy measures both the so-called high-

frequency elasticity as well as viscosity of biological samples, through the frequency shift and spectral linewidth of the Brillouin scattered photons interacting with the sample. While we note that for quantitative measurements of the longitudinal modulus the ratio between the refractive index ($n$) squared and density ($\rho$) of the probed tissue need to be known, prior work has shown that this ratio $\rho/n2$ does not vary significantly in biological materials[51–53], therefore relative changes in the spectral shift and linewidth are indicative of mechanical changes. Utilizing a confocal Brillouin microscope[54], we probed anesthetized *unc-23* mutant and wild-type animals, on L2 stage preceding the establishment of glial defects. Areas neighboring CEPsh glia present decreased tissue viscosity and elasticity in *unc-23* mutants compared to wild-type animals (Fig. 7e, f and Supplementary Fig. 9a–c), as measured by the Brillouin spectral linewidth and shift, respectively[55]. To further ascertain that the observed changes in Brillouin shift and linewidth do in fact have a mechanical origin, we performed additional measurements and analysis ("Methods", Fig. 7 and Supplementary Fig. 9). Specifically, we performed refractive index measurements and we computed the so-called Brillouin loss tangent (BLT), defined as $\tan(\varphi) = \Gamma/\nu$, which by definition does not depend on the sample refractive index and density[53,56]. We found that the average refractive index did not differ significantly between the mutant and wild-type animals, while the BLT was significantly lower in mutants compared to wild-type animals (Supplementary Fig. 9). This further confirms that the observed Brillouin spectral changes (Fig. 7e, f and Supplementary Fig. 9a–d) are due to changes in mechanical properties, and not due to changes in either refractive index and/or density. Therefore, mutant animals lacking functional BAG2/HSP-cochaperone, present already in early stages a reduced mean viscosity and elasticity in tissues neighboring the CEPsh glial cells. These altered material properties and animal biomechanics precede and underly the later, age-progressive disruption of CEPsh glia architecture, as measured by optical microscopy. Our combined results show that animals with reduced viscosity and elasticity suffer disruption of ECM and tissue integrity that can be partially rescued by subjecting them to a substrate of lower viscosity or reduced forces in their environment.

Since forces between tissues and their environment are transmitted through the animal's exoskeleton, we reasoned that by modifying it we may regulate tissue integrity. To modify the cuticle, primarily composed by collagens, we used genetic manipulations of RNAi or chemical drugs BPY (collagen biosynthesis inhibitor), GM6001 (matrix metalloprotease inhibitor/collagen degradation inhibitor), and BAPN (collagen crosslinking inhibitor) ("Methods"). The disrupted CEPsh glia architecture in *unc-23* mutants is aggravated when chemically inhibiting collagen degradation or crosslinking (Fig. 7g). Conversely, the disrupted CEPsh glia architecture in *unc-23* mutants is partially suppressed when inhibiting collagen biosynthesis chemically (using the above drugs), by knocking-down collagen-modifying enzymes ADT-2/ADAMTS, Peroxidasin-like MLT-7/PXDNL, pro-lysyl oxidase protease NAS-37 or specific collagens including DPY-3, COL-41, COL-43, COL-48, COL-109, COL-115, COL-145 (Fig. 7g, h and Supplementary Table 2), many of which have predicted orthologues in

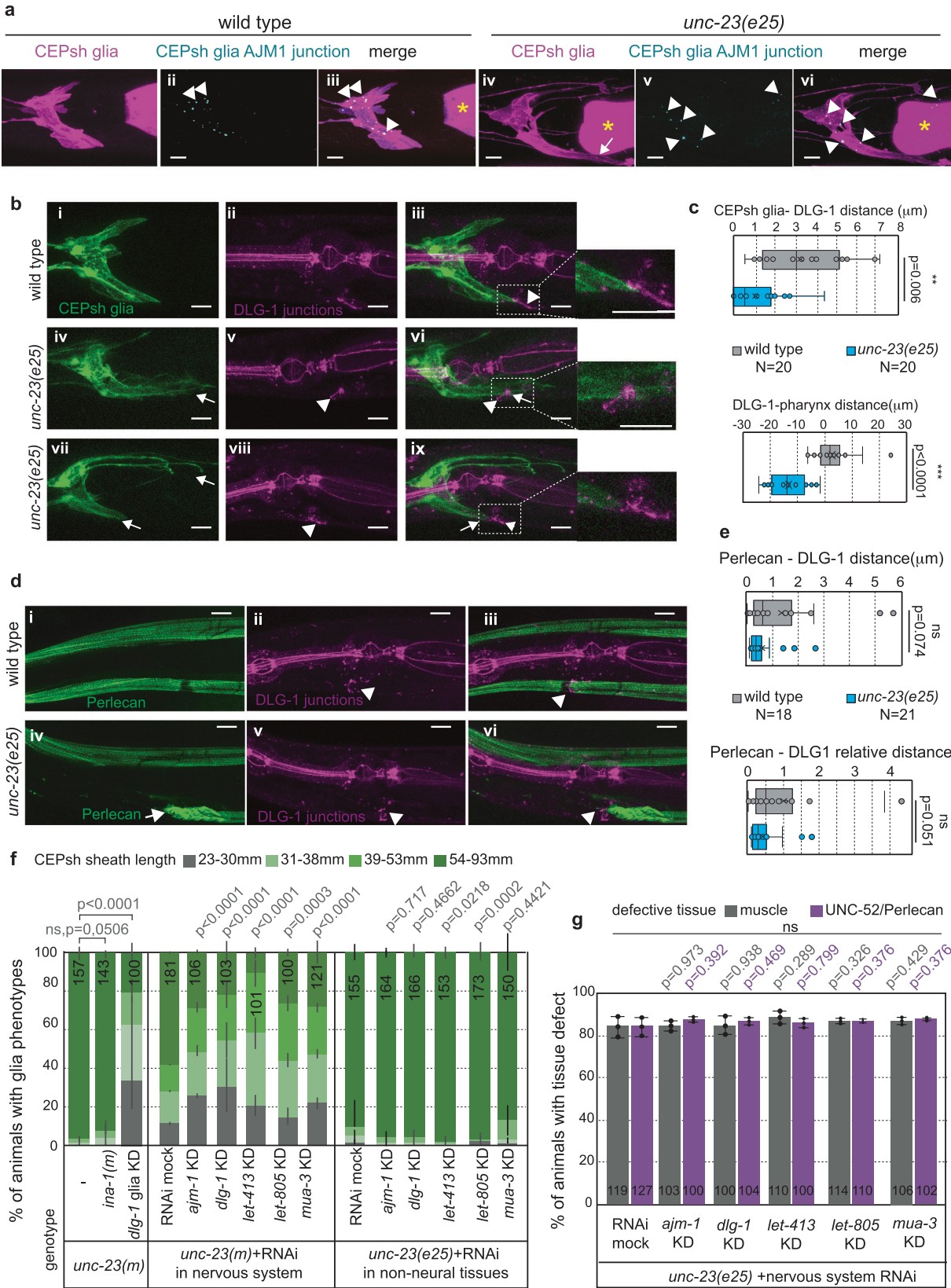

mammals[57]. In conclusion, altering the material properties of the animal's exoskeleton safeguards the age-dependent integrity of ECM and glia architecture.

Altogether, animals lacking epithelial Hsp70/Hsc70-cochaperone BAG2, suffer age-progressive disruption of ECM matrix, decreased tissue viscosity, and are hypersensitive to temperature, the environment's material properties and tension forces. Their CEPsh glia follow mislocalized ECM matrix and epithelial junctions, through glial junctional-components, and experience abnormal stretching to occupy ectopic territories. Disordered glial integrity results in disruption of brain architecture after its establishment, with aberrant localization of neurons, axons, synapses and is associated with

**Fig. 6 | CEPsh glia appose to epithelial junctions and ECM, and disrupting glial junctions affects glia integrity in *unc-23* mutants. a** Adherens-junction marker AJM-1 (cyan) expressed in CEPsh glia (magenta) presents characteristic punctate localization in wild-type animals, largely maintained in *unc-23* mutants. **b, c** CEPsh glia appose epithelial DLG-1 adherens-junction marker in wild-type animals and maintain their apposition while moving posteriorly in *unc-23* mutants. **d, e** Epithelial DLG-1 closely associates with Perlecan matrix in *unc-23* mutants and wild-type animals. **f, g** Vector-induced knock-down of DLG-1 specifically in CEPsh glia ("glia KD") and bacteria-mediated RNAi knocking down of junctional components in the nervous system including in CEPsh glia, but not the KD in non-nervous-system tissues alone, partially suppresses CEPsh glia defects in *unc-23* mutants (**f**). To the contrary, the Perlecan and muscle defects (**g**) are not suppressed during this knock-down. Arrow, glia defects (**a**) or Perlecan/UNC-52 mis-localization (**e**). Arrowhead,

DLG-1 enrichment (**a–e**). Yellow asterisk, pharyngeal bulb (**c–e**). $n = 3$ independent experiments (**a**) with total animal numbers noted inside bar graphs (**f, g**) or noted as *N* with genotypes in box-whisker plots (**c, e**). unpaired *t*-test (**c–e**). *Two-way Fisher's exact test (**f, g**). KD knock-down. Error bars represent mean ± standard deviation. Total animal numbers are noted inside histogram bars or dot-plot graphs. Number of independent experiments for each condition is $n = 3$, unless otherwise noted. Dots, limits, lines, whiskers in box-whisker plots are defined as in Fig. 1. The total number of animals per condition and number of independent experiments, the primary data, and statistical analysis (including two-way ANOVA statistical analysis) are presented in the Source Data. Scale bars, animal axes, reporters, *p* value, ns, and dots, limits, lines, whiskers, error bars in box-whisker and bar graphs are defined as in Fig. 2.

abnormal aging. Modifying glial junctions, the neighboring ECM, or the material properties of the exoskeleton or the environment enhances the robustness of glia integrity against loss of BAG2.

## Discussion

Overall, this work uncovers a coordinated interplay between stress-dependent actions of the Hsp70-chaperone system, ECM architecture, cell-junctions and biomechanics, that regulates age-dependent integrity of astrocyte-like CEPsh glia and subsequent circuit architecture. Dedicated mechanisms are employed for postembryonic maintenance of glia, distinct to those acting for their development.

Upon functional loss of Hsp70-co-chaperone BAG2, the CEPsh glia suffer disrupted architecture with features of lost integrity, age-progressive shearing and possible nuclei fragmentation (Fig. 2 and Supplementary Fig. 2), which is reminiscent of astrocyte degeneration[58]. CEPsh disrupted architecture affects axon-associated glial membranes and is later associated with defects in dendrite-associated processes (Figs. 2 and 3 and Supplementary Fig. 1). It also results in decreased neuron-glia contacts (Fig. 3), also observed in mutants of allostery cues[32], as well as subsequent decreased density of associated presynaptic sites. We demonstrate that these neuronal defects are neither compartment-specific nor neuron-specific, but relate to extended circuit defects. Disrupted CEPsh glial integrity causes neuron and axon mispositioning in diverse neurons, and dopaminergic neurodegeneration (Fig. 3). Postembryonic killing of CEPsh glia, that does not cause defective axon morphologies in wild type animals, can alleviate the neuronal circuit defects in *unc-23* mutants that present ectopic glial membranes (Fig. 3)[29]. Thus, while absence of postembryonic CEPsh glia may not compromise the circuit structure that requires embryonic CEPsh glia to assemble, the age-progressive ectopic positioning of CEPsh glia disrupts circuit maintenance (Fig. 3 and Supplementary Fig. 3). Mutants with functional disruption of UNC-23/BAG2 and deteriorating CEPsh glial integrity present misplaced synaptic vesicle densities in larvae accompanied in later adulthood by decrease in overall content of synaptic vesicle densities, previously described as a sign of deteriorating neuronal healthspan[35]. Therefore, maintenance of proper CEPsh astroglia architecture is key for circuit maintenance. Similarly, ectopic activity of astrocytes, responding to injuries or age progression, contributes to neuropathology[13]. In mammals, gliodegeneration can precede aging and neurodegeneration and contributes to cognitive impairment[7,58]. Uncovering mechanisms of age-dependent astroglial integrity can shed light on understudied aspects of neuropathology.

The UNC-23/BAG2 acts for age-progressive disruption of CEPsh glial cell architecture while it appears largely dispensable for glial cell and circuit development, established by the L2 larval stage (Introduction, Figs. 2 and 3). This age-progressive disruption of CEPsh glia integrity in *unc-23* mutants may arise from a compromised balance of the BAG2-Hsp70/Hsc70 chaperone system (Fig. 4). BAG proteins interacting with Hsp70/Hsc70 promote substrate release, to inhibit or enhance chaperone activity depending on the substrate and cellular

context[42]. BAG proteins support client and nucleotide release while DNJ-13 supports the opposite Hsc70 conformation of a nADP client-bound state, in vitro[36,37]. Upon loss of UNC-23 function, DNJ-13 may block nucleotide and client release, shifting the Hsc70 cycle out of balance. Indeed, *unc-23/BAG2* disruption causes defects in CEPsh glia integrity, that are suppressed by additional decrease of HSP-1/ Hsc70 or DNJ-13/DNAJ function.

The BAG2-HSP-system regulates CEPsh glia integrity from epithelia, by causing mislocalized ECM proteins and epithelial junctions that are followed ectopically by CEPsh glia (Figs. 4 and 5). The abnormal localization of ECM matrix components appears to be dependent on the imbalance of HSP70 upon disruption of UNC-23/Bag2 function, as it is alleviated by HSP-1 decrease in *unc-23* mutants (Supplementary Fig. 8). Vertebrate BAG2 and Hsp70/Hsc70 homologs are implicated in neurodegeneration and aging[4,8], while in other contexts, they function in relation to temperature insults and nutrient changes[3]. Their effects on cell integrity in response to environmental stress are less clear in the nervous system. Our findings establish that upon BAG2-HSP70 disruption, ECM and glia integrity is sensitive to heat and mechanical strain, while this is counteracted by caloric restriction (Fig. 4 and Supplementary Fig. 8). Overall, our findings suggest that UNC-23/ BAG2, through conserved functions and together with the HSP-1/ Hsp70/Hsc70, may integrate tissue responses to temperature, caloric restriction and biomechanics to safeguard the robust integrity of ECM, glial cell and circuit components upon age progression. The exact mechanistic underpinnings of this integration remain to be further elucidated.

The BAG2-HSP complex acts for glia integrity, upstream of ECM and cell junctions. Upon BAG2 impairment, accumulations of basement membrane components (Fig. 5) and mispositioning of epithelial junctions precede the disrupted glial integrity and are followed ectopically by glia membranes (Fig. 6). How CEPsh glia may physically associate with epithelia and the ECM and how this affects age-progressive integrity was previously largely unclear. Our work highlights that the levels of Perlecan, Collagen IV, Laminin and glial cell-junction components are critical for association of glia to ECM and epithelial junctions (Fig. 6). Manipulating these factors can discharge this association and safeguard glia architecture despite their neighbors' mis-localization. Cytoplasmic cues supporting the association of CEPsh glia to ECM/epithelia downstream of glial junctions, may include factors such as spectrins and intermediate filaments[21].

The intimate association of CEPsh glia with epithelia and basement membrane is reminiscent of interactions of astrocytes with endothelial cell junctions, or the Perlecan, Collagen IV and Laminin of the blood-brain-barrier (BBB)[59]. These interactions are key for BBB integrity[60], and understanding which glial components sustain them is crucial. Our work implicates cell-junction components in an intimate glia-epithelia-ECM association, opening inroads to consider glial roles of their vertebrate homologs. Besides the suggested presence of desmosomes in astrocytes, in astrogliosis or Alzheimer's, astrocyte functions of adherens junctions and hemidesmosomes remain unclear.

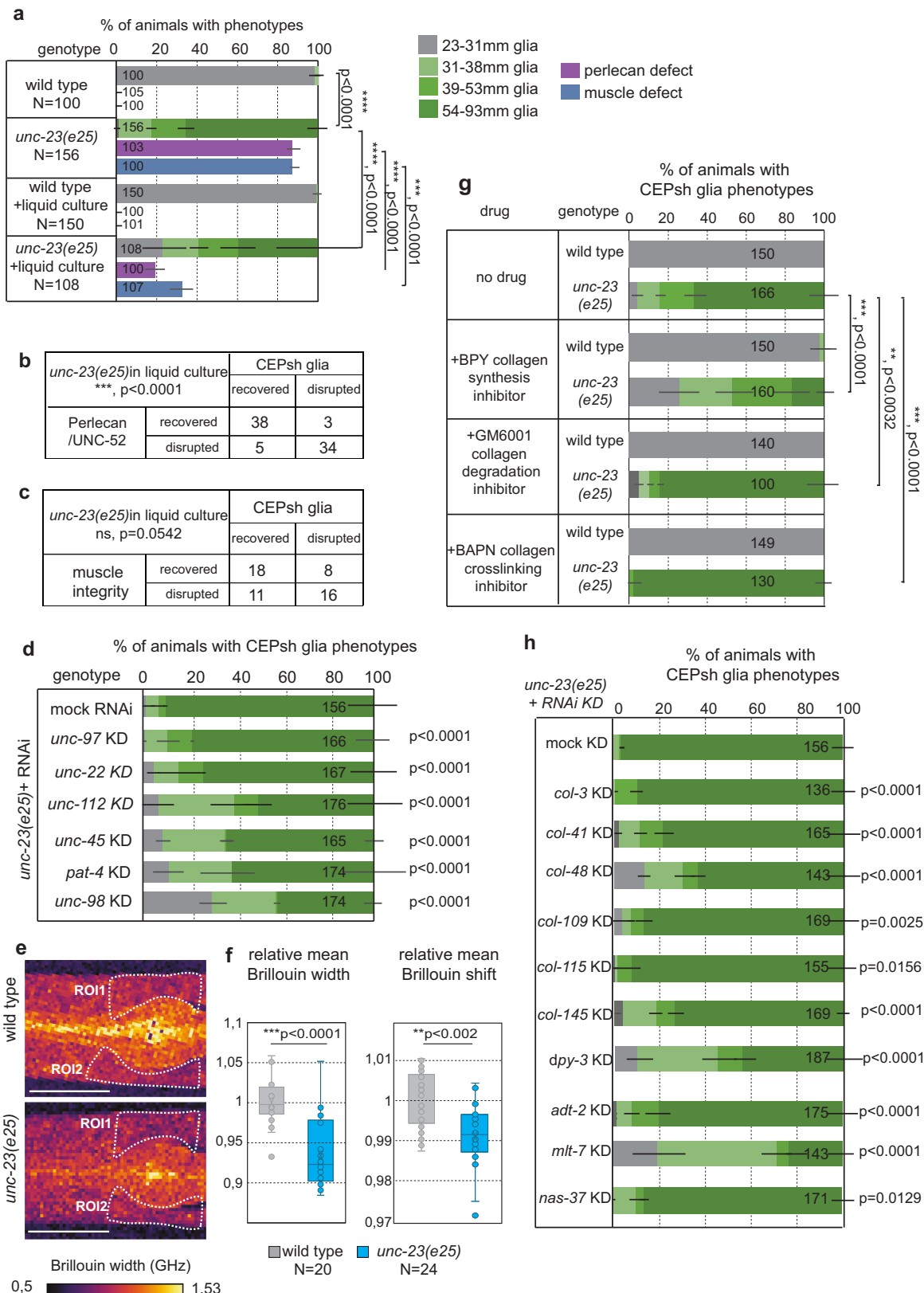

Vertebrate DLG1 is expressed in oligodendrocytes and endothelial cells and LET-413 homolog LRRC1 in astrocytes and oligodendrocytes (https://www.brainrnaseq.org/). Our work opens inroads to consider roles of these factors in vertebrate glial cell architecture.

Epithelial-cell junctions affect integrity and robustness against internal and external forces[6,11], and facilitate tissue sliding during stretch[6]. Stiffness and forces exerted on cells contribute to morphogenesis[6,10,61] while mechanical strain is implicated in neurodegeneration[2]. Astrocytes are sensitive to mechanical tension[62], but how they integrate molecularly such experienced forces to sustain tissue integrity remains understudied. Our findings underscore how glial integrity relies on material properties. Impaired BAG2 is

**Fig. 7 | Altered animal's and environment's biomechanics affect CEPsh glia integrity in *unc-23* mutants. a–c** Growth of *unc-23* mutant animals in liquid media partially suppresses the defects of CEPsh glia, Perlecan matrix and muscle attachment. Liquid-induced recovery of CEPsh glia integrity correlates significantly with the recovery of Perlecan defects (**b**) but not with the recovery of muscle defects (**c**). **d** Genetically-induced paralysis in *unc-23* mutants, by knocking-down (KD) genes of locomotory apparatus, partially suppresses CEPsh glia defects. *p* values from Fisher's test are presented for comparisons to mock RNAi. These and *p* values from two-way ANOVA are also available in Data Source. **e, f** L2 tissue viscosity and elasticity (as measured by the Brillouin spectral linewidth and shift) is decreased in *unc-23* mutants compared to wild-type. Values relative to the wild-type are presented here (for absolute values, see Supplementary Fig. 9b–d). Doted areas are ROIs selected flanking the 2nd pharyngeal bulb dorsally (ROI1) and ventrally (ROI2). These tissue areas neighbor the normal position of glia membranes of the dorsal CEPsh (ROI1) and the ventral CEPsh (ROI2), and present ectopic glial membranes in *unc-23* mutant larvae animals (see "Methods" for more information). The mean Brillouin linewidth and shift of the RO1 and RO2 from distinct animals of the same genotype are grouped together (Brillouin values of ROI1 and ROI2 are not

significantly different in wild-type animals). **g, h** Collagen-modifying drug GM6001 and BAPN enhance CEPsh glia defects in *unc-23* mutants (**g**), while drug BPY or the RNAi knock-down (KD) of collagens or collagen-modifying enzymes (**h**) partially suppress the CEPsh glia defects of *unc-23* mutants. Predicted orthologues of the collagen genes and enzyme genes studied here are presented in detail elsewhere[57]. Asterisks on histogram bars indicate significances compared to mock RNAi KD (**h**). Two-sided unpaired *t*-test used (**f**) or two-sided Fisher's exact test are applied (**a, d, g, h**). Error bars represent mean ± standard deviation. Total animal numbers are noted inside bar graphs (**d, g, h**) or with genotypes in dot-plot graphs (**a, f**). Number of independent experiments for each condition is *n* = 3, unless otherwise noted. Dots, limits, lines, whiskers in box-whisker plots are defined as in Fig. 1. The total number of animals per condition and number of independent experiments, the primary data, and statistical analysis (including two-way ANOVA statistical analysis) are presented in the Source Data. RNAi results and statistics is also provided in Supplementary Table 2. Scale bars, animal axes, reporters, *p* value, ns, and dots, limits, lines, whiskers, error bars in box-whisker and bar graphs are defined as in Fig. 3.

accompanied by reduced tissue viscosity and disruption of ECM components (Figs. 6 and 7 and Supplementary Fig. 9) including of Collagen IV, a major contributor to ECM stiffness that affects local viscosity and the ability of tissues to withstand mechanical forces[9]. Animals experience tension on the order of 10,000 × G, 1000–10,000 times larger on high-viscosity compared to low-viscosity media while differences in undulations size and speed in these media are relatively modest. Our findings suggest that following decreased tissue viscosity and elasticity (as measured using Brillouin microscopy), ECM disruption and glia integrity (as measured using optical microscopy) are more sensitive to high-viscosity than low-viscosity environment (Fig. 7), which is in line with effects of forces and biomechanics on other tissue deformations[61]. Tissue mechanical properties play an intricate role in determining biological function and can serve as indicators for pathology, yet standard mechanobiology techniques used to measure them rely on either invasive procedures or fail to probe the mechanical properties with sufficient high spatial resolution and/or beneath surfaces. Brillouin microscopy offers an unprecedented capability to infer these properties non-invasively, in a 3D and in high spatial resolution. This work, among few others, attests that Brillouin microscopy can highlight early material properties differences between normal and pathological tissues in vivo. It allows us to probe the roles of tissue mechanics in affecting neuroglia tissue architecture and integrity in vivo. Our findings suggest that decreased tissue viscoelasticity affects ECM and neuroglia integrity and that manipulating the ECM and (maybe more surprisingly) manipulating the animal's external environment affects tissue integrity. This understanding may prove valuable in the further future, in the prediction and design of approaches for disease recovery. In this direction, a precise dissection of the biochemical, cellular and material properties of tissues in relation to their environment is necessary. This work highlights an integration of tissue responses to temperature and biomechanical changes. Such multi-disciplinary understanding of biological properties across-scales may later contribute to future applications in disease diagnosis or to future environmental research for a comprehension of biomechanics in animal responses to environmental change.

*C. elegans* animals can be modeled as self-similar, shear-thinning objects, similarly to heart and brain, with gross mechanical properties governed by ECM components[63]. Adding to internal ECM, specialized ECM composes the animal's protective exoskeleton (cuticle) that contributes to force transmission[64]. Physical properties of this ECM are determined by its collagen network[63], and its mechanics depends on its microstructure. Collagen fibers provide ECM and tissues with resistance to traction, and tensile strength to resist deformation, while the cuticle affects the animal's bending

stiffness[63]. Collagen composition can affect biomechanics and shape changes upon applied forces. Here, reducing collagen levels partially safeguards age-dependent glial integrity, while blocking its degradation or crosslinking aggravates the glial defects in *unc-23* mutants (Fig. 7). Thus, manipulating chemically or genetically the internal or external ECM surrounding the juxtaposed glia and epithelia, affects glial integrity regulated by HSP-proteostasis. Our work highlights collagens or collagen-modifying enzymes, that are also associated with age-associated neuropathies[65] or ECM pathologies, glaucoma, glia and neurodevelopmental disorders[66,67].

Tissue responses to forces sometimes involve disaggregation or chaperoning of force-unfolded proteins and proteostasis may support robustness of ECM architecture upon mechanical stress[4,68]. Meanwhile, *C. elegans* epithelia are involved in ECM deposition and force transmission[6,9,10]. Here, epithelial HSP-BAG2-proteostasis responds to mechanical stress upstream of ECM and cell junctions and its compromise causes age-dependent disruption of ECM and glial integrity, upon increased forces. Conversely, reducing the environment's exerted forces, safeguards glia integrity despite impaired proteostasis (Fig. 7). We uncover a proteostasis-tissue-temperature-biomechanics interplay that supports ECM and tissue robustness against mechanical strain. This interplay involves glial-cell junctions connecting glia to tissue and ECM neighbors, to maintain age-progressive integrity of their architecture and support circuit properties. Overall, our work paints a picture of well-balanced and tightly regulated protective mechanism that provides resilience of glial cells to age and environmental stressors.

## Methods
### *C. elegans* methods
Animal handling: *C. elegans* strains are cultured as previously described[18]. Bristol N2 strain is used as wild-type except when Hawaiian CB4856 is used for mapping. Unless otherwise indicated, animals are raised at 20 °C, for two generations under ad libitum feeding conditions and L4 stage animals are scored. Animals used for phenotyping are hermaphrodites (not males).

Germline transformation: unstable extrachromosomal arrays are generated using standard microinjection protocols and integrations are performed using UV with psoralen (Sigma, T6137) and previously established protocols[18].

### Strains and plasmids used in this study
Previously published mutant alleles used in this study are provided by the CGC: LG.II: *unc-52(e444)*, LG.III: *ina-1(gm144)*, LG.IV: *hsp-1(ra807)*, *cima-1(wy)*, *eri-1(mg366)*, LG.V: *unc-23(e25)*, LG.X: *egl-15(n484)*, *nre-1(hd20) lin-15B(hd126)*. Information for newly generated alleles,

mutants, extrachromosomal and integrated arrays is provided in Supplementary Tables 3–6. Information on unstable extrachromosomal and stably-integrated transgenes is provided in Supplementary Tables 4 and 5 and used plasmid list in Supplementary Table 5. Expression patterns of used vectors/ arrays are presented in Supplementary Table 6. Sequences of generated plasmids are available upon request. Oligonucleotide sequences used for genotyping strains and for generating plasmids are provided in Supplementary Table 7.

### Isolation of mutants and genetic mapping
Mutagenesis and genetic screening: animals are mutagenized using 70 mM ethyl methanesulfonate (EMS, Sigma) at 20 °C for 4 h. P0 generation of *nsIs374 LG.III* animals are mutagenized, F1–F2 generations are grown non-clonally. F2 progeny animals are examined on an epifluorescent microscope Zeiss Axioplan 2 (63×/1.4 NA objective, FITCH/GFP filter set (Chroma, Set 51019), or a fluorescent stereomicroscope Leica M165-FC (Objective Planapo 1.6x M-series and Filter set ET GFP–M205FA/M165FC). Animals with aberrant CEPsh glia morphologies are recovered.

Gene mapping: mutant *nsIs374* animals with ectopic CEPsh glia membranes (allele arg5) are crossed with wild-type males of CB4856 Hawaiian strain and recombinant progeny with CEPsh glia defects is isolated. Genetic mapping and SNP analysis highlighted linkage of mutant phenotype to a ~4.6-cM interval from LG V 0.46 cM (SNP K11C4.1: V: 6,898,091) to V 5.11 cM (SNP C26E1: V: 13,293,704). Whole-genome-sequencing detected mutations in 14 genes in this region. Mutants are rescued using fosmid WRM0626bA02, which includes *unc-23*, and tissue-specific expression of *unc-23* cDNA. Mutants are out-crossed at least four times.

### Animal synchronization
Animal synchronization for imaging experiments: Gravid adults are placed on OP50 plates, left for 2 h to lay eggs, then removed. Laid progeny is grown at 20 °C until reaching L4 stage at 48 h. L4 animals are imaged unless otherwise indicated.

Animal synchronization for RNAi/drug treatment experiments: Gravid adults are bleached, and recovered embryos are grown in M9 solution at 20 °C for at least 15 h until L1 larval arrest. Synchronized larvae in L1 arrest are recovered and placed on plates with the relevant treatment.

### Heat-shock and starvation studies
Gravid adults are bleached, embryos are recovered and grown at 20 °C until desired stages are obtained. Synchronized animals are exposed to 30 °C for 5 h (long-term, low heat-shock) or 35 °C for 0.5 h (acute, strong heat-shock), during stage of interest. Controls are kept at 20 °C. After treatment, animals are grown at 20 °C to reach the stage of interest. For starvation studies, synchronized embryos are grown without bacteria and L1 animals are recovered on food, after 24 h or 48 h.

### Dye filling assays
Animals are collected from plates in M9 with 10 μg/ml of lipophilic dye 1,1′-dioctadecyl-3,3,3′,3′-tetramethylindocarbocyanine perchlorate (DiI, prepared in N,N-dimethylformamide), incubated in dye for 30 min in the dark, washed twice with M9, placed on bacteria plates for 3 h. Amphid-commissure axon phenotypes are scored in epifluorescent microscope Zeiss Axiovert (63×/1.4 NA objective, Zeiss Filter set 43 HE Cy3). Different defects in amphid commissures are scored as described below.

### CEPsh glia ablation
Post-embryonic ablation of CEPsh glia is performed using stable genetic array (nsIs180) expressing a reconstituted Caspase-3 in CEPsh glia from the L1 larvae[29].

### Drug applications
Plates containing drugs BPY, GM6001 and BAPN are prepared as follows: 2,2′-Bipyridine (BPY, Thermo Fisher Cat#030569) or GM6001 (Abcam, ab120845) is diluted in DMSO and 3-Amino-propionitril-fumarat (BAPN, Sigma, A3134) is diluted in H2O, each in final concentration 100 mM. Drug-containing plates are prepared by adding drug in both the agar and bacteria, in final concentration of 100 μM. Synchronized eggs are grown on drug plates, at 20 °C, until the L4 larval stage, then used for phenotypic analysis.

### Growth in liquid bacteria media
Embryos synchronized as above are grown at 20 °C in plates containing 1 ml liquid bacteria culture and replenished with liquid media every 12 h. Animals are scored phenotypically at L4 stage. Control animals are prepared identically but grown in solid culture.

### Lifespan assay studies
Lifespan assays were performed using protocols established by others[69]. Specifically, worms were synchronized by egg laying (P0 adults were transferred on fresh plates, allowed to lay eggs and removed after few hours). Synchronization by bleaching was avoided as it may affect animal development and tissue morphogenesis. When F1 progeny reached the L4 larval stage, 120 L4 animals per strain were transferred on NGM plates seeded with UV-killed bacteria and containing carbenicillin (25 μg/ml) and 5-Fluoro-2′-deoxyuridine (FUdR, 50 μM). The use of FUdR causes sterility and allows to follow the parental population without self-progeny production. Daily, alive and dead worms were counted, dead and censored (dried/exploded/) worms were removed from the plates, and alive worms were transferred when necessary to avoid starvation or contamination. The Kaplan–Meier method and the Wilcoxon rank sum test were used for results analysis and statistics. Both the wild-type and *unc-23(e25)* strains used in lifespan assays, contain an integrated array expressing the marker CEPsh::myrGFP; it is unclear if this may slightly affect lifespan compared to N2 strains.

### RNA interference studies
RNA interference (RNAi) is performed as previously described, using vectors expressing double-stranded RNA (dsRNA) or by feeding animals with dsRNA-expressing bacteria. In the first case, knocking-down transcripts in specific cell types is achieved by expressing dsRNA under cell-specific regulatory sequences. In the second case, knocking-down transcripts in nervous system cells is achieved by RNAi applied in genetic backgrounds combining mutations *eri-1(mg366)* and *nre-1(hd20) lin-15B(hd126)* that sensitize nervous system cells to RNAi. Absence of these genetic backgrounds, allow RNAi in other tissues (epithelia, muscle, intestine) without affecting nervous system cells. In this study, RNAi-bacteria-feeding is performed post-embryonically. Synchronized larvae after L1 arrest are fed *E. coli* HT115 with pL4440 vectors targeting specific genes, grown at 20 °C for 3 days and scored as adults unless otherwise indicated. RNAi clones from the Ahringer or Vidal libraries are used (references in Supplementary Table 2). Triplicates for each experiment are performed. RNAi clones presenting statistically significant effects were verified for specificity of the target using DNA sequencing. Imaging of animal phenotypes after certain RNAi applications was performed in Leica M165-FC (Objective Planapo 1.6x M-series and Filter sets ET GFP–M205FA/M165FC and ET RFP–M205FA/M165FC).

### In vivo imaging and phenotypic scoring/quantification
Live imaging is performed on postembryonic stages L1, L2, L3, L4. Animals are anesthetized using M9 buffer containing 20–25 mM sodium azide, mounted on pads of 2% agarose and image acquisition is performed immediately after. Imaging is performed with optical sections of 0.5 μm spacing for the following labeling:

CEPsh glia membranes or nuclei, epidermis, muscle, all axons, AIY neurons, CEP neurons are visualized in vivo using each of *Phlh-17::myrGFP* or *Phlh-17::myrGFP-SL2-NLS-mCherry, Pdpy-7::mKate, Pmyo-3::RFP, Prab-3::mKate-PH, Pttx-3::GFP, Pdat-1::GFP* respectively (Supplementary Tables 4 and 5). Amphid neurons are visualized using dye filling. AIY presynaptic sites and CEPsh glia- AIY synapse contact sites are visualized using the markers *Pttx-3::mCherry::rab-3* and *Phlh-17::CD4::GFP(1-10) + Pttx-3::CD4::GFP(11)* (Supplementary Tables 4 and 5) respectively. Length, area, intensity of fluorescent signals, and distances between fluorescent signals are quantified using Fiji software. Defects of number/position/fragmentation, axon position/ ectopic growth, of amphid or CEP neurons cell body are measured in population.

UNC-52/Perlecan and EMB-9/Collagen IV are visualized using each of CRISPR knock-in strain *unc-52(qy80[mNG+loxP (synthetic exon)::unc-52])* or *emb-9 (qy24[emb-9::mNG+loxP])* respectively (Supplementary Tables 3 and 4). Intensity of fluorescent signals in regions of interest is measured.

Epithelial apical domains are visualized with ApiGreen-GFP marker *Phlh-17::SAX_7delcyt_sfGFP* (Supplementary Tables 4 and 5), a construct derived from sequences of transmembrane protein SAX-7 localized exclusively to apical surfaces in a canonical epithelium[25]. Region of interest for fluorescence signal is defined between the posterior end of the 1st pharyngeal bulb and the anterior of the 2nd pharyngeal bulb (based on images of the bright-field channel). Epithelial *DLG-1* is visualized using marker *Pdlg-1::dlg-1::RFP* (Supplementary Tables 4 and 5), DLG-1 enrichment is quantified ventrally to the 2nd pharyngeal bulb. AJM-1 adherens junctions in CEPsh glia are visualized using marker *Phlh-17::ajm-1-cDNA-CFP* (Supplementary Tables 4 and 5), via the CEPsh glia-specific expression of a previously established CFP-tagged label of AJM-1 protein localization domains[25]. Animals are observed in vivo on a fluorescent stereomicroscope M165FC (Objective Planapo 1.6x M-series, Filter set and ET RFP– M205FA/M165FC) or an epifluorescent microscope Axiovert Zeiss (63×/1.4 NA objective, Zeiss Filter set 43 HE Cy3). Image acquisition is performed using a confocal microscope SP8 Leica (Objective and Filter as above) or a spinning disk Olympus iXplore SPIN SR (100×/1.35 NA objective, Filter set DM D405/488/561/640).

### Time-lapse imaging of postembryonic CEPsh glia
Post-embryonic live time-lapse imaging is performed on synchronized animals, immobilized using Polybeads Microspheres (Polysciences 00876-15), as described previously in ref. 70. No sodium azide is used. Images are acquired using spinning disk microscope Olympus iXplore SPIN SR. Immediately after imaging, animals are recovered by removal of coverslip, addition of M9 buffer, recovery on fresh bacteria plates and grown at 20 °C until the next stage. Individual animals are followed separately. Imaging and recovery are repeated for each of L2, L3, L4 stage.

### Brillouin microscopy and imaging of mechanical properties
Brillouin microscopy measured tissue mechanical properties such as elasticity and viscosity in the GHz frequency range through the interaction of light with the sample's acoustic phonons[24]. The shift and linewidth of the Brillouin scattered light spectrum gives information about the so-called longitudinal modulus which is directly related to the elastic and viscous modulus of the material, respectively. Brillouin imaging is performed using a Brillouin microscope previously described in ref. 54. Briefly, this consists of a commercial Zeiss body (Axiovert 200 M) coupled with a home-built spectrometer based on a 2-VIPA configuration, which provides a precision of 22 and 56 MHz for Brillouin shift and linewidth measurements, respectively, for our measurement parameters (100 ms exposure time, 4 mW optical power on the sample) and a measured spectral resolution of 520 MHz. To account for the finite spectral resolution, we subtract the latter (520 MHz) from the measured linewidth values (the subtraction

corresponds to deconvolution, as detailed in Chan et al.[53]. We note that the effective NA of the objective (0.85) leads to ~100–110 MHz downshift in the Brillouin shift. A 532 nm laser (Torus, Laser Quantum) is used for Brillouin imaging. An 488 nm laser, coaligned with the 532 nm laser, allows for confocal imaging of GFP fluorophores. Wild-type or *unc-23* mutant animals, synchronized at L2 stage, expressing markers of CEPsh glia or UNC-52::GFP are used. Animals are starved for 2 h before experiments to minimize variability of Brillouin signal in the pharynx due to bacteria content. Animals are anesthetized using M9 buffer containing 10 mM sodium azide and mounted on a slide of 2% agarose in 10 mM sodium azide. Images of GFP are acquired before and after the Brillouin acquisition. Brillouin images are acquired with a 40×1.0NA Zeiss objective and an integration time for a single point of 100 ms. The optical power on the sample is kept below 4 mW and no apparent photodamage is observed after imaging. Measurements of refractive index of the animals were acquired by performing label-free 3D Holo-Tomographic Live Cell Imaging of worms, using the Nanolive 3D Cell Explorer Fluo. L2 animals were mounted on slides using Pluronic F127 36% w/v + 1 mM tetramisole solution or embedded in 2% agarose +1 mM tetramisole solution. Brightfield images were acquired using an objective 60× NA 0.8 and fluorescent images were acquired using a FITC filter of the Nanolive Module LED DAPI-FITC-TRITC/Cy5 4X B 000.

### Image analysis and processing
**For analysis of spinning disk/confocal imaging.** Length of distances between structures, intensities and selected areas of interest are quantified using functions "Straight", "Rectangle", "Line Width", "Measure" in Fiji or "Measurement point", "Surfaces" in Imaris. 3D representations are generated using "Surface visualization" in Imaris. Membrane volume quantifications are performed using an ImageJ script written by Christian Tischer at the Advanced Light Microscopy Facility at EMBL, which is described in detail and available at https://git.embl.de/grp-cba/glia-volume-measurement.

**Selecting regions of interest (ROI).** In Fig. 5b, c and Supplementary Figs. 5 and 7, $ROI_1$ is defined from the posterior end of 1st pharyngeal bulb to anterior of 2nd pharyngeal bulb, $ROI_2$ is same size with $ROI_1$ starting at the posterior end of 2nd pharyngeal bulb. In Fig. 5d, UNC-52 signal intensity is obtained on a line parallel to animal's ventral side. In Fig. 6a–d, the distance between DLG-1 AJM-1 is measured between the centers of the 2 signals. In Fig. 6c, d, DLG-1 position is measured with 2nd pharyngeal bulb as external reference. In Fig. 6e, f, distance is from DLG-1 enrichment to the neighboring signal of UNC-52 accumulation. In Fig. 7e, f and Supplementary Fig. 9, ROIs are selected flanking the 2nd pharyngeal bulb and the normal CEPsh glia membranes dorsally (ROI1) and ventrally (ROI2). These areas present ectopic glial membranes in late mutant larvae. Mean Brillouin linewidth and shift of the RO1 and RO2 from distinct animals of the same genotype are grouped together (Brillouin values of ROI1 and ROI2 are not significantly different in wild-type animals).

**For analysis of time-lapse imaging.** Quantifications of glia length are performed using the function "Measure" in Fiji software. Videos are made using Imaris Animation feature. Images have been centered around the centroid of the posterior CEPsh glia in a max-intensity projection. Image alignment, registration, and conversion to movies was performed using a Python code, which is available in the Source Data.

**For imaging with Brillouin microscopy.** Images are acquired as described above. Spatial maps of elasticity and viscosity are plotted from the acquired data, and function "Measure" and Lookup table mpl-Inferno in Fiji and adjusting the "Brightfield and Contrast" at 7.5–8.2 or 1.02–2.05 respectively. ROIs are selected flanking the 2nd pharyngeal bulb dorsally (ROI1) and ventrally (ROI2), since these tissue areas neighbor the normal position of glia membranes of the dorsal CEPsh

(ROI1) and the ventral CEPsh (ROI2), and present ectopic glial membranes in late mutant larvae animals. The pharynx depression allows for defining the same area in different animals. ROIs selected exclude the pharynx to avoid signal variability due to bacteria content. ROIs are selected with the function "Polygon selections" in Fiji and mean values for the shift/linewidth are computed to appropriately represent the mechanics on a tissue level. From the raw Brillouin shift and FWHM linewidth values we compute the Brillouin elastic and viscous contrast analogous to Antonacci et al.[55]. We quantify the mean value of Brillouin linewidth/shift of the selected ROIs, since we are interested in the overall tissue areas neighboring the wild-type CEPsh glial membranes, and which present ectopic glial membranes in the later stages of mutants. The morphology of ectopic membranes varies in different animals but always populates the selected ROIs. Consequently, spatially higher resolved measurements or analysis within these areas would not be more informative for this study, and the mean measurements of ROIs are presented.

### Statistical analysis

Sample sizes and statistical tests are chosen based on previous studies with similar methodologies; the data met the assumptions for each statistical test performed. No statistical method is used to decide sample sizes. All population experiments are performed at least three independent times using at least 100 individuals in total, comprised of independent biological replicates and yielding similar results. Independent transgenic lines or individual days of scoring for mutants are treated as independent experiments for the standard deviation. The same samples were not measured repeatedly (measurements were taken from distinct samples). No multiple comparisons were performed. Error bars represent 95% confidence intervals. Results are represented as mean ± standard deviation (SD), unless otherwise noted. P values are calculated using GraphPad Prism. Fisher's and two-way Anova tests are performed (using Prism, GraphPad) to quantify statistical significance of all phenotypes in population studies, heat-shock experiments, RNAi experiments, drug applications, liquid culture experiments, contingency tables for defect correlations. The student's two-tailed (two-sided) unpaired t-test is performed (using Prism, GraphPad) for comparison of normally distributed values of glia size, nuclei distances, axon lengths, synaptic sites/contacts areas and intensities, signals of UNC-52, DLG-1, AJM-1, and Brillouin values. Specifically, the following statistical tests are used throughout the manuscript: Two-sided unpaired t-test is used for the dot-plots in Figs. 2g–j, 3b, c, e, f, 5c, g, 6c–e, 7f and Supplementary Figs. 2c, d, 3c, 5b, e, 7b, 8b, c, e, f, 9e, f. Two-sided unpaired Chi-square test is used for the bar charts in Figs. 2d, f, 3h, m, k, 4b–f, 5a, 6f, g, 7a, d, g, h and Supplementary Figs. 1d, 3a, 4, 6, 8a, d, and for the contingency tables in Figs. 2l, 3i, l, 5e, f, 7b, c and Supplementary Figs. 1c and 2b. The Kaplan–Meier method and the Wilcoxon rank sum test are used for results analysis and statistics of lifespan assays. The number of degrees of freedom equals the number of pairs minus 1. All sample numbers and p values are presented in relevant figures, and are provided together with all primary results in the Data Source.

### Blinding and randomization during data analysis

Blinding during data analysis is not performed. Samples are allocated to groups of the genetic background (genotype), detected by standard genetic/genomic approaches. Samples are randomly selected within these groups, based on previous studies with similar methodologies. Phenotypic analysis is performed by more than one individual researcher for experiments of RNAi screens and image quantifications CEPsh glia sizes and age-dependent defects.

### Reporting summary

Further information on research design is available in the Nature Portfolio Reporting Summary linked to this article.

## Data availability

All data supporting the paper's conclusions are included in the main text and Supplementary Information file. Source image files have been uploaded to Figshare [https://doi.org/10.6084/m9.figshare.25283635] repositories. Further information and requests for resources can be directed to the lead contact. Source data are provided with this paper.

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

## Acknowledgements

We thank, Michel Labouesse, Jean-Louis Bessereau Lena Kutscher, Alba Diz-Munoz, Shai Shaham, Michael Dorrity, Cori Bargmann, Anne Ephrussi and colleagues in the Developmental Biology Unit at EMBL Heidelberg for scientific input. G. R. acknowledges colleagues at the FENS Kavli Network of Excellence and the Interdisciplinary Center for Neurosciences for discussions. We thank the Heiman, Shaham and Sherwood labs for sharing published reagents. We thank the Rapti group member Chiara Mungo for sharing unpublished reagents. Some strains were provided by CGC, funded by NIH (P40 OD010440). Wormbase (www.wormbase.org) was used for experimental design and analysis. We thank the Advanced Light Microscopy Facility, Marko Lampe, Stefan Terjung and Christian Tischer, at EMBL, and Sarah Kaspar from the EMBL Centre for Statistical Data Analysis for valuable technical support. R.P. acknowledges support of an ERC Consolidator Grant (no. 864027, Brillouin4Life), and the German Center for Lung Research (DZL). This work was supported by the European Molecular Biology Laboratory.

## Author contributions

G.R. and F. Coraggio conceived the project. F. Coraggio., M.B., F. Caroti, S.R., and G.R. performed all experiments. G.R., C.B., and R.P. designed the Brillouin experiments. G.R., F.C., and C.B. performed Brillouin experiments. G.R. led the project and wrote the paper together with F. Coraggio and input from all authors.

## Funding

## Competing interests

The authors declare no competing interests.
