## [Peer Review File · Nature Communications]

Age-progressive interplay of HSP-proteostasis, ECM-cell junctions and biomechanics ensures *C. elegans* astroglial architectureREVIEWER COMMENTS

Reviewer #1 (Remarks to the Author):

In this manuscript, the authors use *C. elegans* CEPsh glia as a model to study glia morphogenesis. In a forward genetic screen, they identified a loss of function mutant of *unc-23* with defects in CEPsh glia morphology. Although the *unc-23* phenotypes appear to be developmental phenotypes, the authors claimed that *unc-23* were important for the maintenance of glial integrity. They then concluded that *unc-23* functions in epithelial cell to regulate glia through ECM. More strikingly, even with the dramatic changes of ECM in *unc-23* mutants, epithelial cells seem to be as normal as in wild type animals. With these surprising findings, they concluded that the abnormality of ECM but not epithelial cells caused CEPsh defects in *unc-23* mutants. They went on to show stresses can modify *unc-23* phenotypes and concluded that *unc-23* and its associated signals are important to maintain glial integrity under stress. Overall, it is an interesting study, but many of the conclusions need to be carefully reevaluated.

Main concerns.

- 1) One of the main points the authors keep bringing up is that this *unc-23*- associated mechanism is important for maintenance of glial integrity. However, the data clearly show that *unc-23(lf)* animals already have strong phenotypes as early as in the L2 stage, when CEPsh glia are still in the developmental stage. Therefore, all defects shown in this study are likely originated from CEPsh developmental abnormality, and it is possible that this *unc-23*-associated mechanism has nothing to do with maintenance of glial integrity. If the author really want to show it is involved in maintenance of glial integrity, they need to examine the effect of knock out/suppress of *unc-23* function after CEPsh finish development (for example in adult animals).
- 2) The author concluded that all *unc-23* neuronal phenotypes are associated with glial abnormality. However, there were no firm data to exclude the possibility that those neuronal phenotypes were directly caused by ECM defects in *unc-23(lf)*, and this may have nothing to do with glia. the only data may support this conclusion is that killing CEPsh in L1 suppressed the neuronal defects in *unc-23(lf)* worms. This data itself is very puzzling. Does that mean CEPsh glia function are not important for neuronal development, or the glia in *unc-23* mutants don't have any normal CEPsh glia function. What happen if killing CEPsh in L1 control animals? Do neurons develop normal as shown in Fig. 3M? If this is true, why the previous studies from the same lab show CEPsh is critical for nerve ring formation.
- 3) Based on that high temperature and starvation partially suppressed glial "integrity" phenotypes in *unc-23* mutants, the authors concluded that "that conserved functions of the UNC-23/BAG2- HSP-1/Hsp70/Hsc70 - DNJ-13/ DNAJB1 complex integrate tissue responses to temperature and nutritional changes to safeguard the the lifelong integrity of glial cells and circuit architecture". This is a such strong statement, but the data only show the correlation between high temperature or starvation and *unc-23(lf)* phenotypes. No direct evidence to support this conclusion. At least some critical control experiments need to be done before reaching such conclusion. One possibility is that high temperature

or starvation could directly affect ECM, and this ECM change may have nothing to do with unc-23. It is also possible that temperature and starvation just affect the nature of unc-23 mutations used in this study.

4) The author show unc-23 mutants doesn't affect epithelial polarity or integrity, and this is opposite to what has been described in a 2015 paper from Donald Moerman lab. Why? Any discussion?

5) Based on "Perlecan and COL4A5 present abnormal localization in unc-23 mutants, are depleted from areas neighboring normal CEPsh glia positions, and accumulate neighboring ectopic glial territories (Fig. 5B-E, S6)" and "Perlecan matrix abnormalities are significantly correlated with glial defects in unc-23 mutant L4 larvae but precede these defects in L2 mutants (Fig. 5F-G)", the author concluded that "ECM components function downstream of UNC-23/BAG2 for glia integrity". However, it is also possible that those mis-location of Perlecan and COL4A5 are the results rather than the cause of abnormal morphology of CEPsh glia.

6) In Fig. 6 the author claimed that "Similarly, mutant CEPsh glia defects are partially suppressed upon bacteria-mediated knock-down of DLG-1 AJM-1, LET-413, LET-805, MUA-3 in the nervous system, using mutations that enhance RNAi-sensitivity in the nervous system (Fig. 6F, Methods). These components are largely absent from postembryonic neurons, thus this knock-down affects their glial transcripts. ". Does that mean it has to be CEPsh glia? could it be other glia ?

Reviewer #2 (Remarks to the Author):

In this review, I addressed only the Brillouin measurements presented in the manuscript -- the other parts of this work are outside my expertise. The authors used Brillouin microscopy to measure the high-frequency elasticity and viscosity of mutant and wild-type *C. elegans* worms. The main conclusion from these measurements was: "different animal biomechanics precede and underly age dependent disruption of ECM and glia".

To correctly image the storage and the loss moduli using Brillouin microscopy, the density and refractive index of the medium should be known or measured. I could not find information on these parameters, therefore I am not convinced about the reliability and interpretation of the data presented. This is a very critical point.

Another point is how the shift and linewidth of the Brillouin spectra were measured given the low spectral resolution of the VIPA spectrometer and the large spectral broadening at NA=1. What are the accuracy and precision of the obtained shift and linewidth values? based on what criteria the ROIs were selected?

Additional points are (1) A quantified analysis of the variation of the mean viscosity and stiffness in Figure S7 and its meaning/implication to the biological questions investigated is missing, (2) Why only mean values were analyzed in Figures 7F and S7? could any useful spatial information be obtained from

the Brillouin images?, (3) The added value and importance of the Brillouin measurements to this work is very unclear, particularly the vague conclusion arising from these measurements. This is a very critical point, (4) Why relative values were presented? would statistical tests on the relative and absolute values yield same results/conclusions?

Reviewer #3 (Remarks to the Author):

NCOMMS-23-00958

Lifelong interplay of HSP-proteostasis, biomechanics and ECM-cell junctions ensures *C. elegans* astroglial architecture

The authors described the morphological size changes of CEPsh glia during *C. elegans* larval development and adult stage (day of adulthood not specified). At L4, they assessed CEPsh glia and their colocalization to hypodermis, muscles, and neurons. In a screen, they identified *unc-23(arg5)* mutant showing longer but less volume of CEPsh glia. They confirmed that the *arg5* mutation is causal to the phenotype by rescuing with wild-type genomic DNA of *unc-23*. Furthermore, another mutant *unc-23(e25)* phenocopied *unc-23(arg5)*. Expression of wild-type UNC-23 in the hypodermis also rescued *unc-23(e25)*, implicating that *unc-23* acts from the hypodermis. Moreover, using the orthologous mouse BAG2 sequences also rescued the *unc-23(e25)* mutant phenotypes, suggesting conservation in function. They showed the *unc-23* mutants have synaptic and axon defects (unclear at which developmental stage. Maybe L4). Knockdown of *hsp-1* or *dj-13* fully rescues the morphological defects in CEPsh glial. However, starvation or heat stress had minor effects on CEPsh glial phenotypes. They found partial and weak rescue with knockdown of members of the basement membrane (BM) extracellular matrix (ECM) (i.e., RNAi against *unc-52*, *emb-9*, and *lam-1*). Strikingly, mutation in *unc-52(e444)* almost completely rescued *unc-23(e25)* glial phenotypes. Disrupted UNC-52mNG, specifically at the ventral side, correlated with disrupted glia at L2 (not or minor at L4). Partial rescue of *unc-23(e25)* by knocking down genes in body wall muscles and integrin signaling. Interfering with collagen synthesis (pharmacological inhibitor or KD of collagens) led to a partial or weak rescue of *unc-23(e25)* glia phenotypes, whereas inhibiting collagen synthesis or crosslinking aggravated the *unc-23(e25)* glia phenotypes.

Overall, the study is well designed, and the experiments are well conducted. I love the beautiful videos. However, conceptually, it is known that *unc-23/BAG2* binds the ATPase domain of the chaperon HSP-1/Hsc70 and thereby regulates their activity, and in *C. elegans*, the maintenance of muscle attachments (PMID: 26435886). The novelty here is that interfering with *unc-23*, *hsp-1*, BM or muscles affects the morphology of glial cells. A major limitation is that besides one lifespan, none of the experiments have been performed during ageing.

Major

1) It seems like hyperbole to make any lifelong (title, abstract) or ageing claims (text) when all

experiments have been performed during larval development L2-L4 and some mention the adult stage (without specifying which day of adulthood). In general, age-related phenotypes start after reproduction. This is usually post day 8 of adulthood (whereby L4 is day 0 of adulthood per definition). Either show age-related, ageing, or lifelong (that means until death) implications of unc-23 or remove these statements from the manuscript. Furthermore, the effects of UNC-23 were not assessed post-development to make any ageing claims (either us RNAi starting from day 1 of adulthood or other techniques to block the function of unc-23).

1.1) p5. lines 138-177. The title of this paragraph says “age-dependent” but all experiments looking at glial morphology were done during larval development. Particularly, lines 153-156, they claim “unc-23 mutants present age-dependent defects” but look at larval stages 1 and 2 (L1 and L2) in development.

1.2) p6. Lines 180-202. Again, the title of the paragraph says “age-processing” and “aging” but base this on day 5 adult animals and on a single lifespan assay. For age-dependent, one would assume a phenotype that is normal in young adults (day 1) that becomes progressively worsen the following days (preferably starting post reproduction (day 8)). Thus, to make such a claim, the authors would need to alter the unc-23 function starting at adulthood and have at least 3 measuring time points to say “age-processing”. Also, line 198. They show that unc-23(e25) mutants have a shortened lifespan compared to the wild type, and this is based on a single lifespan assay that was not even carried out until the end (i.e., 40% of the wild type were still living). Furthermore, this is not decreased “longevity”.. this is decreased lifespan. Per definition, longevity means long-lived.

1.3) p.10 lines 380-382. claim “age-dependent shearing and nuclei fragmentation (fig 2)” but experiments were done during development (fig 2K-L: during L2 nuclei assessments).

2) As mentioned above, it is known that unc-23/BAG2 binds the ATPase domain of the chaperon HSP-1/Hsc70 and thereby regulates their activity, and in *C. elegans*, the maintenance of muscle attachments (PMID: 26435886). Some mechanistic or genetic insights would provide some novelty. For instance, it is remarkable how well hsp-1 rescues the glial phenotypes. In an unc-23 mutant with hsp-1(RNAi), is the UNC-52::mNG ventral basement membrane rescued? Thus, is the phenotype due to improper UNC-52 localization? Or is it due to shear forces from the muscle that disrupts the glial morphology? The latter could explain why unc-52(e444) mutants rescue glial defects of unc-23 mutants. There should be at least some discussion pointing to novel insights.

Minor

1) p.10 lines 363-364 and Fig. 7H. The suggested orthologues of these cuticular collagens are wrong. See <https://doi.org/10.1016/j.mbplus.2018.11.001> for collagen classes.

2) Fig S6: they say EMB-9 in magenta, but it looks green to me. Also, the job of a collagen is to be secreted and integrated into an ECM... for collagen EMB-9, it is integrated into the basement membrane. Unclear why the authors measure these green dots that look like intracellular EMB-9 (are these dots artifacts of the transgene? Aggregation of nMG? in other words, do these green dots represent functional EMB-9?... since ROI1 has less EMB-9 based on these green dots (intracellular?), this suggests may be less synthesis or faster degradation of EMB-9 in that region... can you rescue this by hsp-1 RNAi? In other words, is the lower EMB-9 due to the overactivation of chaperon hsp70c leading to degradation

or turnover of EMB-9?

3) BTW, next time, if the authors want a blind review, they should remove all their info:
p12. line 472: "Further information and requests for information, reagent and resource sharing should be directed to and will be fulfilled by the Lead Contact, Georgia Rapti (grapti@embl.de)."

Reviewer #4 (Remarks to the Author):

In this manuscript, the authors use fluorescence-based markers to track glial cell development and architecture in *C. elegans*. Using classical mutagenesis screens, tissue-selective RNA interference and fluorescence microscopy, the authors show that *unc-23/BAG2* is required in epithelial cells for correct CEPsh glial formation, positioning and association with neuronal and epithelial junctions. Defective glial length/volume and positioning in *unc-23* mutants are dependent on aberrant localisation of the ECM protein, perlecan (*UNC-52*), collagen accumulation and tissue mechanics. The work is thorough, convincing and exciting, but the links to other proteostasis factors requires further interrogation. Conceptually the work is suitable for publication in Nature Communications, but the authors should perform some additional experiments to strengthen their conclusions before publication.

Major comments

1. In Figure 3, the authors show that AIY neuron and Amphid neuron morphology is disrupted in *unc-23(e25)* mutants. Are these phenotypes also rescued by expression of wild type *unc-23* in glial cells (as is observed with glial phenotypes).
2. RNAi against *hsp-1* or *dnj-13* suppresses the glial defects in *unc-23(e25)* mutants. I found this interesting but also quite confusing. If working with *UNC-23*, why would depletion of *HSP-70* or *DNJ-13* not cause similar defects as *UNC-23* knockdown? One possibility is that *hsp-1* or *dnj-13* RNAi can induce a heat shock response by activating *HSF-1*. This could then protect against loss of *unc-23/BAG2*. The authors should test if this is the case by performing experiments with *hsf-1(RNAi)* and *hsf-1(sy441)* mutants.
3. As mentioned above, the impact of *hsp-1* and *dnj-13* RNAi on glial phenotypes is very interesting. I agree that the simplest conclusion is that *HSP-1* and *DNJ-13* are acting in the same tissue as *unc-23*. However, the authors should perform *hsp-1* and *dnj-13* RNAi selectively in glia (as was done for *unc-23*) to determine whether this is the case.
4. Exposure of animals to low-level chronic heat shock does not appear to have much effect on glial phenotypes. The authors should also test the effect of an acute, strong heat shock (e.g. 35C for 30 minutes).
5. The links between mechanical stress/impact and glial architecture are very interesting. Are the effects of starvation observed in Figure 4 simply because the worms now move less?

6. The lifespan of wild type animals in Figure S3C is very short. What were the conditions of this experiment? If conducted at 20C, these lifespan analyses should be repeated.

7. There are many examples of mislabelled panels in figure legends and incorrect references to figure panels in the main text (e.g. The legend of Figure S3 references panel D instead of panel C). The authors should go through their manuscript carefully and correct all these errors.

Minor comments

1. Supplemental table headings have been cut-off

2. Legend in Figure 5A mis-aligned

REVIEWER COMMENTS

Reviewer #1 (Remarks to the Author):

In this manuscript, the authors use C. elegans CEPsh glia as a model to study glia morphogenesis. In a forward genetic screen, they identified a loss of function mutant of unc-23 with defects in CEPsh glia morphology. Although the unc-23 phenotypes appear to be developmental phenotypes, the authors claimed that unc-23 were important for the maintenance of glial integrity. They then concluded that unc-23 functions in epithelial cell to regulate glia through ECM. More strikingly, even with the dramatic changes of ECM in unc-23 mutants, epithelial cells seem to be as normal as in wild type animals. With these surprising findings, they concluded that the abnormality of ECM but not epithelial cells caused CEPsh defects in unc-23 mutants. They went on to show stresses can modify unc-23 phenotypes and concluded that unc-23 and its associated signals are important to maintain glial integrity under stress. Overall, it is an interesting study, but many of the conclusions need to be carefully reevaluated.

Response:

We thank the reviewer for their report and valuable comments that help to enhance the quality of our manuscript. We have addressed in detail each comment and suggestion. We performed the suggested experiments and here and in the revised manuscript we present the results, that largely support our previous conclusions. We have also edited the revised manuscript accordingly, to include clarifications and considerations related to each one of the reviewer's comments. Importantly, some sections subtitles are also shortened according to communicated formatting instructions.

Main concerns.

1) One of the main points the authors keep bringing up is that this unc-23-associated mechanism is important for maintenance of glial integrity. However, the data clearly show that unc-23(lf) animals already have strong phenotypes as early as in the L2 stage, when CEPsh glia are still in the developmental stage. Therefore, all defects shown in this study are likely originated from CEPsh developmental abnormality, and it is possible that this unc-23-associated mechanism has nothing to do with maintenance of glial integrity. If the author really want to show it is involved in maintenance of glial integrity, they need to examine the effect of knock out/suppress of unc-23 function after CEPsh finish development (for example in adult animals).

Response:

We thank the reviewer for their valuable remarks. We agree that distinguishing developmental from maintenance effects of genes for the glial cell architecture is key. CEPsh glial cells of *unc-23(lf)* animals have only mild defects in the L2 stage that worsen progressively throughout later stages (Figure 2f,2j). Post-embryonic *unc-23* and *hsp-1* RNAi after L1, affect CEPsh glial integrity similarly to their genetic mutants (Figure 4B-C, Methods). The CEPsh glial architecture is largely established before the L2 stage, as explained below, thus the described glial defects are maintenance abnormalities and not developmental defects.

Specifically, the development of CEPsh glial cell architecture is patterned and established during embryogenesis and by the L1 stage: CEPsh glia grow embryonic non-branching membrane processes that then get ramified, and this architecture is patterned and established in embryogenesis and L1 stage (Rapti et al, 2017; this manuscript). The key characteristics of the mature CEPsh glial cell architecture, that are detectable in the mature brain neuropil by connectomics (Sulston et al, 1983; Cook et al, 2020; Witvliet et al, 2021) and our own work (Rapti et al, 2017; this manuscript), are largely established before the L2 stage: 1. the ramifications of the early non-branching CEPsh glia membranes (Fig. 1A), 2. the membrane tiling of the 4 CEPsh glia to generate a circumferential ring of glial cells enveloping the brain axonal ring (*nerve ring*) (Fig. 1A), 3. the pairing of CEPsh glial membrane processes with axons to establish tripartite-synapse ensheathment (as shown by connectomics throughout stages, in Witvliet et al, 2021). Beyond the L2 larval stage, the CEPsh glial cells enlarge their surface to maintain this already-established architecture (Fig.1B, 2E). Similarly, most nerve ring axons form their architecture before L2 and then enlarge relative to animal size, through a maintenance process driven by dedicated mechanisms that differ from those driving their. Disruption of these maintenance mechanism causes neuronal defects starting from L2 and worsening with age progression. (Benard & Hobert, 2006; Chen et al., 2001; Pocock et al., 2008; Sasakura et al., 2005; Zallen et al., 1999). Similarly, our study shows that *unc-23* mutants do not present defects in the establishment of CEPsh glial architecture and CEPsh glia employ dedicated mechanisms for their postembryonic maintenance, distinct to their development. We appreciate that clarifying these considerations can contextualize our findings. We edited our revised manuscript to clarify these as follows at the end of this answer.

Nevertheless, we agree that it is interesting to know if *unc-23* is needed in adulthood for CEPsh glia integrity. Following the reviewer's suggestion, we subjected late L4 animals to RNAi by bacteria-feeding to knock-down *unc-23* only in adulthood and quantitatively analyzed the CEPsh glia in day-1, day-3 and day-5 adults. Vector-induced RNAi was not performed due to the lack of regulatory elements inducing expression only in adult epithelia while the *hsp-16* heat-inducible promoter was not used for time-controlled expression as it interferes with heat responses. Yet, bacteria-induced RNAi knock-down in adults causes significant, progressive defects in CEPsh glial cell architecture (revised Figure S3). These defects are milder than those caused by *unc-23* genetic mutation, which is not surprising since RNAi has partial effect, *unc-23* wild-type transcripts are produced before RNAi application, and the protein appears to be stable (only mild defects are detected in CEPsh glia of L4 animals after RNAi in L1 stage onwards, in Fig. 4C and revised Fig. S3). Interestingly, wild-type animals present some CEPsh glial cell defects also in 5-day adults onwards, yet *unc-23* RNAi knock-down causes significantly increased CEPsh glial defects. These results allow us to conclude that UNC-23 function is likely required also in adults for maintenance of glial cell architecture, as discussed in the revised manuscript. We thank the reviewer for the experimental suggestion that strengthens our manuscript.

Moreover, inspired by the above remarks, we sought to identify circuit defects of *unc-23* genetic mutants that become significant after larval stages and progressively aggravate in adults. We detect that the anterior processes and tips of CEPsh glia in *unc-23* mutant adults show extensive defects, of misplacement and losing attachment to neighboring tissues. These defects were not detected in L4 animals (previous submission, Fig. S1B) but are detected only in adult stages (current analysis, Fig. S1B-D). They become significant in day-1 adults and age-progressively deteriorates throughout day-1, day-3 and day-5 adults (Figures S1B-D). Secondly, AIY synaptic vesicle markers apposing to CEPsh glia are sparser but not decreased in L4 mutant animals (Figures 3), but synaptic vesicle content also presents overall decrease in day-5 adult animals of *unc-23* mutants compared to wild-type (Figures S3). In our revised

manuscript, we present representative images and quantifications of these in Figures S1 and S3, and discuss in the text the above considerations and results as follows:

In Introduction, in page 3:

“In *C. elegans*, the brain-like neuropil composed of ~180 axons is patterned in embryogenesis and completes its formation before the 2nd larval stage (L2). Afterwards, the neuropil’s components enlarge relative to animal size to maintain the underlying architecture. The neurons maintain their structural and functional characteristics, correct positions and connections within their ensembles. This neuronal maintenance requires active, dedicated mechanisms with roles that differ from those acting in development: Ig-containing secreted or transmembrane proteins, L1 CAM and FGFR orthologs sustain positioning of neurons, axons²⁴, and synapses²⁵, while hemidesmosomes and spectrin protect axons from degeneration²³. Disruption of these maintenance mechanisms causes neuronal defects sometimes starting from L2 stage and worsening as the animal matures^{24,26–28}. Less is known about mechanisms maintaining glial cell architecture.”

In Results, in pages 5-8:

“CEPsh glia in the embryo grow thin, non-branching processes, that then transform in astrocyte-like, ramified membranes, associating with axons and synapses^{21,33}. This transformation is largely established by the L1 stage (Fig.1A). Key characteristics of the mature architecture of CEPsh glia, detectable in the mature neuropil by connectomics^{37,40,41} and our own work (Rapti et al, 2017; this manuscript), are largely established before the L2 larval stage: 1. the ramifications of the early non-branching membranes of CEPsh glia (Fig. 1A), 2. the membrane tiling of the 4 CEPsh glia to generate a circumferential ring of membranes enveloping the brain axonal *nerve ring* (Fig. 1A), 3. the pairing of CEPsh glial membrane processes with axons to establish tripartite-synapse ensheathment (detected by connectomics throughout the animal’s stages⁴¹. Beyond the L2 stage, the CEPsh glial cells expand their surface in relation to animal growth, to maintain this already-established architecture (Fig. 1B, 2E).”

[...]

“Interestingly, certain defects of CEPsh glial cell architecture appear only in *unc-23* mutant adults. Specifically, *unc-23* mutant adults -but not larvae- present defective anterior membrane tips, that lose attachment with neighboring tissues or show altered morphology, in an age-progressive manner (Fig. S1). Also, neuronal synaptic vesicles associated with CEPsh glial cells present different defects in L4 and adult *unc-23* mutants (see below and Fig S3). To investigate if UNC-23/BAG2 function is also required in adulthood for age-progressive maintenance of glial architecture, we subjected wild-type animals of late L4 stage to *unc-23* RNAi and assessed the CEPsh glial cell architecture in different days of adulthood (Fig. S3A). Interestingly, *unc-23* RNAi in wild-type adult animals results in defects in the architecture of ramified CEPsh glial membranes, that worsen progressively from day-1 to day-5 adults. These defects are similar to defects in L4 animals resulting from RNAi in L1 animals, but milder than the defects in the genetic mutant suggesting that the wild-type UNC-23 accumulating before RNAi application is a rather stable protein (Fig. S3). These findings are in line with UNC-23/BAG2 acting for age-progressive maintenance of glial cell integrity. Overall, UNC-23/BAG2 is required for maintaining CEPsh glia integrity and its functional loss results in age-progressive disruption of glial architecture with less compact, filamentous, discontinuous membranes, occupying ectopic territories (Movies S4-S5).”

[...]

“5-day adults of *unc-23* mutant with defective CEPsh glia present decreased density, like in L4 animals, as well as decreased overall intensity of presynaptic vesicle markers, which is not

observed in L4s (Fig. S3B-C). The latter is a hallmark of deteriorating neuronal healthspan, as described in previous studies ⁴¹. Thus, CEPsh glial defects are succeeded by additional neuronal defects in *unc-23* mutant adults, compared to larval animals.”

[...]

“Importantly, the defects of *unc-23* mutants are suppressed by RNAi of *hsp-1*, *dnj-13*, or *hsf-1* (after L1), demonstrating that postembryonic Bag2/HSP acts after development of CEPsh glia for maintenance of their architecture.”

In Discussion, in page 13:

“Dedicated mechanisms are employed for postembryonic maintenance of glia, distinct to those acting for their development.”

[...]

“The UNC-23/BAG2 acts for age-progressive disruption of CEPsh glial cell architecture while it appears largely dispensable for glial cell and circuit development, established by the L2 larval stages (introduction, Fig. 2-3).”

[...]

“Interestingly, mutants with functional disruption of UNC-23/BAG2 and deteriorating CEPsh glial integrity present misplaced synaptic vesicle densities in larvae accompanied in later adulthood by decrease in overall content of synaptic vesicle densities, previously described as a sign of deteriorating neuronal healthspan ⁴¹. Therefore, maintenance of proper CEPsh astroglia architecture is key for circuit maintenance.”

In Figure S1, Legend

Anterior processes of CEPsh glia are defective in *unc-23* mutant adult but not in L4 stage.

A-B. Each of the 4 CEPsh glia (green) has an anterior process and tip arriving to the sensory endings of the *C. elegans* nose. These are shown here in lateral view of entire CEPsh glia (Ai, Bi, Biv,) or lateral view of their anterior glia processes (Aii, Bvii, Bx) or zoom view of their tips (Aiii, Bviii, Bxi) or in cross-sectional view of their tips (Aiii, Biii, Bvi, Bix, Bxii). Anterior processes and tips of CEPsh glia are largely not defective in *unc-23* mutants compared to wild-type animals in L4 stage (Bi-vi), but they present defects of mispositioning and fragmentation in adult stages (Bvii-Bxi). **C-D.** These CEPsh glia defects appear in significant number of adults *unc-23* mutant animals (C) and they are age-progressive (D). Scale bars, 10µm. Animal axes as in Figure 1.

In Figure S3, legend:

“UNC-23/BAG2 acts in larval and adult stages and affects synapse aging and animal lifespan.

(A). Subjecting wild-type adults in *unc-23* RNAi results in significant defects in CEPsh glial cell architecture, that worsen progressively throughout days 1 and 5 of adulthood.

(B-C). In five-day old adults, AIY presynaptic vesicles (by RAB-3::mCherry marker, magenta) have decreased mean and sum intensity in *unc-23* mutants (Ai-iii, **B**) compared to wild-type animals (Biv-vi, **C**). *, gut autofluorescence. n=12 animals per genotype. unpaired t-test. Animal axes, P values as in Fig. 3. **D.** *unc-23* mutants show decreased lifespan compared to wild-type animals, as assessed by the median lifespan (50% survival) and the maximum lifespan (5% survival) of the two populations. Lifespan assays were performed with or without the use of FuDR. n ≥ 100 animals per genotype. Results were analyzed and compared with the Kaplan–Meier method and the Wilcoxon rank sum test. p value < 0,05 in lifespan experiments, with or without the use of FuDR.”

2) The author concluded that all *unc-23* neuronal phenotypes are associated with glial abnormality. However, there were no firm data to exclude the possibility that those neuronal phenotypes were directly caused by ECM defects in *unc-23(lf)*, and this may have nothing to do with glia. The only data that may support this conclusion is that killing CEPsh in L1 suppressed the neuronal defects in *unc-23(lf)* worms. This data itself is very puzzling. Does that mean CEPsh glia function are not important for neuronal development, or the glia in *unc-23* mutants don't have any normal CEPsh glia function. What happens if killing CEPsh in L1 control animals? Do neurons develop normal as shown in Fig. 3M? If this is true, why the previous studies from the same lab show CEPsh is critical for nerve ring formation.

Response:

We thank the reviewer for pointing out the need for clarification regarding our ablation experiments and the causality of glial and neuronal defects in *unc-23* mutants. CEPsh glia ablation in *unc-23(lof)* is the key experiment to address causality for this correlation, given that *unc-23* acts in epithelia precluding to test cell-autonomy in CEPsh glia. We agree with the reviewer that it is important to understand what happens in the circuit after ablation of CEPsh glia in L1 animals (performed in this manuscript) and how that compares to CEPsh glia ablation in the embryo (performed in Rapti et al, 2017). Embryonic ablation (by EGL-1/BH3 expression in glia from embryo ventral closure onwards) causes defects in pathfinding of axons concurrently growing in the nerve ring. Conversely, the genetic ablation of CEPsh glia at L1 stage kills the CEPsh after embryogenesis and causes no defects in axon pathfinding in the nerve ring (which is largely formed by the L1). The phenotypes of postembryonic ablation were initially reported in Yoshimura et al, 2008 (PMID: 18508862). This control (wild-type animals with CEPsh glia ablated in L1) is included in our submitted manuscript (Fig. 3M, condition wild-type + ablation) and shows that sensory neurons and axons have no significant defect after CEPsh glia ablation at L1. Thus, absence of embryonic but not postembryonic CEPsh glia is critical for brain neuron architecture, in wild-type animals. We hope that this information clarifies our findings. The ectopic CEPsh glial membranes in *unc-23(lof)* affects neuronal maintenance, and CEPsh glia ablation in *unc-23(lof)* L1 animals suppresses the majority (but not the entirety) of neuronal defects. Since this suppression is partial and not complete, we agree with the reviewer, we cannot fully exclude that *unc-23* may also affect neuronal maintenance, at some extent independently of CEPsh glia. To clarify these points and acknowledge the partial suppression of neuronal defects, we edited our revised manuscript to read as follows:

In Results, in page 7:

“CEPsh glial cells drive brain neuropil assembly and axon pathfinding during embryonic morphogenesis. Embryonic ablation (by EGL-1/BH3 expression in glia, from embryo ventral closure onwards) causes defects in pathfinding of pioneer and follower axons concurrently growing in the nerve ring²¹. Conversely, the genetic ablation of CEPsh glia at L1 stage onwards was reported to cause no defects in axon pathfinding in the nerve ring^{21,45}. Yet, postembryonic CEPsh glia envelop axons and synapses throughout larval and adult stages and affect positioning of neuronal synapses^{20,21,35}. Thus, we assessed if age-progressive CEPsh glia integrity affects circuit architecture.”

[...]

“The observed defects in sensory and CEP neurons correlate significantly with CEPsh glial defects (Fig. 3I). To assess the causality of such correlation we ablated postembryonic CEPsh glia, using an established system of caspase expression in L1 animals which does not cause axonal defects in wild-type animals (Fig. 3M, Methods). Postembryonic ablation of CEPsh glia strongly suppresses the majority of the observed amphid neuron defects in *unc-23* mutants. Since this suppression is partial, we cannot fully exclude that *unc-23* may also affect neurons independently of CEPsh glia. Yet, these ablation experiments demonstrate that the observed neuronal defects in *unc-23* mutants largely depend on CEPsh glia disruption.”

In Discussion, in page 13:

“Postembryonic killing of CEPsh glia that does not cause defective axon morphologies in wild type animals, can alleviate the neuronal circuit defects in *unc-23* mutants that present ectopic glial membranes (Fig. 3) ³⁰.”

In Figure 3, Legend:

“L-M. Amphid neuron (AmN) axon defects (I) and CEP defects (L) in *unc-23* mutants correlate with glia defects. Axon defects in *unc-23* mutants are partially suppressed upon postembryonic CEPsh glia ablation (in L1 onwards), that by itself does not cause axon defects (M).”

3) Based on that high temperature and starvation partially suppressed glial “integrity” phenotypes in unc-23 mutants, the authors concluded that “that conserved functions of the UNC-23/BAG2- HSP-1/Hsp70/Hsc70 - DNJ-13/ DNAJB1 complex integrate tissue responses to temperature and nutritional changes to safeguard the the lifelong integrity of glial cells and circuit architecture”. This is a such strong statement, but the data only show the correlation between high temperature or starvation and unc-23(lf) phenotypes. No direct evidence to support this conclusion. At least some critical control experiments need to be done before reaching such conclusion. One possibility is that high temperature or starvation could directly affect ECM, and this ECM change may have nothing to do with unc-23. It is also possible that temperature and starvation just affect the nature of unc-23 mutations used in this study.

Response:

We value the reviewer’s point and following their suggestion we performed experiments to assess the effect of high temperature or starvation on the ECM. We analyzed the localization of Perlecan (UNC-52::mNG KI) in wild-type animals and in *unc-23(lof)*, after heat-shock or after starvation of these animals at the first larval stage. Based on these results, the Perlecan localization in L2 or in L4 animals does not present defects in wild type animals subject to this temperature increase or starvation compared to wild-type in normal temperature and food conditions. Thus, the ECM in wild-type animals present more robust localization, less sensitive to these temperature or caloric changes (Figure S7D). These results are in line with our previous observations that temperature increase or starvation do not cause visible CEPsh glial cell defects in wild type animals (Figure 4E-F). Moreover, our added experiments show that the partial defects of Perlecan localization observed in L2 *unc-23(lof)* mutant animals are enhanced upon temperature increase and suppressed upon starvation, compared to the untreated mutants (Figure S7D-E). Thus, higher temperature accelerates the progressive ECM defects of *unc-*

23(*lof*) mutants, while caloric restriction delays them. Finally, several additional experiments of the revised manuscript support our model. We observe that starvation suppresses ECM and CEPsh glial cells defects in *unc-23* mutants, not because of a change in animal's locomotion (Figure S7F). Finally, the *unc-23* mutant defects slowed down by starvation still aggravate in later stages, suggesting there is no permanent change of the mutated protein by the external condition changes (Figure S7D-E). Finally, we attribute this role to conserved functions of BAG2 in the HSP complex since we showed that BAG2 functions with HSP-1 in this context, in line with previous reports and that human BAG2 cDNA rescues these mutant defects (Fig. 4B-C). Indeed, we observe that *hsp-1* knock-down (suppressing CEPsh glial cells defects in *unc-23* mutants) also suppresses the ECM defects of Perlecan/UNC-52 and Collagen/ EMB-9 in *unc-23(lof)* mutants (FigureS7A-C). Altogether, these results are in line with our model that epithelial HSP proteostasis has conserved functions in the HSP-complex to integrate responses to temperature and nutritional changes in order to safeguard CEPsh glial cell integrity, and ECM integrity as demonstrated in these suggested experiments). Yet, comprehending the exact mechanism of this integration by UNC-23/Bag2 requires future investigation. Thus, and in line with the reviewer's view, we edited our text in our revised manuscript to remove strong statements. The added experiments and results are presented in the revised Figure S7 and discussed in the text as follows:

In Results, in pages 8-10:

“Conversely to high temperature, starvation partially suppresses glial integrity phenotypes in *unc-23* mutants (Fig. 4F). Importantly, these temperature and nutritional changes do not affect the integrity of CEPsh glial architecture in wild-type animals (Fig. 4E-F). Thus, temperature and caloric changes affect the roles of UNC-23/BAG2 in CEPsh glial cell and circuit maintenance, through mechanisms that we further study below.”

[...]

“Since temperature and starvation changes affect CEPsh glial defects in *unc-23* mutants, we assessed if they also affect ECM responses in relation to HSP proteostasis (in *unc-23* mutants) or they affect glia and ECM also in wild-type animals. Interestingly, Perlecan localization in wild type animals of L2 or L4 stage does not present defects subject to temperature increase or starvation, compared to wild-type animals in normal conditions (Fig. S7D). In line with this observation, temperature increase or starvation do not cause CEPsh glial cell defects in wild type animals as shown above (Fig. 4E-F). Conversely, the mild defects of Perlecan localization observed in L2 *unc-23* mutants are enhanced upon temperature increase and suppressed upon starvation, compared to untreated mutants (Fig. S7D-E). Importantly, no altered locomotion of L2 or L4 animals (as quantified by body bends per minute) is detected in starved versus well-fed *unc-23* mutants (Fig. S7E) suggesting that starvation affects UNC-23/BAG2 functions in glia integrity through mechanisms other than altered locomotion. Thus, the progressive ECM defects of *unc-23* mutants, are dependent in HSP-1/Hsc70 function, accelerated by temperature increases and delayed by caloric restriction, while such changes do not affect the robust ECM localization in wild-type animals. Altogether, ECM components act upstream of CEPsh glial cell integrity, accumulate abnormally in *unc-23* mutants because of abnormal HSP function, are affected by temperature and nutritional changes, they correlate with and are followed by defective CEPsh glia, and altering their level suppresses CEPsh glial defects in *unc-23* mutants.”

In Discussion, in page 14:

“Overall, our findings suggest that UNC-23/BAG2, through conserved functions and together with the HSP-1/Hsp70/Hsc70, may integrate tissue responses to temperature, caloric restriction and biomechanics to safeguard the robust integrity of ECM, glial cell and circuit components

upon age progression. The exact mechanistic underpinnings of this integration remain to be further elucidated.”

In Figure S7, legend:

“ECM defects in *unc-23* mutants are affected by HSP-1, heat and caloric restriction but do not result from locomotion differences.

(A-C) *hsp-1* knock-down by RNAi or *hsp-1(ra807)* mutation suppresses the localization defects of Perlecan/UNC-52 (A-B) and Collagen/ EMB-9 in *unc-23* mutant L4 animals (C). This is quantified by the % of animals in populations of different genotypes/conditions that present the Perlecan/UNC-52 defects in *unc-23* mutants, as observed per Fig. 5 (A), or by quantifications of relative sum intensity of Perlecan/UNC-52 content in ROI₂/ROI₁ (posterior/anterior of glia,) as initially quantified in Fig. 5B-C (B). The Collagen/ EMB-9 localization is examined using quantification of its relative sum intensity in the ROI₂/ROI₁ (posterior/anterior of glia), as initially quantified in Fig. 5B-C (B). (n ≥ 100 animals in A, C. n ≥ 6 animals in B, t-test, p values as in Fig.

(D-E) The mild defects of Perlecan/UNC-52 observed in L2 animals of *unc-23* mutants are enhanced by subjecting animals to acute temperature-increase and suppressed by subjecting animals to starvation (at the L1 stage). Defects of Perlecan/UNC-52 in L4 animals of *unc-23* mutants are similar in all conditions. No such Perlecan/UNC-52 defects are observed in wild-type animals subject to these conditions of temperature increase or starvation. (n ≥ 100 animals for D, n ≥ 10 animals for E).

(F) L2 or L4 animals of *unc-23* mutants present similar locomotion (as quantified by body bends per minute) in normal conditions or after starvation at L1 stage. (n ≥ 10 animals, bends measured for 1 minute/animal), t-test, p values as in Fig. 2.”

4) *The author show unc-23 mutants doesn't affect epithelial polarity or integrity, and this is opposite to what has been described in a 2015 paper from Donald Moerman lab. Why? Any discussion?*

Response:

We thank the reviewer for remarking on this point. The cited paper (Rahmani et al, 2015), describes that “epithelia hemidesmosomes and intermediate filament networks are disrupted when muscle cells detach in *unc-23(lop)* hermaphrodites”. Based on their Methods, this was detected by antibody staining (possibly after centrifugation and collagenase treatment) in fixed adult animals (unclear of which day), while which quantifications were performed and in which epithelial cells is unclear to us. Our experiments were performed in intact worms of L4 larval stage, using live-fluorescent imaging with no prior treatments and quantifications were performed in the epithelia neighbouring the CEPsh glia cells.

Our manuscript describes three characteristics of epithelia specifically neighbouring the CEPsh glia of L4 animals: the continuity of epithelial membrane, the intensity of an apical-surface marker and the integrity of epithelial cell junctions assessed by a DLG-1 enrichment neighbouring the CEPsh glia cells. We detected no significant epithelial differences in these aspects and concluded that “no changes in epithelial polarity and integrity may account for impaired CEPsh glial integrity” in *unc-23(lop)*. We cannot exclude that epithelial cells have other defects that we did not assess, for example in areas far from the CEPsh glia and/or later stages that we did not investigate. Meanwhile, it is not clear whether the defects detected by Rahmani et al, 2015 affect areas proximal to the CEPsh glia nor whether they are established

before the glial defects. Overall, the two studies report distinct (not opposite) effects, analysing different animal stages and different epithelial properties, by different methodologies. To discuss these considerations, we have edited our Results sections to read as follows:

In Results, in page 9:

“Previous immunostaining studies in fixed adult animals proposed defects in epithelial hemidesmosomes in *unc-23* mutants (Rahmani et al, 2015). While we cannot exclude *unc-23* mutant defects in other epithelial properties, domains far from CEPsh glial membrane sheaths or other stages not assessed here, we detect no significant changes in epithelial polarity and cell junctions’ integrity neighbouring the CEPsh glia that may account for impaired CEPsh glial integrity in *unc-23* mutants.”

5) Based on “Perlecan and COL4A5 present abnormal localization in unc-23 mutants, are depleted from areas neighboring normal CEPsh glia positions, and accumulate neighboring ectopic glial territories (Fig. 5B-E, S6)” and “Perlecan matrix abnormalities are significantly correlated with glial defects in unc-23 mutant L4 larvae but precede these defects in L2 mutants (Fig. 5F-G)”, the author concluded that “ECM components function downstream of UNC-23/BAG2 for glia integrity”. However, it is also possible that those mis-location of Perlecan and COL4A5 are the results rather than the cause of abnormal morphology of CEPsh glia.

Response:

We thank the reviewer for their observation which points out the need for clarification. We agree that the observations cited by the reviewer above are in line with our model but do not prove it. But several other findings point out that ECM defects are the cause (and not result) of the CEPsh glial abnormalities. If ECM abnormalities are the cause (upstream) of CEPsh glial abnormalities, we would expect them to appear earlier (or consequently, but not after) the glial defects and to influence glia morphology in *unc-23* mutants but not to be themselves influenced by CEPsh glia changes. Importantly, we show that “Perlecan matrix abnormalities [...] precede [glial] defects in L2 mutants (revised Fig. 5E-F)”. Specifically, approximately 20% of L2 mutant animals present abnormalities of the Perlecan matrix while only 3% of L2 mutants have CEPsh glia defects (revised Fig. 5E-F). Importantly, we also show that “reducing the levels of ECM components UNC-52/Perlecan, EMB-9/COL4A5, and laminin LAM-1/LAMB1-2, partially suppresses the CEPsh glial defects of *unc-23* mutants (Fig. 5A). *unc-52* genetic mutation also suppresses CEPsh glial phenotypes in *unc-23* mutants. This suggests that ECM components act downstream of UNC-23 functions and upstream the CEPsh glial phenotypes.” Both observations place ECM upstream the glial defects. Nevertheless, to challenge our conclusion in view of the reviewer’s point, we performed an additional experiment. We examined the Perlecan matrix in *unc-23(lop)* mutants with post-embryonically ablated CEPsh glia. If ECM defects are the results of abnormal CEPsh glia morphology, glial ablation could cause suppression of the ECM defects, similar to the suppression of neuronal defects. However, our results do not support this hypothesis. To the contrary, Perlecan matrix defects in *unc-23(lop)* mutants are not suppressed upon CEPsh glia ablation. We present the new experiment in Figure 5G of the revised manuscript and discussed these considerations in the text as follows:

In Results, in page 9:

“We found that reducing the levels of ECM components UNC-52/Perlecan, EMB-9/COL4A5, and laminin LAM-1/LAMB1-2, partially suppresses the CEPsh glial defects of *unc-23* mutants (Fig. 5A). *unc-52* genetic mutation also suppresses CEPsh glial phenotypes in *unc-23* mutants. This suggests that ECM components act downstream of UNC-23 functions and upstream the CEPsh glial phenotypes. [...] Importantly, Perlecan matrix abnormalities are significantly correlated with glial defects in *unc-23* mutant L4 larvae but these ECM defects precede CEPsh glial defects in L2 mutants (Fig. 5F-G). Importantly, upon ablation of CEPsh glial cells at L1 stage onwards in *unc-23* mutants the defects of Perlecan/UNC-52 localization are not suppressed but remain similar to the *unc-23* mutants (Fig. 5G). This is contrary to the suppression of neuronal defects, upon CEPsh glia ablation in L1 *unc-23* mutants. Moreover, *hsp-1* knock-down (suppressing CEPsh glial cells defects) also suppresses the ECM defects of Perlecan/UNC-52 and Collagen/ EMB-9 localization in *unc-23* mutants (Fig. S7A-C). Based on all these observations combined, the ECM components appear to act downstream UNC-23/HSP proteostasis and upstream of CEPsh glia for maintenance of their glial integrity.”

In Figure 5, legend:

[...]

“**A.** Knocking-down epithelial-expressed ECM components Perlecan, Collagen and Laminin by RNAi (KD) or the *unc-52(e444)* genetic mutation of Perlecan partially suppresses CEPsh glia defects in *unc-23* mutants. $n \geq 3$ independent experiments, ≥ 150 animals total, Chi-square test. Arrows, glia defects. Arrowhead, Perlecan defects. **B-C.** Perlecan/UNC-52 (green) has disrupted localization and accumulates posteriorly of the pharynx in *unc-23* mutants, compared to wild-type L4 animals. $n=20$ animals per genotype. In *unc-23* mutants, mean intensity of UNC-52::KI is significantly decreased anterior to the 2nd pharyngeal bulb, and the ratio of mean and sum intensity between posterior and anterior regions is increased, compared to wild-type animals. ROI₁, ROI₂ correspond in C are defined in B. **D.** Perlecan mean intensity at distance posterior to 1st pharyngeal bulb, represented by average (grey or orange) \pm standard deviation (light grey or yellow), 20 animals/ genotype. **E-F.** Perlecan defects correlate to defects of CEPsh glia in L4 (**E**) but precede these defects in L2 animals (**F**), $n \geq 100$ animals. **G.** CEPsh glia ablation (at L1 stage onwards) does not significantly suppress the defects of Perlecan UNC-52 in *unc-23* mutants. **(A-G).** Scale bars, animal axes, reporters, error bars, p-value, ns, as in Fig. 2. RNAi results in Table S1. (See Fig. S5).”

6) In Fig. 6 the author claimed that “Similarly, mutant CEPsh glia defects are partially suppressed upon bacteria-mediated knock-down of *DLG-1*, *AJM-1*, *LET-413*, *LET-805*, *MUA-3* in the nervous system, using mutations that enhance RNAi-sensitivity in the nervous system (Fig. 6F, Methods). These components are largely absent from postembryonic neurons, thus this knock-down affects their glial transcripts. “. Does that mean it has to be CEPsh glia? could it be other glia ?

Response:

The reviewer raises an important point here for which we need to clarifications and present control experiments we previously performed. Our studies indicate that in wild-type animals, CEPsh glial cells are refractory to RNAi, similarly to neurons, while RNAi is more efficient in most other glial cells. Specifically, RNAi against GFP in wild-type animals confers low GFP knock-down in CEPsh glia, in only $8,18 \pm 5,99\%$ of animals examined. Conversely, GFP RNAi

confers partial or complete knock-down in AMsh (amphid sheath) glia, in $84,42 \pm 11,25\%$ of animals examined, and in other head anterior glia in $76,40 \pm 18,46\%$ of animals examined. We present these control experiments in the revised Table S1. Thus, other head glial cells besides the CEPsh are rather sensitive to RNAi in wild-type animals. Therefore, RNAi against junctional components in non-sensitized backgrounds is expected to knock-down their transcripts in non-nervous system cells and in head glial cells other than the CEPsh. Consequently, RNAi in RNAi-sensitized backgrounds largely reveals additional effects of gene knock-down in CEPsh glial cells. In line with this, similar suppression effects are caused by bacteria-fed RNAi and knock-down of DLG-1 specifically in postembryonic CEPsh glia, using CEPsh glia-specific promoter for vector -mediated antisense expression. A comprehensive presentation of these results is found in Table S1 of the revised manuscript, and discussed in the Results as follows:

In Results, in page 11:

“Similarly, mutant CEPsh glia defects are partially suppressed upon bacteria-mediated knock-down of DLG-1 AJM-1, LET-413, LET-805, MUA-3 in the nervous system, using mutations that enhance RNAi-sensitivity in the nervous system, including in CEPsh glia (Fig. 6F, Table S1, Methods). Transcripts of these components are largely absent from scRNAseq of postembryonic neurons (Cao et al, 2017), thus this knock-down affects their glial transcripts. Conversely, CEPsh glia defects are not suppressed by knocking-down epithelial junctional components, in *unc-23* mutants with RNAi-insensitive nervous system (Fig. 6F, Methods). Such RNAi in non-sensitized backgrounds affects transcripts in non-nervous system cells as well as in head glial cells other than CEPsh (amphid sheath, AMsh and other head anterior sheath and socket glia) (Table S1). Thus, the differential RNAi effect in non-sensitized and sensitized backgrounds highlights effects in CEPsh glia, but not in epithelia, muscle or other head glial cells. Overall, this suggests that glia junctional components affect CEPsh glial integrity in *unc-23* mutants.”

Reviewer #2 (Remarks to the Author):

Reviewer #2 Comments:

*In this review, I addressed only the Brillouin measurements presented in the manuscript -- the other parts of this work are outside my expertise. The authors used Brillouin microscopy to measure the high-frequency elasticity and viscosity of mutant and wild-type *C. elegans* worms. The main conclusion from these measurements was: "different animal biomechanics precede and underly age dependent disruption of ECM and glia".*

Response:

We thank the Reviewers for their report and valuable comments, which helped to clarify important aspects of the Brillouin measurements, and thus to enhance the quality of our presentation. In short, we address all reviewer's points in detail, add new experiments (refractive index measurements) and analysis (Brillouin loss tangent), and re-quantify/re-plot previous data (absolute Brillouin shift/ linewidth) as per the reviewer's suggestions. In addition, we provide explanations in answers to the reviewer, and all-encompassing edits in the text of the revised manuscript to include the discussed considerations, clarify our conclusions and highlight the relevance and importance of the Brillouin microscopy in our study. We trust that using Brillouin provides us with a unique approach to gain an understanding of tissue mechanical aspects relation to glial cell and circuit maintenance and we hope that reviewer agrees with our assessment.

To correctly image the storage and the loss moduli using Brillouin microscopy, the density and refractive index of the medium should be known or measured. I could not find information on these parameters, therefore I am not convinced about the reliability and interpretation of the data presented. This is a very critical point.

Response:

We agree that additional information on the density and refractive index of the sample are required to deduce quantitative mechanical properties from Brillouin measurements, especially with respect to the storage and loss moduli. As the reviewer might appreciate, measuring these parameters at the required high spatial resolution in biology and especially in scattering samples such as *C. elegans*, is currently very challenging. However, we note that while both the refractive index and the density may vary with conditions, their ratio ρ/n^2 does not vary significantly in biological materials (see references PMID: 33091376, PMID: 26436482, PMID: 30122291, now also cited in our revised manuscript). Therefore, the value of Brillouin frequency shift and linewidth are often reported in the field as direct indicators of the mechanical properties. Likewise, in our manuscript we also do not claim absolute values of storage or loss moduli, but rather focus on the relative changes in terms of shift and linewidth between the mutant and wild-type, which, from a biological perspective, are the important readouts of the Brillouin measurements. Nevertheless, we realize that more information on this topic is warranted. Thus, we now have removed any ambiguous statements towards the interpretation and added clarifying sentences throughout our revised manuscript. These changes are presented below this answer.

Furthermore, to ascertain that the observed changes in Brillouin shift do in fact have a mechanical origin, we have performed the following additional measurements and analysis:

First, we performed *in-vivo* measurements of the refractive index (RI) in wild-type and *unc-23* mutant animals in the stage and ROIs of interest, and using a commercial holotomography microscope (NanoLive). We find no significant differences on the average RI of the tissue regions of interest between wild-type and mutant animals. Specifically, mean RI values (average \pm SD) are 1,3727 \pm 0,0041 and 1,3730 \pm 0,0051 (p value= 0.8386) in wild-type and mutant animals, respectively (as presented now in the revised Fig. S8).

Second, to further address the technical limitation that we currently do not have the capability to measure both Brillouin shift/width and RI simultaneously for each point in the sample, we have further computed the so-called Brillouin loss tangent, defined as $\tan(\varphi) = M''/M' = \Gamma/\nu$ (BLT, as per references PMID: 31012711 and PMID: 34580426, also cited in our revised manuscript). By its definition, the BLT does not depend on the sample refractive index and density and thus provides a simple approach to determine whether mechanical properties are the main contributor to observed changes in the Brillouin spectrum and their spatial maps. Here, we also find that BLT is significantly lower in *unc-23* mutants compared to wild-type animals of the same life stage (see Fig. S8). Specifically, BLT absolute values (average \pm SD) are 0,1227 \pm 0,0033 and 0,1163 \pm 0,0056 (p value < 0.0001) in wild-type and mutant animals, respectively. The relative values (to wild-type) are 1,0000 \pm 0,0268 and 0,9476 \pm 0,0455 in wild-type and mutant animals, respectively. This significant decrease of BLT in the mutants, further corroborates our reported results of lower Brillouin shift and width in mutant animals.

Altogether, we hope that the additional data, new analysis and rephrasing (below) convince the reviewer about the reliability and interpretation of our data. We have edited our revised manuscript to read as follows:

In Results, page 12:

“Therefore, Brillouin microscopy provides unprecedented capability to measure mechanical properties *in-vivo*. In particular, Brillouin microscopy measures both the so-called high-frequency elasticity as well as viscosity of biological samples, through the frequency shift and spectral linewidth of the Brillouin scattered photons interacting with the sample. While we note that for quantitative measurements of the longitudinal modulus the ratio between the refractive index (n) and density (ρ) of the probed tissue need to be known, prior work has shown that this ratio ρ/n^2 does not vary significantly in biological materials⁶⁵⁻⁶⁷, therefore relative changes in the spectral shift and linewidth are indicative of mechanical changes. Utilizing a confocal Brillouin microscope⁶⁸, we probed anesthetized *unc-23* mutant and wild-type animals, on L2 stage preceding the establishment of integrity defects. Interestingly, areas neighbouring CEPsh glia present decreased tissue viscosity (Brillouin linewidth) and elasticity in *unc-23* mutants compared to wild-type animals (Fig. 7E-F, Fig. S8), as measured by the Brillouin spectral linewidth and shift, respectively⁵⁷. To further ascertain the mechanical origin of the observed changes in Brillouin shift and linewidth, we performed refractive index measurements (Methods, Fig. S8). Moreover, we computed the so-called Brillouin loss tangent (BLT), defined as $\tan(\varphi) = \Gamma/\nu$, which by definition does not depend on the sample refractive index and density^{67,70}. Importantly, based on our findings, the average refractive index does not differ significantly between L2 *unc-23* mutant and wild-type animals, while the relative Brillouin loss tangent was significantly lower in mutants, (Methods, Fig. 7F, S8), in line with the lower Brillouin width and shift in the mutants. Therefore, mutant animals lacking functional BAG2/HSP-cochaperone, present already in early stages a reduced mean viscosity and

elasticity in tissues neighboring the CEPsh glial cells. These altered material properties and animal biomechanics precede and underly the later, age-progressive disruption of CEPsh glia architecture, as measured by optical microscopy.”

In Figure 7, legend:

“**E-F.** L2 tissue viscosity and elasticity (as measured by the Brillouin spectral linewidth and shift) is decreased in *unc-23* mutants compared to wild-type. Values relative to the wild-type are presented here (for absolute values, see Fig. S8). (E) Dotted areas are ROIs selected flanking the 2nd pharyngeal bulb dorsally (ROI1) and ventrally (ROI2). These tissue areas neighbor the normal position of glia membranes of the dorsal CEPsh (ROI1) and the ventral CEPsh (ROI2), and present ectopic glial membranes in *unc-23* mutant larvae animals (see Methods for more information). Mean Brillouin linewidth and shift of the ROI1 and ROI2 from distinct animals of the same genotype are grouped together (Brillouin values of ROI1 and ROI2 are not significantly different in wild-type animals).”

In Figure S8, legend:

“Brillouin width, shift, loss tangent and refractive index measurements of *unc-23* mutants and wild-type animals.

A. Representative Brillouin microscopy images showing the spectral width and thus viscous contrast (corrected for image deconvolution, see Methods) in L2 individuals of *unc-23* mutant and wild-type animals. Depicted ROIs are in regions neighboring CEPsh glia localization (see Methods). Scale bars, 10 μ m. White dotted rectangles in (A), ROIs in (B, C) as in Fig. 7.

B-C. Quantification of the tissue ROIs show decreased Brillouin width/ viscous contrast (“viscosity”) and decreased Brillouin shift/ elastic contrast (“elasticity”) of *unc-23* mutants compared to wild-type animals. Absolute values (average \pm SD) of mean Brillouin width (for viscosity) are 0,9712 \pm 0,0274 and 0,9122 \pm 0,0443 after deconvolution, in wild-type and *unc-23* mutant animals, respectively. Relative values (average \pm SD) of mean Brillouin width are 1,0000 \pm 0,0283 and 0,9393 \pm 0,0456 for wild-type and mutant animals, respectively, presented in Fig. 7F. The values of *unc-23* mutant and wild-type animals are significantly different, p value < 0.0001. Absolute values (average \pm SD) of mean Brillouin shift are 7,9106 \pm 0,0548 and 7,8380 \pm 0,0756 for wild-type and *unc-23* mutant animals, respectively. Relative values (average \pm SD) of mean Brillouin shift are 1,0000 \pm 0,0069 and 0,9908 \pm 0,0096 for wild-type and mutant animals, respectively, presented in Fig. 7F. These values are significantly different with p value<0.005.

D-E. Absolute and relative values of Brillouin loss tangent represent viscoelasticity, independently of the sample’s refractive index. The Brillouin loss tangent is significantly lower in *unc-23* mutants compared to wild-type animals (p value < 0.0001). Absolute values of mean loss tangent (Average \pm SD) are 0,1227 \pm 0,0033 and 0,1163 \pm 0,0056 for wild-type and mutant animals, respectively. Relative values of mean loss tangent (Average \pm SD) are 1,0000 \pm 0,0268 and 0,9476 \pm 0,0455 for wild-type and mutant animals, respectively.

F. Absolute values of refractive index in tissue ROIs of *unc-23* mutants and wild-type animals as assessed by commercial holotomography are not significantly different (p value= 0.8386). (B-F) n \geq 20 ROIs, 10 animals. unpaired t-test.”

In Material & Methods, in page 19:

“Brillouin imaging is performed using a Brillouin microscope previously described in ⁷¹. Briefly, this consists of a commercial Zeiss body (Axiovert 200M) coupled with a home-built spectrometer based on a 2-VIPA configuration, which provides a measured spectral resolution

of 520MHz and a precision of 22 and 56 MHz for Brillouin shift and linewidth measurements, respectively.”

[...]

“Measurements of refractive index of the animals were acquired by performing label-free 3D Holo-Tomographic Live Cell Imaging of worms, using the Nanolive 3D Cell Explorer Fluo. L2 animals were mounted on slides using Pluronic F127 36% w/v + 1 mM tetramisole solution or embedded in 2% agarose+1 mM tetramisole solution. Brightfield images were acquired using an objective 60× NA 0.8 and fluorescent images were acquired using a FITC filter of the Nanolive Module LED DAPI-FITC-TRITC/Cy5 4X B 000.”

[...]

“In Fig. 7E-F and S8, ROIs are selected flanking the 2nd pharyngeal bulb and the normal CEPsh glia membranes dorsally (ROI1) and ventrally (ROI2). These areas present ectopic glial membranes in late mutant larvae. Mean Brillouin linewidth and shift of the ROI1 and ROI2 from distinct animals of the same genotype are grouped together (Brillouin values of ROI1 and ROI2 are not significantly different in wild-type animals).”

Another point is how the shift and linewidth of the Brillouin spectra were measured given the low spectral resolution of the VIPA spectrometer and the large spectral broadening at NA=1. What are the accuracy and precision of the obtained shift and linewidth values? based on what criteria the ROIs were selected?

Response:

The Reviewer raises an important point here. Indeed, a large aperture influences the Brillouin shift, due to the high variability in possible scattering angles, and it is well known that NA~1 results in a corresponding downshift of ~100MHz, as previously measured and reported for the Brillouin microscope used in this work (c.f. Bevilacqua et al, Biomed. Opt. Exp. 2019, Fig. 2D). However, since we’re mostly focusing on relative changes between mutants and wild-type animals, this does not affect the main conclusions presented in our work.

We also value the comment of the Reviewer about our accuracy and precision, and we have added this information to the Methods section of the revised manuscript. In short, the precision is 22 (56) MHz for shift (width) for the parameters (100ms exposure time, 4mW optical power on the sample) used for our measurements. Here we note that the spectral resolution of our VIPA spectrometer is 520MHz (c.f. Bevilacqua et al, Biomed. Opt. Exp. 2019) and that the uncertainty in our shift/width measurements are about ~1/350 and ~1/17 of a typical shift/linewidth, respectively, measured inside the *C. elegans* tissue. Regarding the accuracy, to convert from the spectrometer camera readout (‘pixel’) to GHz, we use distilled water as a reference and assume a linear VIPA dispersion. The accuracy on the value of the Brillouin shift of water is mainly determined by the dependence of the Brillouin shift on temperature. The latter is ~12MHz/C around room temperature [ref. <https://doi.org/10.3390/s8095820>] and, assuming a variation of ±1C of the temperature of the lab, it corresponds to an accuracy of ±25MHz. The error due to linear assumption is less than 0.1% (7.5MHz) [ref. A2 of https://archiv.ub.uni-heidelberg.de/volltextserver/32423/1/Phd_Thesis_Carlo_Bevilacqua.pdf]. Therefore, the maximum systematic error on the Brillouin shift is ~40MHz.

We included this information in the Materials & Methods of our revised manuscript as follows:

“Brillouin imaging is performed using a Brillouin microscope previously described in ⁷¹. Briefly, this consists of a commercial Zeiss body (Axiovert 200M) coupled with a home-built spectrometer based on a 2-VIPA configuration, which provides a measured spectral resolution of 520MHz and a precision of 22 and 56 MHz for Brillouin shift and linewidth measurements, respectively, for our measurement parameters (100ms exposure time, 4mW optical power on the sample).”

The ROIs were selected based on the following criteria as provided in the sections of Material & Methods and in the legend of Figure 7:

“ROIs are selected flanking the 2nd pharyngeal bulb dorsally (ROI1) and ventrally (ROI2). These tissue areas neighbor the normal position of glia membranes of the dorsal CEPsh (ROI1) and the ventral CEPsh (ROI2), and present ectopic glial membranes in *unc-23* mutant larvae animals. The pharynx depression allows for defining the same area in different animals. ROIs selected exclude the pharynx to avoid signal variability due to bacteria content. ROIs are selected with the function “Polygon selections” in Fiji and mean values for the shift/linewidth are computed to appropriately represent the mechanics on a tissue level. From the raw Brillouin shift and FWHM linewidth values we compute the Brillouin elastic and viscous contrast analogous to Antonacci et al, 2020 ⁶⁹.”

Additional points are (1) A quantified analysis of the variation of the mean viscosity and stiffness in Figure S7 and its meaning/implication to the biological questions investigated is missing,

Response:

We thank the Reviewer for suggesting the addition of this analysis in the manuscript. We have better highlighted and represented our results in the revised manuscript. We present relative values of Brillouin mean viscosity or the longitudinal elasticity in Fig. 7 and absolute mean values of these measurements in the revised Fig. S8 (previously S7). All of these values (relative mean viscosity, relative mean longitudinal elasticity, absolute mean viscosity, absolute mean elasticity) are significantly different between the mutant and wild-type animal populations (p value<0.005). We also present the Brillouin loss tangent values which are also significantly different between mutant and wild-type animal populations (p value<0.005). This has major implications to the biological questions we investigate here. We present the data and values in the Fig. S8 and we discuss these points in the Results section to read as follows:

In Results, in page 12:

“Therefore, mutant animals lacking functional BAG2/HSP-cochaperone, present already in early stages a reduced mean viscosity and elasticity in the tissues neighboring the CEPsh glial cells. These altered material properties and animal biomechanics precede and underly the later, age-progressive disruption of the ECM and CEPsh glia architecture, as measured by optical microscopy. Interestingly, our combined results show that animals with reduced viscosity

suffer disruption of ECM and tissue integrity that can be partially rescued by subjecting them to substrate of lower viscosity or reduced forces in their environment.”

In Figure S8, legend:

[...]

B-C. Quantification of the tissue ROIs show decreased Brillouin width/ viscous contrast (“viscosity”) and decreased Brillouin shift/ elastic contrast (“elasticity”) of *unc-23* mutants compared to wild-type animals. Absolute values (average±SD) of mean Brillouin width (for viscosity) are $0,9712\pm0,0274$ and $0,9122\pm0,0443$ after deconvolution, in wild-type and *unc-23* mutant animals, respectively. Relative values (average±SD) of mean Brillouin width are $1,0000\pm0,0283$ and $0,9393\pm0,0456$ for wild-type and mutant animals, respectively, presented in Fig. 7F. The values of *unc-23* mutant and wild-type animals are significantly different, p value < 0.0001. Absolute values (average±SD) of mean Brillouin shift are $7,9106\pm0,0548$ and $7,8380\pm0,0756$ for wild-type and *unc-23* mutant animals, respectively. Relative values (average±SD) of mean Brillouin shift are $1,0000\pm0,0069$ and $0,9908\pm0,0096$ for wild-type and mutant animals, respectively, presented in Fig. 7F. These values are significantly different with p value<0.005.

D-E. Absolute and relative values of Brillouin loss tangent represent viscoelasticity, independently of the sample’s refractive index. The Brillouin loss tangent is significantly lower in *unc-23* mutants compared to wild-type animals (p value < 0.0001). Absolute values of mean loss tangent (Average±SD) are $0,1227\pm0,0033$ and $0,1163\pm0,0056$ for wild-type and mutant animals, respectively. Relative values of mean loss tangent (Average±SD) are $1,0000\pm0,0268$ and $0,9476\pm0,0455$ for wild-type and mutant animals, respectively.

F. Absolute values of refractive index in tissue ROIs of *unc-23* mutants and wild-type animals as assessed by commercial holotomography are not significantly different (p value= 0.8386). (B-F) n ≥20 ROIs, 10 animals. unpaired t-test.”

(2) Why only mean values were analyzed in Figures 7F and S7? could any useful spatial information be obtained from the Brillouin images?,

Response:

We thank the Reviewer for this suggestion. However, we note that spatially highly resolved measurements are not very informative, as biologically we are interested in the overall tissue region neighboring the CEPsh glial membranes. Therefore, a mean value of each ROI makes the most sense for our study. To further clarify this point, we have added the following sentence to the Materials & Methods section of the main manuscript:

“We quantify the mean value of Brillouin linewidth/ shift of the selected ROIs, since we are interested in the overall tissue areas neighboring the wild-type CEPsh glial membranes, and which present ectopic glial membranes in the later stages of mutants. The morphology of ectopic membranes varies in different animals but always populates the selected ROIs. Consequently, spatially higher resolved measurements or analysis within these areas would not be more informative for this study, and the mean measurements of ROIs are presented.”

(3) The added value and importance of the Brillouin measurements to this work is very unclear, particularly the vague conclusion arising from these measurements.

Response:

We thank the Reviewer for this feedback which made us realize that the Brillouin measurements and their results warrant further discussions and elaboration. Above and in our revised manuscript, we have performed new experiments, re-analyzed or re-plotted our original data and we provided clarifications for the above considerations, throughout the revised manuscript. We hope that all these revisions address the reviewer's concern for any previous unintentional vagueness. Moreover, we have edited the sections of Introduction and Discussion in the revised manuscript to further highlight the value, but also the limitations, of Brillouin microscopy for the non-invasive assessment of tissue mechanics and the value of our specific findings.

In short, the mechanical properties of cells and tissues play an intricate role in determining biological function and can serve as indicators for various diseases, yet standard techniques used in the field of mechanobiology to measure them rely on either invasive procedures or fail to probe the mechanical properties with sufficient high spatial resolution and/or beneath surfaces. Here, Brillouin microscopy offers an unprecedented capability to infer these properties in a high-resolution, 3D and non-invasive manner. Therefore, we regard it as the only method to probe tissue mechanics *in vivo*, and their role in affecting neuroglia tissue morphogenesis and integrity throughout animal growth. Even in the absence of a quantitative measurement of the storage/loss modulus, we believe that Brillouin microscopy adds highly valuable information by being able to measure relative visco-elastic, mechanical changes between animals of different genetic background (i.e. wild-type vs. mutant).

We realize we have devoted too little room in the original manuscript to motivate and highlight the advantages of Brillouin microscopy and the reasoning of our conclusions, and have thus added more information throughout the manuscript to read as follows:

In Introduction, in page 4:

“How glial cells face and withstand mechanical stress and other environmental changes to sustain their architecture throughout the animal's life is poorly understood. Furthermore, current techniques to measure mechanical properties such as elasticity and viscosity non-invasively, *in-vivo*, are lacking (Prevedel et al, 2019). Combining such biophysical measurements with studies of genetics and animal physiology would allow for a unique understanding of the interactions between genes, cells and tissue material properties in physiological and pathological conditions of complex tissues, such as the nervous system.”

[...]

Therefore -by combining advanced genetics, live quantitative imaging, biophysical measurements and manipulation of genes, cells, and their environment- we uncover a key interplay of ECM proteostasis, glial-cell junctions and environmental factors of temperature and material properties, that ensures maintaining of CEPsh glia integrity.

In Discussion, in page 15:

“Our findings suggest that following decreased tissue viscosity and elasticity (as measured through Brillouin microscopy), ECM disruption and glia integrity (as measured through optical microscopy) are more sensitive to high-viscosity than low-viscosity environment (Fig.7), which is in line with effects of forces and biomechanics on other tissue deformations⁷⁷. Tissue mechanical properties play an intricate role in determining biological function and can serve as indicators for pathology, yet standard mechanobiology techniques used to measure them rely on either invasive procedures or fail to probe the mechanical properties with sufficient high spatial resolution and/or beneath surfaces. Brillouin microscopy offers an unprecedented

capability to infer these properties non-invasively, in a 3D and in high-resolution. This work, among few others, attests that Brillouin microscopy can highlight early material properties differences between normal and pathological tissues *in vivo*. It allows us to probe the roles of tissue mechanics in affecting neuroglia tissue architecture and integrity *in vivo*. Our findings suggest that decreased tissue viscoelasticity affects ECM and neuroglia integrity and that manipulating the ECM and (maybe more surprisingly) manipulating the animal's external environment affects tissue integrity. This understanding may prove valuable in the further future, in prediction and design of approaches for disease recovery. In this direction, a precise dissection of the biochemical, cellular and material properties of tissues in relation to their environment is necessary. This work highlights an integration of tissue responses to temperature and biomechanical changes. Such multi-disciplinary understanding of biological properties across-scale may later contribute to future applications in disease diagnosis or to future environmental research for a comprehension of biomechanics in animal responses to environmental change.”

This is a very critical point, (4) Why relative values were presented? would statistical tests on the relative and absolute values yield same results/conclusions?

Response:

We appreciate the Reviewer's comment and suggestion on the presentation, analysis and statistical significance of the absolute vs. relative values. Importantly, both the relative and absolute values yield same results/ conclusions, and with equal significance, using the same statistical tests, as what we previously presented (see legend of Fig. S8, presented below).

In short, when comparing either relative and absolute values of Brillouin mean shift/width, the *unc-23* mutant animals present decreased mean shift and width compared to wild-type (p value<0,005). We would like to highlight again that from a biological perspective, and keeping in mind the limitations regarding absolute moduli measurements as remarked above, the relative mechanical measurements between mutant and wild-type worms are indeed the most informative for this study. Absolute values of Brillouin linewidth/ shift would be less informative to a general audience due to the yet limited studies of Brillouin microscopy in living animal tissue and unknowns on reference absolute values in *C. elegans* larvae in particular.

Nevertheless, following the reviewer's remark, we present in the revised manuscript both relative values (Fig. 7) and absolute values of the measurements of Brillouin mean linewidth/ shift (Fig. S8), as presented below We also provide clarifications in the Materials & Methods. The associated edits in our revised manuscript read as follows:

In Figure 7, legend:

“**E-F.** L2 tissue viscosity and elasticity (as measured by the Brillouin spectral linewidth and shift) is decreased in *unc-23* mutants compared to wild-type. Values relative to the wild-type are presented here (for absolute values, see Fig. S8). (E) Dotted areas are ROIs selected flanking the 2nd pharyngeal bulb dorsally (ROI1) and ventrally (ROI2). These tissue areas neighbor the normal position of glia membranes of the dorsal CEPsh (ROI1) and the ventral CEPsh (ROI2), and present ectopic glial membranes in *unc-23* mutant larvae animals (see Methods for more information). Mean Brillouin linewidth and shift of the ROI1 and ROI2 from distinct animals of

the same genotype are grouped together (Brillouin values of ROI1 and ROI2 are not significantly different in wild-type animals).”

In Figure S8, legend:

[...]

B-C. Quantification of the tissue ROIs show decreased Brillouin width/ viscous contrast (“viscosity”) and decreased Brillouin shift/ elastic contrast (“elasticity”) of *unc-23* mutants compared to wild-type animals. Absolute values (average \pm SD) of mean Brillouin width (for viscosity) are 0,9712 \pm 0,0274 and 0,9122 \pm 0,0443 after deconvolution, in wild-type and *unc-23* mutant animals, respectively. Relative values (average \pm SD) of mean Brillouin width are 1,0000 \pm 0,0283 and 0,9393 \pm 0,0456 for wild-type and mutant animals, respectively, presented in Fig. 7F. The values of *unc-23* mutant and wild-type animals are significantly different, p value < 0.0001. Absolute values (average \pm SD) of mean Brillouin shift are 7,9106 \pm 0,0548 and 7,8380 \pm 0,0756 for wild-type and *unc-23* mutant animals, respectively. Relative values (average \pm SD) of mean Brillouin shift are 1,0000 \pm 0,0069 and 0,9908 \pm 0,0096 for wild-type and mutant animals, respectively, presented in Fig. 7F. These values are significantly different with p value<0.005.

D-E. Absolute and relative values of Brillouin loss tangent represent viscoelasticity, independently of the sample’s refractive index. The Brillouin loss tangent is significantly lower in *unc-23* mutants compared to wild-type animals (p value < 0.0001). Absolute values of mean loss tangent (Average \pm SD) are 0,1227 \pm 0,0033 and 0,1163 \pm 0,0056 for wild-type and mutant animals, respectively. Relative values of mean loss tangent (Average \pm SD) are 1,0000 \pm 0,0268 and 0,9476 \pm 0,0455 for wild-type and mutant animals, respectively.

F. Absolute values of refractive index in tissue ROIs of *unc-23* mutants and wild-type animals as assessed by commercial holotomography are not significantly different (p value= 0.8386). (B-F) n \geq 20 ROIs, 10 animals. unpaired t-test.”

Materials & Methods:

We quantify the mean value of Brillouin linewidth/ shift of the selected ROIs, since we are interested in the overall tissue region neighboring the CEPsh glial membranes. We note that spatially higher resolved measurements or analysis within these areas would not be more informative for this study, thus the mean measurements of ROIs are presented. Values are presented as absolute or as relative to the wild-type average. From a biological perspective, and taking into account the limitations of absolute moduli measurements, the relative mechanical measurements between mutant and wild-type worms are the most informative for this study.

Reviewer #3 (Remarks to the Author):

NCOMMS-23-00958

Lifelong interplay of HSP-proteostasis, biomechanics and ECM-cell junctions ensures C. elegans astroglial architecture

The authors described the morphological size changes of CEPsh glia during C.elegans larval development and adult stage (day of adulthood not specified). At L4, they assessed CEPsh glia and their colocalization to hypodermis, muscles, and neurons. In a screen, they identified unc-23(arg5) mutant showing longer but less volume of CEPsh glia. They confirmed that the arg25 mutation is causal to the phenotype by rescuing with wild-type genomic DNA of unc-23. Furthermore, another mutant unc-23(e25) phenocopied unc-23(arg5). Expression of wild-type UNC-23 in the hypodermis also rescued unc-23(e25), implicating that unc-23 acts from the hypodermis. Moreover, using the orthologous mouse BAG2 sequences also rescued the unc-23(e25) mutant phenotypes, suggesting conservation in function. They showed the unc-23 mutants have synaptic and axon defects (unclear at which developmental stage. Maybe L4). Knockdown of hsp-1 or dnj-13 fully rescues the morphological defects in CEPsh glial. However, starvation or heat stress had minor effects on CEPsh glial phenotypes. They found partial and weak rescue with knockdown of members of the basement membrane (BM) extracellular matrix (ECM) (i.e., RNAi against unc-52, emb-9, and lam-1). Strikingly, mutation in unc-52(e444) almost completely rescued unc-23(e25) glial phenotypes. Disrupted UNC-52mNG, specifically at the ventral side, correlated with disrupted glia at L2 (not or minor at L4). Partial rescue of unc-23(e25) by knocking down genes in body wall muscles and integrin signaling. Interfering with collagen synthesis (pharmacological inhibitor or KD of collagens) led to a partial or weak rescue of unc-23(e25) glia phenotypes, whereas inhibiting collagen synthesis or crosslinking aggravated the unc-23(e25) glia phenotypes.

Overall, the study is well designed, and the experiments are well conducted. I love the beautiful videos. However, conceptionally, it is known that unc-23/BAG2 binds the ATPase domain of the chaperon HSP-1/Hsc70 and thereby regulates their activity, and in C. elegans, the maintenance of muscle attachments (PMID: 26435886). The novelty here is that interfering with unc-23, hsp-1, BM or muscles affects the morphology of glial cells. A major limitation is that besides one lifespan, none of the experiments have been performed during ageing.

Response:

We thank the reviewer for their compliment for our project design, experiments and timelapse videos. Indeed, it is previously shown that UNC-23 binds the ATPase domain of HSP-1 and affects muscle integrity, as we also cite. Our manuscript demonstrates roles of epithelial HSP proteostasis in glial cell integrity through an interplay with ECM, cell junctions and integration of mechanical and temperature challenges. We also uncover how maintenance of glial cell integrity affects neuronal architecture, which remains understudied. Below we replied in detail to the reviewer's comments and experimental suggestions. We performed the suggested experiments and in the revised manuscript we present the results, and provide clarifications and discussions to address all the reviewer's comments. Overall, these new insights to strengthen our manuscript, and we thank the reviewer for their suggestions. Finally, some sections subtitles are also shortened according to communicated formatting instructions.

Major

1) It seems like hyperbole to make any lifelong (title, abstract) or ageing claims (text) when all experiments have been performed during larval development L2-L4 and some mention the adult stage (without specifying which day of adulthood). In general, age-related phenotypes start after reproduction. This is usually post day 8 of adulthood (whereby L4 is day 0 of adulthood per definition). Either show age-related, ageing, or lifelong (that means until death) implications of *unc-23* or remove these statements from the manuscript.

Response:

We value the reviewer's remarks and after careful consideration we agree that the term *lifelong* is not the most appropriate as it indicates performing functional experiments throughout animals' life. Indeed, some experiments are performed in day-5 adults but not until the day all animals die. As the reviewer may know, such experimentation would be very challenging if not impossible using available methodologies, taking into account that the majority of *unc-23* mutant animals die as ~day-10 adults. Thus, we edited our manuscript (including title and abstract) to omit the term *lifelong* or *aging* when referring to the described defects/mechanisms. Instead, we use the terms *age-dependent / age-progressive and maintenance*, that more accurately describe our findings as we present below. Our manuscript provides evidence that the studied ECM and tissue alterations depend on the animal's age (L2/L3/L4/adult stage), they aggravate progressively as the animal's age progresses throughout these stages. We would like to note here that *age-dependent/-progressive* refers to defects that aggravate during the progression of animal's age (throughout larval/ adult stages), and differs from the terms *aging-dependent/-progressive* which would refer to specific effects during *aging*. (Interestingly, other studies discover that gene disruption in early animal life sometimes affects early-manifesting, age- progressive neurodegeneration, separately from development; i.e. studies from the Hilliard lab show that *lin-14* gene acts in L1 *C. elegans* larvae to protect from age-progressive neurodegeneration, and *lin-14* mutants have mild neuronal defects first appearing at L1 and aggravating at L4 animals, prior to animal's aging (PMID: 28930688). Such distinctions appear valid also beyond *C. elegans*; some patients with Alzheimer's disease manifest early onset symptoms from the age of late 30's-40's, thus during reproductive age. These symptoms first appear before aging but are *age-dependent, age-progressive*.) We hope that these clarifications and discussion as well as the manuscript edits below address the reviewer's concern. The revised text reads as follows:

Title:

"Age-progressive interplay of HSP-proteostasis, ECM-cell junctions and biomechanics ensures *C. elegans* astroglial architecture"

Abstract:

"Safeguarding the remarkably-complex architectures of neurons and glia ensures maintenance of age-dependent integrity of functional circuits."

[...]

"Overall, we present a finely-regulated interplay of proteostasis-ECM and cell junctions with conserved components that ensures age-progressive robustness of glial architecture."

In the Introduction:

"Maintaining tissue integrity throughout age progression and across diverse environments is an outcome of the evolution of multicellular life."

[...]

"Investigating how proteostasis, ECM, and cell interactions integrate responses to external factors, to sustain tissue integrity against pathology, is key to comprehending tissue biology."

[...]

“Therefore, it is crucial to understand the molecular mechanisms underlying maintenance of neuronal and glial cell architecture to ensure proper circuit structure and function as age progresses.”

[...]

“Therefore, we uncover a key interplay of ECM proteostasis, glial-cell junctions and environmental factors of temperature and material properties, that ensures maintaining of CEPsh glia integrity.”

In the Results:

“Age-progressive glia disruption affects circuit architecture maintenance and relates to abnormal aging “

[...]

“Thus, we assessed if CEPsh glia integrity affects circuit architecture after development.”

[...]

“Altogether, our results highlight that age-progressive integrity of CEPsh glia is key to maintaining circuit architecture including axon positioning and synapse density. Disruption of glial cell and circuit integrity is associated with decreased animal lifespan. The mechanistic underpinnings of these effects remain to be further elucidated.”

[...]

“Overall, our findings suggest that UNC-23/BAG2, through conserved functions and together with the HSP-1/Hsp70/Hsc70, may integrate tissue responses to temperature, caloric restriction and biomechanics to safeguard the robust integrity of ECM, glial cell and circuit components upon age progression. The exact mechanistic underpinnings of this integration remain to be further elucidated.”

[...]

“Accordingly, the epithelial UNC-23/ BAG2, may safeguard age-dependent glia integrity by regulating proteostasis required for epithelia-glia communication.”

[...]

“Thus, altering the material properties of the exoskeleton safeguards the age-dependent robustness of ECM and glia architecture.”

[...]

“Disordered glial integrity results in disruption of brain architecture after its establishment, with aberrant localization of neurons, axons, synapses and is associated with abnormal aging.”

[...]

In Discussion, in page 13:

“Uncovering mechanisms of age-dependent astroglial integrity can shed light on understudied aspects of neuropathology.”

[...]

“Thus, while absence of postembryonic CEPsh glia may not compromise the circuit structure that requires embryonic CEPsh glia to assemble, the age-progressive ectopic positioning of CEPsh glia disrupts circuit maintenance (Fig. 3, S3). [...] Therefore, maintenance of proper CEPsh astroglia architecture is key for circuit maintenance. Similarly, ectopic activity of astrocytes, responding to injuries or age progression, contributes to neuropathology²². In mammals, gliodegeneration can precede aging and neurodegeneration and contributes to cognitive impairment^{7,58}. Uncovering mechanisms of age-dependent astroglial integrity can shed light on understudied aspects of neuropathology.”

[...]

“How CEPsh glia physically associate with epithelia and the ECM and how this affects age-progressive integrity was previously unclear.”

[...]

“This interplay involves glial-cell junctions connecting glia to tissue and ECM neighbors, to maintain age-progressive integrity of their architecture and support circuit properties.”

Furthermore, the effects of UNC-23 were not assessed post-development to make any ageing claims (either us RNAi starting from day 1 of adulthood or other techniques to block the function of unc-23).

Response:

We value the reviewer’s remark and we performed the suggested experiments. In short, *unc-23* RNAi in adults results in significant and age-progressive CEPsh glial cell defects. We report the results below this answer. Meanwhile, we wish to provide an important clarification here and in our revised manuscript, regarding development versus post-development processes/maintenance of the studied cells. CEPsh glia of *unc-23* mutants have no defects in L1 animals and only mild defects in L2 that worsen progressively later. These are maintenance abnormalities and not developmental defects, since the development of CEPsh glial architecture is largely established before L2. CEPsh glia grow embryonic non-branching membrane processes that then get ramified, and this architecture is patterned and established in embryogenesis and L1 stage (Rapti et al, 2017; this manuscript). Key characteristics of the mature CEPsh glial cell architecture, as they are detectable in the adult neuropil by connectomics (Sulston et al, 1983; Cook et al, 2020; Witvliet et al, 2021) and our own work (Rapti et al, 2017; this manuscript), are largely established before the L2 stage: 1. ramifications of their early, embryonic non-branching glial membranes (Fig. 1A), 2. membrane tiling of CEPsh to generate a circumferential glial cell ring enveloping the axonal nerve ring (Fig. 1A), 3. the pairing of CEPsh glial membrane processes with specific axons to establish tripartite-synapse ensheathment (by comparative connectomics, in Witvliet et al, 2021). After the L2 stage, the CEPsh glia enlarge their surface to maintain this established architecture (Fig. 1B, 2E). Similarly, most nerve ring axons form their architecture before L2 and then enlarge relative to animal size, through a maintenance process driven by dedicated mechanisms that differ from those driving their development DOI: 10.1016/S0070-2153(09)01206-X). Disruption of these maintenance mechanism causes neuronal defects starting from L2 and worsening with age progression. (Benard & Hobert, 2006; Chen et al., 2001; Pocock et al., 2008; Sasakura et al., 2005; Zallen et al., 1999). This is in line with the maintenance (not developmental) age-progressive defects in CEPsh glial integrity, observed here. We discuss these considerations in our revised manuscript (below this answer).

Nevertheless, we agree that it is interesting to know if *unc-23* is needed in adulthood for CEPsh glia integrity. Following the reviewer’s suggestion, we subjected animals at the end of L4 stage to RNAi by bacteria-feeding to knock-down *unc-23* only in adulthood and quantitatively analyzed the CEPsh glia in day-1, day-3 and day-5 adults. Vector-induced RNAi was not performed due to the lack of regulatory elements inducing expression only in adult epithelia while the *hsp-16* heat-inducible promoter was not used for time-controlled expression as it interferes with heat responses. Yet, bacteria-induced RNAi knock-down in adults causes significant, progressive defects in CEPsh glial cell architecture (revised Fig. S3). These defects are milder than those caused by *unc-23* genetic mutation, which is not surprising since RNAi has partial effect, *unc-23* wild-type transcripts are produced before RNAi application, and the

protein appears to be stable (only mild defects are detected in CEPsh glia of L4 animals after RNAi in L1 stage onwards, in Fig. 4C and revised Fig. S3). RNAi of *unc-23* in adults results in age-progressive CEPsh glial cell defects, throughout day-1,-3,-5 adulthood. Interestingly, wild-type animals present some CEPsh glial cell defects also in 5-day adults onwards, yet *unc-23* RNAi knock-down causes significantly increased CEPsh glial defects. We also identified glial and neuronal defects in *unc-23* mutants that become significant after larval stages and progressively aggravate in adults. Specifically, the anterior processes and tips of CEPsh glia in *unc-23* mutant adults show extensive defects, of misplacement and losing attachment to neighboring tissues. These defects are not detected in L4 animals (previous submission, Fig. S1B) but are detected only in adult stages (current analysis, Fig. S1B-D). They become significant in day-1 adults and age-progressively deteriorates throughout day-1, day-3 and day-5 adults (Figures S1B-D). Secondly, AIY synaptic vesicle markers apposing to CEPsh glia are sparser but not decreased in L4 mutant animals (Figures 3), but synaptic vesicle content also presents overall decrease in day-5 adult animals of *unc-23* mutants compared to wild-type (Figures S3). These results combined suggest that UNC-23 function is likely required also after larval stages for proper maintenance of glial architecture. We thank the reviewer for the experimental suggestion that strengthens our manuscript. These considerations and results are presented in the revised Fig. S1, S3 and discussed in the text as follows:

In Introduction, in page 3:

“In *C. elegans*, the brain-like neuropil composed of ~180 axons is patterned in embryogenesis and completes its formation before the 2nd larval stage (L2). Afterwards, the neuropil’s components enlarge relative to animal size to maintain the underlying architecture. The neurons maintain their structural and functional characteristics, correct positions and connections within their ensembles. This neuronal maintenance requires active, dedicated mechanisms with roles that differ from those acting in development: Ig-containing secreted or transmembrane proteins, L1 CAM and FGFR orthologs sustain positioning of neurons, axons²⁴, and synapses²⁵, while hemidesmosomes and spectrin protect axons from degeneration²³. Disruption of these maintenance mechanisms causes neuronal defects sometimes starting from L2 stage and worsening as the animal matures^{24,26–28}. Less is known about mechanisms maintaining glial cell architecture.”

In Results, in page 5:

“CEPsh glia in the embryo grow thin, non-branching processes, that then transform in astrocyte-like, ramified membranes, associating with axons and synapses^{21,37}. This transformation is largely established by the L1 stage (Fig. 1A). Key characteristics of the mature architecture of CEPsh glia, detectable in the mature neuropil by connectomics^{37,40,41} are largely established before the L2 larval stage: 1. the ramifications of the early non-branching membranes of CEPsh glia (Fig. 1A), 2. the membrane tiling of the 4 CEPsh glia to generate a circumferential ring of membranes enveloping the brain axonal *nerve ring* (Fig. 1A), 3. the pairing of CEPsh glial membrane processes with axons to establish tripartite-synapse ensheathment (detected by connectomics throughout the animal’s stages⁴¹. Beyond the L2 stage, the CEPsh glial cells expand their surface in relation to animal growth, to maintain this already-established architecture (Fig. 1B, 2E).”

[...]

“Importantly, the defects of *unc-23* mutants are suppressed by RNAi of *hsp-1*, *dnj-13*, or *hsf-1* (after L1), demonstrating that postembryonic Bag2/HSP acts after development of CEPsh glia for maintenance of their architecture.”

[...]

“Interestingly, certain defects of CEPsh glial cell architecture appear only in *unc-23* mutant adults. Specifically, *unc-23* mutant adults -but not larvae- present defective anterior membrane tips, that lose attachment with neighboring tissues or show altered morphology, in an age-progressive manner (Fig. S1). Also, neuronal synaptic vesicles associated with CEPsh glial cells present different defects in L4 and adult *unc-23* mutants (see below and Fig S3). To investigate if UNC-23/BAG2 function is also required in adulthood for age-progressive maintenance of glial architecture, we subjected wild-type animals of late L4 stage to *unc-23* RNAi and assessed the CEPsh glial cell architecture in different days of adulthood (Fig. S3A). Interestingly, *unc-23* RNAi in wild-type adult animals results in defects in the architecture of ramified CEPsh glial membranes, that worsen progressively from day-1 to day-5 adults. These defects are similar to defects in L4 animals resulting from RNAi in L1 animals, but milder than the defects in the genetic mutant suggesting that the wild-type UNC-23 accumulating before RNAi application is a rather stable protein (Fig. S3). These findings are in line with UNC-23/BAG2 acting for age-progressive maintenance of glial cell integrity. Overall, UNC-23/BAG2 is required for maintaining CEPsh glia integrity and its functional loss results in age-progressive disruption of glial architecture with less compact, filamentous, discontinuous membranes, occupying ectopic territories (Movies S4-S5).”

[...]

“5-day adults of *unc-23* mutant with defective CEPsh glia present decreased density, like in L4 animals, as well as decreased overall intensity of presynaptic vesicle markers, which is not observed in L4s (Fig. S3B-C). The latter is a hallmark of deteriorating neuronal healthspan, as described in previous studies ⁴¹. Thus, CEPsh glial defects are succeeded by additional neuronal defects in *unc-23* mutant adults, compared to larval animals.”

In Discussion, in page 13:

“Dedicated mechanisms are employed for postembryonic maintenance of glia, distinct to those acting for their development.”

[...]

“The UNC-23/BAG2 acts for age-progressive disruption of CEPsh glial cell architecture while it appears largely dispensable for glial cell and circuit development, established by the L2 larval stages (introduction, Fig. 2-3).”

[...]

“Interestingly, mutants with functional disruption of UNC-23/BAG2 and deteriorating CEPsh glial integrity present misplaced synaptic vesicle densities in larvae accompanied in later adulthood by decrease in overall content of synaptic vesicle densities, previously described as a sign of deteriorating neuronal healthspan ⁴¹. Therefore, maintenance of proper CEPsh astroglia architecture is key for circuit maintenance.”

In Figure S1, Legend

Anterior processes of CEPsh glia are defective in *unc-23* mutant adult but not in L4 stage.

A-B. Each of the 4 CEPsh glia (green) has an anterior process and tip arriving to the sensory endings of the *C. elegans* nose. These are shown here in lateral view of entire CEPsh glia (Ai, Bi, Biv,) or lateral view of their anterior glia processes (Aii, Bvii, Bx) or zoom view of their tips (Aiii, Bviii, Bxi) or in cross-sectional view of their tips (Aiii, Biii, Bvi, Bix, Bxii). Anterior processes and tips of CEPsh glia are largely not defective in *unc-23* mutants compared to wild-type animals in L4 stage (Bi-vi), but they present defects of mispositioning and fragmentation in adult stages (Bvii-Bxi). **C-D.** These CEPsh glia defects appear in significant number of

adults *unc-23* mutant animals (C) and they are age-progressive (D). Scale bars, 10 μ m. Animal axes as in Figure 1.

In Figure S3, legend:

“UNC-23/BAG2 acts in larval and adult stages and affects synapse aging and animal lifespan.

(A). Subjecting wild-type adults in *unc-23* RNAi results in significant defects in CEPsh glial cell architecture, that worsen progressively throughout days 1 and 5 of adulthood.

(B-C). In five-day old adults, AIY presynaptic vesicles (by RAB-3::mCherry marker, magenta) have decreased mean and sum intensity in *unc-23* mutants (Ai-iii, **B**) compared to wild-type animals (Biv-vi, **C**). *, gut autofluorescence. n=12 animals per genotype. unpaired t-test. Animal axes, P values as in Fig. 3. [...]”

1.1) p5. lines138-177. The title of this paragraph says “age-dependent” but all experiments looking at glial morphology were done during larval development. Particularly, lines153-156, they claim “unc-23 mutants present age-dependent defects” but look at larval stages 1 and 2 (L1 and L2) in development.

Response:

As discussed above, with terms *age-dependent/-progressive* we refer to phenotypes/defects that depend on/ aggravate throughout the animal’s age (animal’s stage, not animal’s aging), thus differing from terms *aging-dependent/-progressive*. In the specific part referred above, we assessed quantitatively in all L1-L4 larval stages the glial cell architecture of *unc-23* animals (Fig. 2F-J). We identified that CEPsh glial cells have no defects in L1 stage and only mild ones in L2 stage, which worsen progressively with age throughout the next stages (Figure 2f,2J). In the revised manuscript, following the reviewer’s suggestion, we also show that upon *unc-23* knock-down in adults CEPsh glial defects are age-progressive throughout 1-5 adulthood days (see above answer, revised Figure S3 and edits in several manuscript sections). To more accurately describe our findings also in the aforementioned title, we edited this title using the term age-progressive instead of the term age-dependent, as follows:

“*unc-23/BAG2* mutants suffer age-progressive disruption of CEPsh glia”

1.2) p6. Lines 180-202. Again, the title of the paragraph says “age-processing” and “aging” but base this on day 5 adult animals and on a single lifespan assay. For age-dependent, one would assume a phenotype that is normal in young adults (day 1) that becomes progressively worsen the following days (preferably starting post reproduction (day 8)). Thus, to make such a claim, the authors would need to alter the unc-23 function starting at adulthood and have at least 3 measuring time points to say “age-processing”.

Response:

Following the reviewer’s remark and experimental suggestion, we performed these experiments for adult alteration of *unc-23* and phenotypic assessments in 3 timepoints, as

presented in response to the previous comments above and in the revised Figure S3 (see above comments and below this answer). In summary, the RNAi knock-down of *unc-23* in adulthood results in progressive defects in CEPsh glial cells, throughout adulthood days 1-5. We present these results in the revised Figure S3 and in the text as discussed above. As presented above, we omit the term “aging” but use the term *age-progressing*, which differs from the term *aging-progressing*. We edited the revised manuscript to read as follows:

Subtitle, in page 7:

“Age-progressive glia disruption affects circuit architecture maintenance and relates to abnormal aging”

Regarding the lifespan assays, we address the point in detail below. We value the reviewer’s remark, we repeated and replotted more than one lifespan assays and discuss them below.

In Figure S3, legend:

“UNC-23/BAG2 acts in larval and adult stages and affects synapse aging and animal lifespan.

(A). Subjecting wild-type adults in *unc-23* RNAi results in significant defects in CEPsh glial cell architecture, that worsen progressively throughout days 1 and 5 of adulthood.

(B-C). In five-day old adults, AIY presynaptic vesicles (by RAB-3::mCherry marker, magenta) have decreased mean and sum intensity in *unc-23* mutants (Ai-iii, **B**) compared to wild-type animals (Biv-vi, **C**). *, gut autofluorescence. n=12 animals per genotype. unpaired t-test. Animal axes, P values as in Fig. 3. **D.** *unc-23* mutants show decreased lifespan compared to wild-type animals, as assessed by the median lifespan (50% survival) and the maximum lifespan (5% survival) of the two populations. Lifespan assays were performed with or without the use of FuDR. n ≥ 100 animals per genotype. Results were analyzed and compared with the Kaplan–Meier method and the Wilcoxon rank sum test. p value < 0,05 in lifespan experiments, with or without the use of FuDR.”

Also, line 198. They show that unc-23(e25) mutants have a shortened lifespan compared to the wild type, and this is based on a single lifespan assay that was not even carried out until the end (i.e., 40% of the wild type were still living). Furthermore, this is not decreased “longevity”.. this is decreased lifespan. Per definition, longevity means long-lived.

Response:

We value the reviewer’s remarks and corrections. We regret plotting the lifespan assays only until the last survival day of the *unc-23* mutants. We performed lifespan assay and performed additional lifespan experiments of wild-type and mutant animals, without and with the use of FuDR and replotted them appropriately (revised Figure S3, Methods). In both conditions, the lifespan of *unc-23* mutants is significantly shorter than the one of wild-type animals; also based on the median lifespan and the “maximum lifespan” (5% survival) of the two populations. The FuDR use appears to increase the lifespan of both populations, similarly to previous reports. Importantly we recognize that the mechanistic underpinnings of the lifespan changes in *unc-23* mutants remain to be elucidated. We thank the reviewer for their experimental suggestion that enhances the quality of our manuscript. We present these results in the revised Figure S3 and discuss these results and considerations in the text as follows:

In Results, in page 7:

“Moreover *unc-23* mutants have shorter lifespan compared to wild-type animals, as indicated by their median and maximum lifespan (Fig. S3D). [...] Altogether, our results highlight that age-progressive integrity of CEPsh glia is key to maintain circuit architecture including axon positioning and synapse density, and its disruption is associated with decreased animal lifespan. The mechanistic underpinnings of these effects remain to be further elucidated.”

In Figure S3, legend:

“UNC-23/BAG2 acts in larval and adult stages and affects synapse aging and animal lifespan.

(A). Subjecting wild-type adults in *unc-23* RNAi results in significant defects in CEPsh glial cell architecture, that worsen progressively throughout days 1 and 5 of adulthood.

(B-C). In five-day old adults, AIY presynaptic vesicles (by RAB-3::mCherry marker, magenta) have decreased mean and sum intensity in *unc-23* mutants (Ai-iii, **B**) compared to wild-type animals (Biv-vi, **C**). *, gut autofluorescence. n=12 animals per genotype. unpaired t-test. Animal axes, P values as in Fig. 3. **D.** *unc-23* mutants show decreased lifespan compared to wild-type animals, as assessed by the median lifespan (50% survival) and the maximum lifespan (5% survival) of the two populations. Lifespan assays were performed with or without the use of FuDR. n ≥ 100 animals per genotype. Results were analyzed and compared with the Kaplan–Meier method and the Wilcoxon rank sum test. p value < 0,05 in lifespan experiments, with or without the use of FuDR.”

1.3) p.10 lines 380-382. claim “age-dependent shearing and nuclei fragmentation (fig 2)” but experiments were done during development (fig 2K-L: during L2 nuclei assessments).

Response:

We phenotypically analyzed nuclei of the CEPsh glial cells in L2 and L4 animals and found that defects aggravate from the L2 to the L4 stage. We repeated these experiments to increase the number of assessed animals and the results support our previous conclusion. We present these results in Figure 2, S2. As discussed above, CEPsh glial cell defects established in L2 onwards affect maintenance -and not development- of their architecture, since the key characteristics of their mature architecture (membrane ramifications, tiling, and synapse ensheathment) are already established before L2. *Age* refers to stages of animal’s life (not to *aging*) and our results establish progressive defects. Taking into account the above and the reviewer’s point we edited this text to read as follows:

In Results, in page 13:

“Upon functional loss of Hsp70-co-chaperone BAG2, the CEPsh glia suffer disrupted architecture with features of lost integrity, age-progressive shearing and possible nuclei fragmentation (Fig. 2, S2) [...]”

2) As mentioned above, it is known that *unc-23/BAG2* binds the ATPase domain of the chaperon HSP-1/Hsc70 and thereby regulates their activity, and in *C. elegans*, the maintenance of muscle attachments (PMID: 26435886). Some mechanistic or genetic insights

would provide some novelty. For instance, it is remarkable how well *hsp-1* rescues the glial phenotypes. In an *unc-23* mutant with *hsp-1(RNAi)*, is the UNC-52::mNG ventral basement membrane rescued? Thus, is the phenotype due to improper UNC-52 localization? Or is it due to shear forces from the muscle that disrupts the glial morphology? The latter could explain why *unc-52(e444)* mutants rescue glial defects of *unc-23* mutants. There should be at least some discussion pointing to novel insights.

Response:

We appreciate the reviewer's suggestions to provide more insights into the function of HSP proteostasis for tissue integrity. Following their suggestion, we knocked-down *hsp-1* by RNAi, in *unc-23* mutant animals with endogenous labeling of Perlecan (UNC-52::mNG). In line with the reviewer's idea, the defects of UNC-52::mNG present in *unc-23* mutants are rescued upon *hsp-1* knock-down (revised Fig. S7A-C). This is in line with our model that the UNC-23/HSP affects UNC-52 localization upstream of the glial effects in *unc-23* mutants. This hypothesis is also supported by other results in our manuscript (Perlecan defects arise prior to glial defects and knock-down of ECM components suppress the glia defects, as presented in Fig 5A, 5F-G). Instead, the alternative hypothesis that shear forces from the muscle disrupt the glial morphology is not supported by our results; while animal growth in liquid cultures partially suppresses Perlecan defects, glial defects and muscle defects, the suppression of glial defects correlate significantly with Perlecan defects but with the defects of the muscle (Fig. 7B-C). These results and considerations are presented in the revised Figure S7 and discussed in the text in our revised manuscript, as follows:

In Results, in page 10:

“Moreover, *hsp-1* knock-down (suppressing CEPsh glial cells defects) also suppresses the ECM defects of Perlecan/UNC-52 and Collagen/ EMB-9 localization in *unc-23* mutants (Fig. S7A-C). Based on all these observations combined, the ECM components appear to act downstream of UNC-23/HSP proteostasis and upstream of CEPsh glia for maintenance of their glial integrity.”

[...]

“Conversely, concurrent recovery of muscle integrity does not correlate with recovered glial integrity, suggesting that the defects in CEPsh glial integrity are not caused directly by the muscle defects in *unc-23* mutants (Fig.7 C).”

In Discussion, in page 14:

“The abnormal localization of ECM matrix components is dependent on the imbalance of HSP70 upon disruption of UNC-23/Bag2 function, and is alleviated by HSP-1 decrease in *unc-23* mutants (Fig. S6).”

In Figure S7, legend:

“(A-C) *hsp-1* knock-down by RNAi or *hsp-1(ra807)* mutation suppresses the localization defects of Perlecan/UNC-52 (A-B) and Collagen/ EMB-9 in *unc-23* mutant L4 animals (C). This is quantified by the % of animals in populations of different genotypes/conditions that present the Perlecan/UNC-52 defects in *unc-23* mutants, as observed per Fig. 5 (A), or by quantifications of relative sum intensity of Perlecan/UNC-52 content in ROI₂/ROI₁ (posterior/anterior of glia,) as initially quantified in Fig. 5B-C (B). The Collagen/ EMB-9 localization is examined using quantification of its relative sum intensity in the ROI₂/ROI₁ (posterior/anterior of glia), as initially quantified in Fig. 5B-C (B). (n ≥ 100 animals in A,C. n ≥ 6 animals in B, t-test, p values as in Fig.

(D-E) The mild defects of Perlecan/UNC-52 observed in L2 animals of *unc-23* mutants are enhanced by subjecting animals to acute temperature-increase and suppressed by subjecting animals to starvation (at the L1 stage). Defects of Perlecan/UNC-52 in L4 animals of *unc-23* mutants are similar in all conditions. No such Perlecan/UNC-52 defects are observed in wild-type animals subject to these conditions of temperature increase or starvation. (n ≥ 100 animals for D, n ≥ 10 animals for E).

(F) L2 or L4 animals of *unc-23* mutants present similar locomotion (as quantified by body bends per minute) in normal conditions or after starvation at L1 stage. (n ≥ 10 animals, bends measured for 1 minute/animal), t-test, p values as in Fig. 2.”

Minor

1) p.10 lines 363-364 and Fig. 7H. The suggested orthologues of these cuticular collagens are wrong. See <https://doi.org/10.1016/j.mbplus.2018.11.001> for collagen classes.

Response:

We thank the reviewer for pointing this out and for highlighting this study that we now cite in our manuscript. We opt to omit the orthologue names from the figure due to space limitations, as certain genes present different predicted orthologues depending on software predictions. We refer to the above citation for the predicted orthologues in the revised manuscript as follows:

In Results, in page 12:

“Conversely, the disrupted CEPsh glia architecture in *unc-23* mutants is partially suppressed when inhibiting collagen biosynthesis chemically (using the above established drugs) or by knocking-down collagen-modifying enzymes ADT-2/ADAMTS, Peroxidase-like MLT-7/PXDNL, pro-lysyl oxidase protease NAS-37 or specific collagens including DPY-3, DPY-5, COL-41, COL-43, COL-48, COL-109, COL-115, COL-145 (Fig. 7G-H, Table S1). Many of these genes have recognized predicted orthologues in mammals⁷¹. In conclusion, altering the material properties of the exoskeleton safeguards the age-dependent integrity of ECM and glia architecture.”

In Figure 7, legend:

“Predicted orthologues of the collagen genes and enzyme genes studied here are presented in detail elsewhere⁷¹.”

2) Fig S6: they say EMB-9 in magenta, but it looks green to me. Also, the job of a collagen is to be secreted and integrated into an ECM... for collagen EMB-9, it is integrated into the basement membrane. Unclear why the authors measure these green dots that look like intracellular EMB-9 (are these dots artifacts of the transgene? Aggregation of nMG? in other words, do these green dots represent functional EMB-9? ...since ROI1 has less EMB-9 based on these green dots (intracellular?), this suggests may be less synthesis or faster degradation of EMB-9 in that region... can you rescue this by *hsp-1* RNAi? In other words, is the lower EMB-9 due to the overactivation of chaperon *hsp70c* leading to degradation or turnover of EMB-9?

Response:

We thank the reviewer for pointing out this mistyping, we have now corrected it as follows:

In Figure S6 Legend:

“A-D. EMB-9 (green) localization is impaired in *unc-23* mutants compared to wild-type animals, accumulating in the posterior part of CEPsh glia (magenta).”

Regarding the concern on the localization of the collagen EMB-9, we visualize endogenous localization of EMB-9 using a CRISPR knock-in strain generated and characterized by Keeley et al, 2020 (PMID: 32585132), as we explain and cite in our manuscript. . Thus, since there is no overexpression, the visualized localization is expected to be functional EMB-9 and not an artifact. Also, the utilized knock-in strain has no phenotypic defects (unlike *emb-9* mutants) thus the tagged allele is expected to be functional. A similar (dotted) pattern of tagged EMB-9 is observed in L4/adult stages in the study of Shao et al, 2017 (PMID: 32255430) using a EMB-9::mCherry translational reporter by Ihara et al., 2011. These previous studies reported that this (dotted) pattern of EMB-9 arises due to its increased levels after the early larval stages.

Following the suggestion of the reviewer, we performed experiments of *hsp-1* RNAi, to see if this affects the EMB-9 localization we observe in *unc-23* mutants

Our findings suggest that abnormal localization of tagged EMB-9 in *unc-23* mutants may result from abnormal aggregation rather than abnormal synthesis or degradation. RNAi knock-down of EMB-9 in *unc-23* mutants suppresses the defects which is contrary to the hypothesis of decreased synthesis/ faster degradation of EMB-9 in *unc-23(m)*.

Nevertheless, we performed the suggested experiment, of assessing EMB-9 in *unc-23(m)* after knocking down *hsp-1*. *hsp-1* knock-down by RNAi or genetic mutant suppresses the EMB-9 defects in *unc-23* mutants (revised Figure S7E). Thus, the abnormal localization pattern of EMB-9 matrix in *unc-23* mutant is indeed due to the overactivation of Hsp70 chaperon. We thank the reviewer for this suggested experiment, that strengthens our manuscript. These results are now presented in the revised figure S6-7 and discussed in the text, to read as follows:

In Results, in page 10:

“Moreover, *hsp-1* knock-down (suppressing CEPsh glial cells defects) also suppresses the ECM defects of Perlecan/UNC-52 and Collagen/ EMB-9 localization in *unc-23* mutants (Fig. S7A-C). Based on all these observations combined, the ECM components appear to act downstream of UNC-23/HSP proteostasis and upstream of CEPsh glia for maintenance of their glial integrity.”

In Discussion, in page 14:

“Overall, our findings suggest that UNC-23/BAG2, through conserved functions and together with the HSP-1/Hsp70/Hsc70, may integrate tissue responses to temperature, caloric restriction and biomechanics to safeguard the robust integrity of ECM, glial cell and circuit components upon age progression. The exact mechanistic underpinnings of this integration remain to be further elucidated.”

In Figure S7, legend:

“ECM defects in *unc-23* mutants are affected by HSP-1, heat and caloric restriction but do not result from locomotion differences.

(A-C) *hsp-1* knock-down by RNAi or *hsp-1(ra807)* mutation suppresses the localization defects of Perlecan/UNC-52 (A-B) and Collagen/ EMB-9 in *unc-23* mutant L4 animals (C).

This is quantified by the % of animals in populations of different genotypes/conditions that present the Perlecan/UNC-52 defects in *unc-23* mutants, as observed per Fig. 5 (A), or by quantifications of relative sum intensity of Perlecan/UNC-52 content in ROI₂/ROI₁ (posterior/anterior of glia,) as initially quantified in Fig. 5B-C (B). The Collagen/ EMB-9 localization is examined using quantification of its relative sum intensity in the ROI₂/ROI₁ (posterior/anterior of glia), as initially quantified in Fig. 5B-C (B). (n ≥ 100 animals in A, C. n ≥ 6 animals in B, t-test, p values as in Fig. [...])”

3) *BTW, next time, if the authors want a blind review, they should remove all their info: p12. line 472: “Further information and requests for information, reagent and resource sharing should be directed to and will be fulfilled by the Lead Contact, Georgia Rapti (grapti@embl.de).”*

Response:

We thank the reviewer for the advice. We do not strictly require a blind review and have now opted to include information on the authors.

Reviewer #4 (Remarks to the Author):

In this manuscript, the authors use fluorescence-based markers to track glial cell development and architecture in C. elegans. Using classical mutagenesis screens, tissue-selective RNA interference and fluorescence microscopy, the authors show that unc-23/BAG2 is required in epithelial cells for correct CEPsh glial formation, positioning and association with neuronal and epithelial junctions. Defective glial length/volume and positioning in unc-23 mutants are dependent on aberrant localisation of the ECM protein, perlecan (UNC-52), collagen accumulation and tissue mechanics. The work is thorough, convincing and exciting, but the links to other proteostasis factors requires further interrogation. Conceptually the work is suitable for publication in Nature Communications, but the authors should perform some additional experiments to strengthen their conclusions before publication.

Response:

We thank the reviewer for compliment on our work quality, for their excitement and their positive consideration. We value the reviewer's suggestions for additional experiments that can provide new insights and strengthen our manuscript, and we have performed the applicable experiments. Here and in the revised manuscript, we present the experimental results and answers to all the points of the reviewer. Importantly, we also shortened some sections subtitles according to communicated formatting instructions.

Major comments

1. In Figure 3, the authors show that AIY neuron and Amphid neuron morphology is disrupted in unc-23(e25) mutants. Are these phenotypes also rescued by expression of wild type unc-23 in glial cells (as is observed with glial phenotypes).

Response:

The CEPsh glial cell defects of the *unc-23* mutant are rescued by expression of *unc-23* in epithelial cells and not rescued by providing it in glial cells. So, the experiment is not applicable as suggested. However, along the lines of the suggested experiment, we quantified the rescue of neuron phenotypes in *unc-23* mutant lines expressing the *unc-23* cDNA in epithelial cells (which also rescue the glial defective phenotypes). As expected, this *unc-23* expression rescued the neuronal defects (revised Figure 4D). We present the new results in the revised Figure 4D and discuss them in the text of our revised manuscript as follows:

In Results, in page 8:

“Specific expression of *unc-23* cDNA in epithelia, or the epithelial-expression of mouse BAG2, but not *unc-23* expression in CEPsh glia or muscle, rescues the CEPsh glia defects in *unc-23* mutants (Fig. 4C). Epithelial expression of *unc-23* cDNA also rescues the neuronal defects observed in amphid sensory neurons of *unc-23* mutants (Fig. 4D). Thus, age-dependent integrity of CEPsh glia and circuit components requires conserved functions of UNC-23 as a BAG2 cochaperone in epithelia, that neighbor CEPsh glia.”

In Figure 4, Legend:

Epithelial UNC-23 acts with HSP-1 and DNJ-13 to regulate glial integrity in relation to temperature and nutrition.

A-C. CEPsh glia defects (schematics, A) in *unc-23* mutants are suppressed by loss of HSP-1 or DNJ-13 using mutants or post-embryonic RNAi (B), mimicked by post-embryonic RNAi of *unc-23* (C), rescued by UNC-23 expression from epithelia, but not from glia or muscle, and rescued by mouse BAG2 (C). (D) Epithelial cell rescue of *unc-23* also rescues the neuronal defects observed in amphid sensory neurons. Color-coding of amphid neuron phenotypes as presented in Figure 3N. [...]

In Discussion, in page 13:

“The UNC-23/BAG2 acts for age-progressive disruption of CEPsh glial cell architecture while it appears largely dispensable for the development of the glial cell and circuit architecture, established by the L2 larvae stages. This age-progressive disruption of CEPsh glia integrity in *unc-23* mutants arises from compromised balance of the BAG2-Hsp70/Hsc70 chaperone system (Fig.4). BAG proteins interacting with Hsp70/Hsc70 promote substrate release, to inhibit or enhance chaperone activity depending on the substrate and cellular context⁴³. BAG proteins support client and nucleotide release while DNJ-13 supports the opposite Hsc70 conformation of an ADP client-bound state, *in vitro*^{43,61}. Upon loss of UNC-23 function, DNJ-13 may block nucleotide and client release, shifting the Hsc70 cycle out of balance. Indeed, *unc-23/BAG2* disruption causes defects in CEPsh glia integrity, that are suppressed by additional decrease of HSP-1/ Hsc70 or DNJ-13/DNAJ function.”

2. RNAi against *hsp-1* or *dnj-13* suppresses the glial defects in *unc-23(e25)* mutants. I found this interesting but also quite confusing. If working with UNC-23, why would depletion of HSP-70 or DNJ-13 not cause similar defects as UNC-23 knockdown? One possibility is that *hsp-1* or *dnj-13* RNAi can induce a heat shock response by activating HSF-1. This could then protect against loss of *unc-23/BAG2*. The authors should test if this is the case by performing experiments with *hsf-1*(RNAi) and *hsf-1(sy441)* mutants.

Response:

This is an interesting observation. Papsdorf et al.2014 (DOI 10.1074/jbc.M114.565234), also cited in our manuscript, demonstrated biochemically that DNJ-13 and BAG proteins promote Hsc70 substrate-binding and substrate release, respectively. Thus, UNC-23 and DNJ-13 oppositely regulate HSP-1, and loss of only one causes a misbalance of Hsc70 regulation. Along these lines, absence of Hsc70/HSP-1 alleviates CEPsh glial cell defects caused by disrupted Bag2/UNC-23, since there is no/less mis-regulated Hsc70/HSP-1. Thus, our results are in line with previous biochemical analysis of BAG2/DNAJ/Hsc70 complex.

Nevertheless, it would be interesting to know how HSF-1 acts in relation to HSP proteostasis effects, in line with the reviewer's experimental suggestions. We generated *unc-23(m); hsf-1(m)* double mutants and show that *hsf-1* loss suppresses the CEPsh glial defects of *unc-23(m)* mutants. So, HSF-1 affects Bag2/Hsc70 functions in this context. Unfortunately, despite trying two complementary strategies, triple mutant homozygotes *unc-23(m);hsp-1(m);hsf-1(m)* cannot be isolated due to lethality/ sterility. In line with the reviewer's suggestion, we also performed *hsf-1* RNAi knock-down on *unc-23; hsp-1* double mutants. This RNAi does not cause de-repression of CEPsh glial cell defects of *unc-23* mutants, that are

suppressed in *unc-23*; *hsp-1* mutants. RNAi knock-down of *hsp-1* or *djn-13* in *unc-23(m)*; *hsf-1(m)* mutants does not de-repress the CEPsh glial defects. The results are presented in the Figure below:

These results suggest HSF-1 may act upstream Bag2/Hsc70 here. Yet, since mutant combination of *hsp-1* and *hsf-1* mutations is inviable, we cannot exclude that absence of de-repression here is due to quantifications in viable animals with inefficient RNAi knock down (in case of lethality in *unc-23(m)* animals with efficient *hsf-1* knock-down). Thus, we cannot confidently conclude in a direct or indirect function of HSF1 in this context. We opt to not add these data in our manuscript as they may be inconclusive, unless it is required by the reviewer. We discuss these considerations in the text as follows:

In Results, in page 8:

“Otherwise, the HSF-1 transcription factor can also affect UNC-23 function since loss of HSF-1 partially suppresses the CEPsh glial defects of *unc-23(m)* (F. Caroti & G.Rapti, unpublished observations). Yet, whether these effects are direct/indirect, upstream/downstream of the Bag/HSP function, remains unclear since triple mutants *unc-23*; *hsp-1*; *hsf-1* cannot be isolated due to lethality/.”

3. As mentioned above, the impact of *hsp-1* and *djn-13* RNAi on glial phenotypes is very interesting. I agree that the simplest conclusion is that HSP-1 and DNJ-13 are acting in the same tissue as *unc-23*. However, the authors should perform *hsp-1* and *djn-13* RNAi selectively in glia (as was done for *unc-23*) to determine whether this is the case.

Response:

As mentioned in answer to the first question above, the CEPsh glial cell defects in the *unc-23* mutant are rescued by providing *unc-23* cDNA in epithelial cells and not by providing it in glial cells. Thus, the suggested experiment of knock-down in glial cells is not applicable. Our model suggests that HSP-1 and DNJ-13 acting together with UNC-23 function in epithelia and not in the glial cells. In line with our hypothesis, glial phenotypes of *unc-23* mutants are impacted by knock-down of *hsp-1* and *djn-13* in animals with CEPsh glia insensitive to RNAi, where these genes are knocked-down in epithelia but not CEPsh glia (shown in Figure 4B). We are now better explaining this in our text as follows:

In Results, in page 9:

“The HSP-1/Hsp70/Hsc70, DNJ-13/DNAJB1 proteins acting with BAG2//UNC-23 also function non-cell-autonomously since their knock-down suppresses *unc-23* mutant defects, upon RNAi in animals with CEPsh glia insensitive in glia effects (Fig. 4B).”

4. Exposure of animals to low-level chronic heat shock does not appear to have much effect on glial phenotypes. The authors should also test the effect of an acute, strong heat shock (e.g. 35C for 30 minutes).

Response:

Based on our experiments, exposure of embryos or L1 to “low-level chronic heat shock” significantly enhances glial cell defects in *unc-23* mutant L4 animals (compared to non-exposed mutants) (Figure 4E). (No differences in adults may indicate acceleration of age-progressive defects). Following the reviewer’s suggestion, we exposed animals to acute, strong heat-shock (35C for 30 minutes, revised Methods) to assess effects on glial phenotypes. Acute-strong heat shock exposure also enhances glial cell defects in *unc-23* mutants, (see revised Figure 4E). We also sought to provide more mechanistic insight into the effect of heat-shock in enhancing glial defects by assessing quantitatively its effect on ECM. Interestingly, heat-shock exposure significantly enhances the Perlecan defects in L2 animals of *unc-23* mutants, in line with our model for an interplay between HSP proteostasis – ECM - glia integrity (Figure S7B). These results are added in the revised Figure 4E-F and a new Figure S7B and discussed in the text of Results as follows:

In Results, in page 8:

“Interestingly, applying low-level chronic heat shock or acute, strong heat-shock in *unc-23* mutants (Methods) enhances the disruption of glial cell integrity in the earlier stages of L2 animals (Fig. 4E).”

[...]

the mild defects of Perlecan localization observed in L2 *unc-23* mutants are enhanced upon temperature increase and suppressed upon starvation, compared to untreated mutants (Figure S7D-E). [...] Thus, the progressive ECM defects of *unc-23* mutants, are dependent in HSP-1/Hsc70 function, accelerated by temperature increases and delayed by caloric restriction, while such changes do not affect the robust ECM localization in wild-type animals. Altogether, ECM components act upstream of CEPsh glial cell integrity, accumulate abnormally in *unc-23* mutants because of abnormal HSP function, are affected by temperature and nutritional changes, they correlate with and are followed by defective CEPsh glia, and altering their level suppresses CEPsh glial defects in *unc-23* mutants.”

In Discussion, in page 14:

“Overall, our findings suggest that UNC-23/BAG2, through conserved functions and together with the HSP-1/Hsp70/Hsc70, may integrate tissue responses to temperature, caloric restriction and biomechanics to safeguard the robust integrity of ECM, glial cell and circuit components upon age progression. The exact mechanistic underpinnings of this integration remain to be elucidated.”

In Figure 4, Legend:

[...]

(E-F) The environment's temperature or caloric changes can affect glial cell integrity in *unc-23* mutants. Long-term or short and acute temperature increase (in embryos, L1, L2 animals), accelerates the glial cell defects of *unc-23* mutants in phenotyped L4 animals (E) while to the contrary, starvation can partially suppress and delay these defects (F). $n \geq 3$ independent experiments, ≥ 150 animals. HS, heat shock. Error bars, p-value, ns as in Fig. 2.”

In Figure S7, legend:

“(A-C) *hsp-1* knock-down by RNAi or *hsp-1(ra807)* mutation suppresses the localization defects of Perlecan/UNC-52 (A-B) and Collagen/ EMB-9 in *unc-23* mutant L4 animals (C). This is quantified by the % of animals in populations of different genotypes/conditions that present the Perlecan/UNC-52 defects in *unc-23* mutants, as observed per Fig. 5 (A), or by quantifications of relative sum intensity of Perlecan/UNC-52 content in ROI₂/ROI₁ (posterior/anterior of glia,) as initially quantified in Fig. 5B-C (B). The Collagen/ EMB-9 localization is examined using quantification of its relative sum intensity in the ROI₂/ROI₁ (posterior/anterior of glia), as initially quantified in Fig. 5B-C (B). ($n \geq 100$ animals in A,C. $n \geq 6$ animals in B, t-test, p values as in Fig.

(D-E) The mild defects of Perlecan/UNC-52 observed in L2 animals of *unc-23* mutants are enhanced by subjecting animals to acute temperature-increase and suppressed by subjecting animals to starvation (at the L1 stage). Defects of Perlecan/UNC-52 in L4 animals of *unc-23* mutants are similar in all conditions. No such Perlecan/UNC-52 defects are observed in wild-type animals subject to these conditions of temperature increase or starvation. ($n \geq 100$ animals for D, $n \geq 10$ animals for E).

(F) L2 or L4 animals of *unc-23* mutants present similar locomotion (as quantified by body bends per minute) in normal conditions or after starvation at L1 stage. ($n \geq 10$ animals, bends measured for 1 minute/animal), t-test, p values as in Fig. 2.”

Methods, page 17

“Synchronized animals are exposed to 30 °C for 5 hours (long-term, low heat-shock) or 35 °C for 0,5 hours (acute, strong heat-shock), during stage of interest.”

5. The links between mechanical stress/impact and glial architecture are very interesting. Are the effects of starvation observed in Figure 4 simply because the worms now move less?

Response:

This is an interesting idea and we performed experiments to address it. We subjected synchronized *unc-23* mutant populations to starvation or normal bacteria feeding, and assessed the locomotion of these populations at the L2 and L4 stages. The locomotion was assessed by counting the number of bends per minute in each animal, for several animals of each populations. No significant differences were detected in the locomotion between well-fed and starved *unc-23* mutant populations (new Figure S7B). To provide insight into how starvation suppresses the glial cell defects, we assessed whether it affects their observed ECM changes. Interestingly, the defects of Perlecan are significantly decreased in L2 animals of *unc-23* mutants subject to starvation compared to well-fed mutants. Thus, the starvation delays both the ECM defects and subsequent CEPsh glial cell defects. In our revised manuscript, we present these results in a new Figure S7B and discuss in the text as follows:

In Results, in page 10:

“Importantly, no altered locomotion of L2 or L4 animals (as quantified by body bends per minute) is detected in starved versus well-fed *unc-23* mutants (Fig. S7E) suggesting that starvation affects UNC-23/BAG2 functions in glia integrity through mechanisms other than altered locomotion.”

In Figure S7, legend:

ECM defects in *unc-23* mutants are affected by HSP-1, heat and caloric restriction but do not result from locomotion differences.

[...]

(F) L2 or L4 animals of *unc-23* mutants present similar locomotion (as quantified by body bends per minute) in normal conditions or after starvation at L1 stage. ($n \geq 10$ animals, bends measured for 1 minute/animal), t-test, p values as in Fig. 2.

6. The lifespan of wild type animals in Figure S3C is very short. What were the conditions of this experiment? If conducted at 20C, these lifespan analyses should be repeated.

Response:

We thank the reviewer for pointing this out. We regret plotting the lifespan assays only until the last survival day of the mutants. We have replotted our lifespan experiments and performed additional ones, with the use of FuDR and without (revised Figure S3, Methods). In both conditions, the lifespan of *unc-23* mutants is significantly shorter than the one of wild-type animals, also based on the median lifespan and the “maximum lifespan” (5% survival) of the two populations. The FuDR use appears to increase the lifespan of both populations, in line with previous reports. Thus, the new plots and experiments align with our previous conclusions. Importantly we recognize that the mechanistic underpinnings of the lifespan changes in *unc-23* mutants remain to be elucidated. These results and considerations are presented in the revised Figure S3 and discussed in the text as follows:

In Results, in page 7:

“Moreover, *unc-23* mutants have shorter lifespan compared to wild-type animals, as indicated by their median and maximum lifespan (Fig. S3D). Thus, loss of CEPsh glia membrane integrity results in defects in brain axons and concomitant synaptic defects which deteriorate with age, from larvae to adult animals. Altogether, our results highlight that age-progressive integrity of CEPsh glia is key to maintaining circuit architecture including axon positioning and synapse density, and its disruption is associated with decreased animal lifespan. The mechanistic underpinnings of these effects remain to be further elucidated.”

In Figure S3 legend:

“UNC-23/BAG2 acts in larval and adult stages and affects synapse aging and animal lifespan.

[...]

D. *unc-23* mutants show decreased lifespan compared to wild-type animals, as assessed by the median lifespan (50% survival) and the maximum lifespan (5% survival) of the two populations. Lifespan assays were performed with or without the use of FuDR. $n \geq 100$ animals per genotype. Results were analyzed and compared with the Kaplan–Meier method and the Wilcoxon rank sum test. $p \text{ value} < 0,05$ in lifespan experiments, with or without the use of FuDR.”

7. There are many examples of mislabelled panels in figure legends and incorrect references to figure panels in the main text (e.g. The legend of Figure S3 references panel D instead of panel C). The authors should go through their manuscript carefully and correct all these errors.

Response:

We thank the reviewer for alerting us about these mistakes, we have edited the manuscript to correct them.

Minor comments

1. Supplemental table headings have been cut-off

Response:

We thank the reviewer for pointing this out, we have worked to update the tables.

2. Legend in Figure 5A mis-aligned

Response:

We thank the reviewer for pointing this out, we have worked to update the figures.

REVIEWER COMMENTS

Reviewer #1 (Remarks to the Author):

The authors have addressed all my concerns regarding the initial submission, and I recommend to accept for publication.

Reviewer #2 (Remarks to the Author):

The authors addressed adequately most of my concerns. However, the ones below should be further examined:

1. The conclusion from the BLT calculation of “lower Brillouin shift and width in mutant animals” is confusing as the BLT is a ratio of width to shift and lower values for both may result in the increase rather than the decrease of the BLT. This point should be carefully addressed.
2. Is the stated $>0.5\text{GHz}$ spectral resolution the nominal value? Does this value include spectral broadening due to the NA? The absolute values of the Brillouin width are meaningless and need to be deconvolved, particularly as it is stated that “live quantitative imaging” is used in this work. For the same reason, the down shift of the Brillouin shift due to the NA should be pointed out. Finally, units (GHz) are missing in Figure S8, B and C.
3. While it is stated that “spatially highly resolved measurements are not very informative, as biologically we are interested in the overall tissue region neighboring the CEPsh glial membranes”, it is also stated that “Brillouin microscopy offers an unprecedented capability to infer these properties non-invasively, in a 3D and in high-resolution”. (1) So why high spatial resolution is desirable if only spatially averaged data is analyzed? Would imaging with a lower NA be sufficient for this research? This point should be carefully addressed so that Brillouin microscopy is not oversold. (2) For clarity, please replace “high-resolution” with “high spatial resolution”, so there is no confusion with the spectral resolution.
4. While the analysis of the Brillouin shift is presented, no images of the Brillouin shift are shown. Why? The authors may reconsider adding such images.

Reviewer #3 (Remarks to the Author):

The authors have addressed and corrected all my concerns.

Reviewer #4 (Remarks to the Author):

In their revised manuscript, Coraggio et al. include additional experiments and revisions to the text that address all of the points I originally raised. Specifically, the authors now show that epithelial expression of *unc-23* is sufficient to rescue neuronal defects as well as glial defects in *unc-23(e25)* mutants, that the suppression of glial defects in *unc-23* mutants upon loss of *hsp-1* or *dnj-13* is not due to activation of the heat shock response (HSR), and that effects of starvation on CEPsh glia phenotypes is not due to differences in movement (i.e. not due to altered levels of mechanical stress). Furthermore, the authors include data demonstrating that the enhancement of glial defects by heat stress correlates with increased perlecan defects, while starvation suppresses perlecan defects. These additions/revisions strengthen the authors original conclusions and improve the paper. However, there are still some minor issues that need to be addressed before publication (please see below).

Specific points to address

1. It is great that the authors include experiments to show that expression of WT *unc-23* in epithelial cells is sufficient to suppress glial defects caused by *unc-23(e25)*. However, the new data have been added to Figure 4 with no label – presumably it would be Fig. 4F. The authors should correct this in the text and figure legend. In addition, the new data states “epidermis” rather than “epithelial” *unc-23*. Was this experiment performed the same way as for measuring glial morphology? If so, the authors should use the same labelling. If not, why was this the case?

2. I appreciate the authors attempts to ascertain whether the activation of the HSR is responsible for the suppression of glial defects upon *hsp-1* or *dnj-13* RNAi as these are tricky experiments with the potential for synthetic lethality (as was observed by the authors).

The finding that loss of *hsf-1* function also suppresses glial defects in *unc-23(e25)* animals is interesting and in keeping with the authors *hsp-1* and *dnj-13* data, as the expression of both genes will be reduced in *hsf-1(sy441)* mutants. In addition, the *unc-23(e25); hsp-1(ra807); hsf-1* RNAi data is good evidence that the suppression of glial defects by the loss of *hsp-1* is not simply due to activation of the HSR.

The caution shown by the authors in the interpretation of their *hsf-1* RNAi data is great. However, it may simply be that the absence of lethality in *unc-23(e25);hsp-1(ra807);hsf-1(RNAi)* animals is because functional knockdown of *hsf-1* is not achieved until after the mid-L2 stage (which would not necessarily undermine the authors conclusions). Furthermore, the experiments with *dnj-13* RNAi + *hsf-1(sy441)* also support the conclusion that activation of the HSR is not responsible for suppression of glial defects. Therefore, I suggest that the authors include these new data in the manuscript.

3. The data showing that acute HS also enhances glial defects is important and the additional evidence that this is linked to increased perlecan defects (while starvation suppresses perlecan defects) adds weight to the conclusion that perlecan destabilisation is at the heart of the glial defects observed. However, these new data are currently in Fig. 4C and Fig. S7D, not Fig. 4D and S7D-E as stated. This should be corrected in the text and legends.

4. The experiments presenting motility (body bends) in fed and fasted *unc-23* mutants is presented in Figure S7F, not Figure S7B. This should be corrected in the main text and legend.

5. In the lifespan data shown, the controls are not behaving as expected. Wild type animals appear to be dying before the end of reproduction and the median lifespan is approximately 2-4 days lower than would be expected. Furthermore, the authors state that lifespan assays were performed in the presence of carbenicillin and UV irradiated bacteria (i.e. non-proliferating bacteria). As such, median lifespans should be even further increased (see Zhao et al., Nat. Comms. 2017, "Two forms of death...", for an example of this).

Also, how many times were these trials performed? The statistics for all lifespan trials should be included in a supplementary table that includes the numbers of animals censored and scored (I could not find this in the revised manuscript). Furthermore, the median and maximal lifespans for the trial presented should be quoted in the accompanying figure legend.

My concern is that the authors are losing worms to high levels of "bagging", "rupturing" or some other phenotype that should be censored, and that this is creating the large difference in lifespan that the authors see. I understand that the use of FUDR will prevent bagging, but the concentration used is incredibly high and could instead cause other issues.

Whatever the cause, the authors should report full lifespan statistics for all trials performed. Alternatively, given that this experiment is not central to the authors overarching conclusions, the lifespan experiments could be removed.

REVIEWER COMMENTS

Reviewer #1 (Remarks to the Author):

The authors have addressed all my concerns regarding the initial submission, and I recommend to accept for publication.

Response:

We thank the reviewer for their positive recommendation.

Reviewer #2 (Remarks to the Author):

The authors addressed adequately most of my concerns. However, the ones below should be further examined:

1. The conclusion from the BLT calculation of “lower Brillouin shift and width in mutant animals” is confusing as the BLT is a ratio of width to shift and lower values for both may result in the increase rather than the decrease of the BLT. This point should be carefully addressed.

Response:

We apologize for not being sufficiently clear and elaborate in our statement. Indeed, we realize that our statement “the relative Brillouin loss tangent was significantly lower in mutants (Methods, Fig. 7F, S8-now revised name S9), in line with the lower Brillouin width and shift in the mutants.” might be ambiguous or confusing. As is obvious from Fig. S8 (now Fig.S9), the loss tangent is lower in mutants, together with both the Brillouin width and shift being lower. This is a consequence of the fact that, in mutants, the relative decrease in width is larger than in the shift, leading to an overall decrease in Brillouin loss tangent.

In order to be more clear in the main text, we have now slightly rewritten this statement to:

Results, Page 21:

“To further ascertain that the observed changes in Brillouin shift and linewidth do in fact have a mechanical origin, we performed additional measurements and analysis (Methods, Fig. 7, S9). Specifically, we performed refractive index measurements and computed the so-called Brillouin loss tangent (BLT), defined as $\tan(\varphi) = \Gamma/v$, which by its definition, does not depend on the sample refractive index and density (Ref. 53,56). We found that the average refractive index did not differ significantly between the mutant and wild-type animals, while the BLT was significantly lower in mutants (Fig. S9). This further confirms that the observed Brillouin spectral changes (Fig. 7F, S9A-D) are due to changes in mechanical properties, and not due to changes in either refractive index and/or density.”

2. Is the stated >0.5GHz spectral resolution the nominal value? Does this value include spectral broadening due to the NA? The absolute values of the Brillouin width are meaningless and need to be deconvolved, particularly as it is stated that “live quantitative imaging” is used in this work. For the same reason, the down shift of the Brillouin shift due to the NA should be pointed out. Finally, units (GHz) are missing in Figure S8, B and C.

Response:

We appreciate the technical follow-up questions, some of which are actually already answered in the Methods section of the revised manuscript. The spectral resolution stated is indeed the nominal value and does not include spectral broadening due to the NA (it was measured with the delta-like back-scattered laser light which is not subjective to NA broadening). We agree that the measured linewidth requires deconvolution, which we actually did for the analysis in this study, as stated in the Methods section of the revised version (specifically, in line 796 we wrote “To account for the finite spectral resolution, we subtract the latter (520MHz) from the measured linewidth values (the subtraction corresponds to deconvolution, as detailed in Chan et al, 2021)).

We further thank the Reviewer for pointing out the additional down-shift of the Brillouin shift because of the high NA objective used in this work. We are indeed aware of this effect as we have characterized it previously for this microscope (Bevilacqua et al., Biomed. Opt. Exp. 2019). For the effective NA (0.85) utilized in the present work, this downshift corresponds to ~100-110MHz. We now include this information in the Methods section of the newly revised manuscript:

Line 797: “We note that the effective NA of the objective (0.85) leads to ~100-110MHz downshift in the Brillouin shift.”

Finally, we thank the Reviewer for pointing out the missing unit - this has now been corrected (in the Figure S9, as by revised name).

3. While it is stated that “spatially highly resolved measurements are not very informative, as biologically we are interested in the overall tissue region neighboring the CEPsh glial membranes”, it is also stated that “Brillouin microscopy offers an unprecedented capability to infer these properties non-invasively, in a 3D and in high-resolution”. (1) So why high spatial resolution is desirable if only spatially averaged data is analyzed? Would imaging with a lower NA be sufficient for this research? This point should be carefully addressed so that Brillouin microscopy is not oversold. (2) For clarity, please replace “high-resolution” with “high spatial resolution”, so there is no confusion with the spectral resolution.

Response:

We appreciate the Reviewer’s comment, and realize additional information and clarifications are required.

Regarding (1), we agree that averaging the spatially resolved data appears counterintuitive at first sight, since we’re interested in tissue-scale properties. However, we would like to point out that imaging with a lower NA would quickly degrade the axial resolution and sectioning

capability (due to the NA^2 dependence), which would then make the measurements presented in this paper impossible, as the microscope's PSF would extend beyond the ROI (in the axial direction) and also capture tissue outside of the location of interest of our study (the epithelial cell neighborhood of CEPsh glia, at the depth of the pharyngeal bulb as external reference). In this respect, already going to an NA of ~ 0.5 would result in an axial resolution worse than $5\mu m$, which we regard as the relevant size of the ROI (see also Fig. 1A for CEPsh glia volume and animal cross-sections). Since this is anyway very close to the effective NA used in this work, we did not consider changing the NA any further.

Regarding (2), we thank the reviewer for pointing this out and have made the requested change.

4. While the analysis of the Brillouin shift is presented, no images of the Brillouin shift are shown. Why? The authors may reconsider adding such images.

We appreciate this comment and agree that Brillouin shift images could also be informative for the reader. Therefore for completeness, we included images of the Brillouin shift and loss tangent in Fig. S9A of the revised manuscript, and we revised the text of the accompanying figure legend accordingly.

Reviewer #3 (Remarks to the Author):

The authors have addressed and corrected all my concerns.

Response:

We thank the reviewer for their positive recommendation.

Reviewer #4 (Remarks to the Author):

*In their revised manuscript, Coraggio et al. include additional experiments and revisions to the text that address all of the points I originally raised. Specifically, the authors now show that epithelial expression of *unc-23* is sufficient to rescue neuronal defects as well as glial defects in *unc-23(e25)* mutants, that the suppression of glial defects in *unc-23* mutants upon loss of *hsp-1* or *dnj-13* is not due to activation of the heat shock response (HSR), and that effects of starvation on CEPsh glia phenotypes is not due to differences in movement (i.e. not due to altered levels of mechanical stress). Furthermore, the authors include data demonstrating that the enhancement of glial defects by heat stress correlates with increased perlecan defects, while starvation suppresses perlecan defects. These additions/revisions strengthen the authors original conclusions and improve the paper. However, there are still some minor issues that need to be addressed before publication (please see below).*

Specific points to address

1. It is great that the authors include experiments to show that expression of WT *unc-23* in epithelial cells is sufficient to suppress glial defects caused by *unc-23(e25)*. However, the new data have been added to Figure 4 with no label – presumably it would be Fig. 4F. The authors should correct this in the text and figure legend. In addition, the new data states “epidermis” rather than “epithelial” *unc-23*. Was this experiment performed the same way as for measuring glial morphology? If so, the authors should use the same labelling. If not, why was this the case?

2. I appreciate the authors attempts to ascertain whether the activation of the HSR is responsible for the suppression of glial defects upon *hsp-1* or *dnj-13* RNAi as these are tricky experiments with the potential for synthetic lethality (as was observed by the authors).

The finding that loss of *hsf-1* function also suppresses glial defects in *unc-23(e25)* animals is interesting and in keeping with the authors *hsp-1* and *dnj-13* data, as the expression of both genes will be reduced in *hsf-1(sy441)* mutants. In addition, the *unc-23(e25); hsp-1(ra807); hsf-1* RNAi data is good evidence that the suppression of glial defects by the loss of *hsp-1* is not simply due to activation of the HSR.

The caution shown by the authors in the interpretation of their *hsf-1* RNAi data is great. However, it may simply be that the absence of lethality in *unc-23(e25);hsp-1(ra807);hsf-1(RNAi)* animals is because functional knockdown of *hsf-1* is not achieved until after the mid-L2 stage (which would not necessarily undermine the authors conclusions). Furthermore, the experiments with *dnj-13* RNAi + *hsf-1(sy441)* also support the conclusion that activation of the HSR is not responsible for suppression of glial defects. Therefore, I suggest that the authors include these new data in the manuscript.

3. The data showing that acute HS also enhances glial defects is important and the additional evidence that this is linked to increased perlecan defects (while starvation suppresses perlecan defects) adds weight to the conclusion that perlecan destabilisation is at the heart of the glial defects observed. However, these new data are currently in Fig. 4C and Fig. S7D, not Fig. 4D and S7D-E as stated. This should be corrected in the text and legends.

4. The experiments presenting motility (body bends) in fed and fasted *unc-23* mutants is presented in Figure S7F, not Figure S7B. This should be corrected in the main text and legend.

5. In the lifespan data shown, the controls are not behaving as expected. Wild type animals appear to be dying before the end of reproduction and the median lifespan is approximately 2-4 days lower than would be expected. Furthermore, the authors state that lifespan assays were performed in the presence of carbenicillin and UV irradiated bacteria (i.e. non-proliferating bacteria). As such, median lifespans should be even further increased (see Zhao et al., Nat. Comms. 2017, “Two forms of death...”, for an example of this).

Also, how many times were these trials performed? The statistics for all lifespan trials should be included in a supplementary table that includes the numbers of animals censored and scored (I could not find this in the revised manuscript). Furthermore, the median and maximal lifespans for the trial presented should be quoted in the accompanying figure legend.

My concern is that the authors are losing worms to high levels of “bagging”, “rupturing” or some other phenotype that should be censored, and that this is creating the large difference in lifespan that the authors see. I understand that the use of FUdR will prevent bagging, but the concentration used is incredibly high and could instead cause other issues.

Whatever the cause, the authors should report full lifespan statistics for all trials performed. Alternatively, given that this experiment is not central to the authors overarching conclusions, the lifespan experiments could be removed.

Response:

We thank the reviewer for acknowledging our work and for their positive recommendation. We also appreciate the technical follow-up on the minor issues, and we address them below:

1. We thank the reviewer for pointing out this typographical issue in Figure 4. The panel label F is now included and the axis label is edited to read “*unc-23(e25) + unc-23* in epithelia”, since the rescue of neuronal phenotypes was assessed upon the exact same *unc-23* expression with the one used to assess rescue of glial phenotypes.

2. We agree with the reviewer’s points and since the reviewer suggests to included the data despite the challenging nature of the experiment, we now include the data in the manuscript in a new figure S4 and corresponding legend as below (and renamed accordingly all the next supplementary figures). We also added in the manuscript’s text, a short explanation of our observations, in line with the reviewer’s points, which reads as follows:

In Results, in page 13:

Otherwise, the HSF-1 transcription factor can also affect UNC-23 function since loss of HSF-1 partially suppresses the CEPsh glial defects of *unc-23* mutants (Fig. S4). Whether these effects are direct/indirect and upstream/downstream of the Bag/HSP function remains unclear, since triple mutants *unc-23; hsp-1; hsf-1* cannot be isolated due to lethality/sterility. Yet, our experiments suggest that the suppression of glial defects in *unc-23/Bag2* mutants by loss of HSP-1 or DNJ-13 is not simply due to activation of HSF-1 and heat shock response. Specifically, CEPsh glial defects are suppressed in double mutants *unc-23; hsp-1* upon RNAi knock-down of HSF-1, and in double mutants *unc-23; hsf-1* upon RNAi knock-down of HSP-1 or DNJ-13 (Fig. S4).

Legend Fig. S4

HSP-1 and DNJ-13 can affect roles of UNC-23/BAG2 in CEPsh glial integrity independently of HSF-1 activation

- A. CEPsh glia defects (as presented in schematics in Fig. 4A) in *unc-23* mutants are suppressed when mutating *hsf-1*, suggesting that HSF-1 affects UNC-23/BAG2 roles in CEPsh glial integrity. Post-embryonic RNAi knock down of HSP-1 or DNJ-13 suppress the CEPsh glia defects, even in absence of HSF-1, suggesting they can act independently of HSF-1 activation in this context. $n \geq 3$ independent experiments, ≥ 150 animals. Error bars, p-value, \neg ns as in Fig. 2.

3. We thank the reviewer for pointing this out, the panel numbers are now corrected, taking into account the new numbering or supplementary figures). The relevant figure panels are Fig. 4C and Fig. S8D-E (the latter includes 2 different methods of quantification for Perlecan upon HS treatment, as seen in genotype/ conditions of panel E).

4. We thank the reviewer for pointing this out, this has been corrected (represented in S8F).

5. We thank the reviewer for the detailed points on the lifespan assays, which we address below.

Both the *wild-type* and *unc-23(e25)* strains used in our lifespan contain an integrated array expressing Promoter[CEPsh>::myrGFP (it remains unclear if this may slightly affect lifespan). We realize that this may not have been clear previously. We have now clarified this in the manuscript, pointing out the genotype of the used strains in the legend of figure S3 and explaining the above considerations in the Methods. Regarding the result analysis, censorship of worms was appropriately performed (for all worms dried, exploded/ruptured, bagging, etc), numbers are calculated as presented in Methods. Besides, inter-lab variance in *C. elegans* lifespan of wild-type strains is observed across different studies, and may be due to slight alterations in worm handling and experimental design (PMID: 34793939).

We opted to include the lifespan experiment so as to not omit experiments initially included in the manuscript and because it can provide interesting information to the readers. Since the reviewer expressed concerns about the FuDR concentration, we have now repeated the experiment using a lower, commonly used concentration of 50uM FuDR. We only present this experiment in our revised manuscript. Experimental trials are n=3, total>100 animals per genotype. These details are now noted in Figure S3 legend and Methods. We also note the median and maximal lifespans in the Figure S3 legend. We include the primary data of lifespan assays in the new table S1 (and rename all other tables accordingly).

We hope these changes, that help ameliorate our manuscript, address all the reviewer's points.

REVIEWERS' COMMENTS

Reviewer #2 (Remarks to the Author):

[Editorial Note: Reviewer #2 confidentially signs off.]

Reviewer #4 (Remarks to the Author):

The authors have satisfactorily addressed all my points. In particular, the explanation for the short lifespan of their WT animals makes sense, and I am happy for the data to be included. I recommend the work for publication.